# Longitudinal localization of leukaemic stem cells between the metaphysis and central marrow governs their behaviour

Chen Wang [1,2,3,8], Yi Pan [1,2,3,8], Ruochen Dong[4], Wenxuan Zhou[1,2,3], Xiaduo Meng[1,2,3], Xi Kang[2], Ravi Nistala[1,5], Richard D. Hammer [6], Linheng Li [4,7] & XunLei Kang [1,2,3] ✉

Leukaemic stem cells (LSCs) reside in protective bone marrow (BM) niches that promote therapeutic resistance and relapse. Here we characterized longitudinal BM niches supporting LSC survival, distinguishing the metaphysis from the central marrow. Quiescent LSCs preferentially localized to the metaphysis and exhibited reduced stemness and aggressiveness upon mobilization to the central marrow. Targeting DPP4 in acute myeloid leukaemia (AML) cells altered CXCL12 gradients at three spatial scales. Systemically, reversal of the BM–peripheral blood CXCL12 gradient confined AML cells within the BM. At the BM level, disruption of the metaphysis–central marrow gradient displaced LSCs from their protective niche. At the microscale, loss of the CXCL12 gradient between N-cadherin⁺ stromal cells and the surrounding matrix impaired LSC recruitment. These effects arise from the CXCL12–DPP4–GPC3 axis, in which DPP4 truncates and inactivates CXCL12, whereas stromal GPC3 restrains DPP4 activity. Modulating this axis disrupts niche protection and enhances therapeutic vulnerability in AML.

Acute myeloid leukaemia (AML) is an aggressive haematopoietic malignancy driven by leukaemic stem cells (LSCs), which are central to disease initiation, progression and relapse. Thus, targeting LSCs has emerged as a promising strategy to combat AML effectively. LSC behaviour is governed by two critical factors: their trafficking through circulation and their localization within specialized bone marrow (BM) niches. However, despite extensive studies on LSC–niche interactions and growing translational efforts, clinically validated strategies that reliably restrict leukaemia dissemination or eradicate LSCs remain limited[1–6]. This highlights an incomplete understanding of the complex interplay between LSCs and their niches.

Previous research has extensively characterized how niche cells regulate LSCs through single-direction communication[3,7]. However, the dynamic cross talk between LSCs and niche cells that co-creates a leukaemia-supportive microenvironment remains poorly understood. In addition, two BM niches have been well described in regulating LSC behaviour: the endosteal niche, which promotes LSC quiescence and survival, and the sinusoidal niche, which facilitates LSC proliferation and dissemination[8]. The endosteal niche comprise two anatomically distinct regions: the metaphysis (trabecular bone) and the diaphysis (cortical bone). While interactions between LSCs and the diaphyseal endosteum at a local transverse scale (diaphysis versus sinusoid) have

[1]Division of Hematology and Oncology, Center for Precision Medicine, Department of Medicine, University of Missouri School of Medicine, Columbia, MO, USA. [2]Center for Precision Medicine, Department of Medicine, University of Missouri School of Medicine, Columbia, MO, USA. [3]Ellis Fischel Cancer Center at MU Health Care, University of Missouri, Columbia, MO, USA. [4]Stowers Institute for Medical Research, Kansas City, MO, USA. [5]Division of Nephrology, Department of Medicine, University of Missouri School of Medicine, Columbia, MO, USA. [6]Department of Pathology and Anatomical Sciences, University of Missouri School of Medicine, Columbia, MO, USA. [7]Department of Pathology and Laboratory Medicine, Division of Molecular Oncology, University of Kansas Medical Center, Kansas City, KS, USA. [8]These authors contributed equally: Chen Wang, Yi Pan. ✉e-mail: kangxu@health.missouri.edu

been well studied, the broader longitudinal scale (metaphysis versus sinusoid) remains underexplored.

CXCL12 is a well-established chemoattractant for leukaemic cells[9,10], playing a crucial role in cell trafficking and localization. DPP4, a membrane-bound extracellular peptidase, cleaves Xaa–Pro or Xaa–Ala dipeptides from the N terminus of substrates such as CXCL12, leading to its functional inactivation[11,12], has been implicated in promoting AML progression[13]. Conversely, glypican-3 (GPC3) has been identified as an endogenous inhibitor of DPP4[14,15]. However, how this DPP4–GPC3–CXCL12 regulatory axis operates within the leukaemic niche, and how it collectively governs AML stem cell localization and maintenance, remains largely unexplored.

Here, we provide evidence that longitudinal positioning within the BM—metaphysis versus central marrow (CM)—fundamentally shapes LSC fate. We show that metaphyseal LSCs remain quiescent and stem-like, while displacement into the CM induces cycling, loss of stemness, and apoptosis. Mechanistically, we identify a regulatory axis involving AML-derived DPP4, niche-derived CXCL12 and GPC3 from N-cadherin-expressing mesenchymal stromal/stem cells (N-cad+ MSCs) that governs these processes[16–18]. Importantly, mobilizing LSCs from metaphyseal niches with DPP4 inhibition not only induces their exhaustion but also reduces leukaemic dissemination, a strategy with immediate translational potential given the availability of US Food and Drug Administration-approved DPP4 inhibitors[13,19].

## Results

### Dpp4 deficiency redistributes AML cells and induces LSC exhaustion

Our group recently identified DPP4 as a promising therapeutic target for AML owing to its selectively high expression in AML cells and its role in blocking AML progression upon depletion[13]. Strikingly, our experiments revealed that *Dpp4* deletion alters the spatial distribution of GFP+ myeloid leukaemia cells within the BM, as observed in whole BM sections in both MLL-AF9 (MA9) and AML-ETO9a (AE9) models (Fig. 1a,b and Extended Data Fig. 1a). In mice transplanted with control AML cells (*Dpp4*+/+ AML mice), AML cells predominantly localize around the metaphysis area, especially the proximal metaphysis (PM). Notably, at equivalent leukaemic burden, *Dpp4*−/− AML cells displayed a more uniform distribution across the BM, with increased localization to the CM in both MA9 and AE9 models (Fig. 1a,b). To eliminate potential bias due to differences in AML progression because *Dpp4*-knockout (KO) AML mice rarely reach moribund stage and sustain minimal AML cells in peripheral blood (PB), we transplanted 1 × 10⁶ control or *Dpp4*-KO primary AML cells (GFP+) into C57/B6 recipients and compared AML distribution within the BM when PB GFP+ cells reach ~30% (Fig. 1c). This

tumour burden-matched approach ruled out the possibility that AML redistribution was secondary to delayed disease progression in *Dpp4*-KO mice. Consistently, *Dpp4* KO resulted in a similar redistribution of AML cells across the BM. This redistribution correlated with a significant numerical increase in LSC populations (L-GMPs: GFP+IL-7R−Lin−Sca-1−c-Kit+CD34+FcγRII/III+, see Extended Data Fig. 10 for gating strategy) in the BM of the MA9 model (Fig. 1d–f). By contrast, the frequency of c-Kit+Sca-1+ cells, c-Kit+Sca-1−IL-7Rα+ cells, leukaemic common myeloid progenitors (L-CMPs) and leukaemic megakaryocyte–erythroid progenitors (L-MEPs) was significantly reduced in *Dpp4*-KO mice compared with controls (Extended Data Fig. 1b–e). To evaluate the functional consequences of altered LSC localization, we performed colony-forming unit (CFU) assays. *Dpp4* depletion in L-GMPs resulted in a dramatically reduced serial-replating capacity, a defect also observed in other leukaemic cell populations (Fig. 1g and Extended Data Fig. 1f). Furthermore, analysis of LSC properties in the BM revealed that *Dpp4*−/− AML mice exhibited markedly increased cell-cycle activity and elevated metabolic features, indicating a substantial loss of LSC quiescence (Fig. 1h,i and Extended Data Fig. 1e). These findings are consistent with the previously reported reduction in LSC stemness upon *Dpp4* deletion[13]. Notably, LSCs from *Dpp4*−/− mice also displayed a significantly higher rate of apoptosis compared to those from *Dpp4*+/+ AML mice (Fig. 1j,k), aligning with the established role of DPP4 in supporting AML cell survival[13,20]. Together, these results demonstrate that *Dpp4* loss uncouples LSC proliferation from self-renewal by redistributing AML cells from metaphyseal niches into less supportive CM regions, thereby driving proliferation-associated LSC exhaustion.

### The PM and DM mark distinct niches versus CM for maintaining LSCs

While we observed that L-GMPs exhibit reduced quiescence and increased exhaustion-like behaviour after *Dpp4* KO, it is not clear whether this phenomenon results from the intracellular effects of *Dpp4* KO or is associated with an altered LSC niche. To understand this, we evaluated L-GMPs across the three anatomical partitions (PM, distal metaphysis (DM) and CM) of the BM in control AML mice. Our analysis revealed that the PM and DM harboured a significantly higher percentage of L-GMPs compared with the CM (Fig. 2a,b). Among the L-GMPs, in each niche, those derived from the PM and DM displayed lower cell division activity (Fig. 2c,d). To functionally assess niche-specific LSC properties, we transplanted equal numbers of L-GMPs isolated from PM, DM or CM regions into C57/B6 recipients. PM- and DM-derived cells produced a more aggressive AML progression (Fig. 2e), indicating that these regions harbour LSCs with higher functional leukemogenic potential. Furthermore, limiting dilution assays showed that PM/DM-derived AML cells

---

**Fig. 1 | *Dpp4* deficiency redistributes AML cells and induces LSC exhaustion. a,b**, Representative images of BM sections from recipients transplanted with *Dpp4*+/+ (left) or *Dpp4*−/− (right) primary AML cells, including MLL-AF9 (MA9, **a**) and AML-ETO9a (AE9, **b**). For the MA9 model, these images were captured at weeks 2 and 12 after transplantation and depict similar AML cell compositions in the BM (~30%). Images for the AE9 model were obtained at weeks 10 and 20 after transplantation. Green, GFP+ AML cells; red, endomucin staining blood vessels; blue, DAPI for nuclei. The calculated proportion of GFP+ signals in each of the three anatomical BM areas is listed to the left of the images. The dashed box indicates the area of focus. Scale bars, 100 μm. **c**, Schematic representation of the secondary AML transplantation models. In total, 1 × 10⁶ primary AML cells were transplanted into sublethally irradiated recipients, and AML distribution within the BM was compared when the AML mice reached approximately 30% GFP+ cells in the PB. **d**, Summary of secondary MA9 and AE9 GFP+ signals in each BM area (PM, CM and DM), showing that *Dpp4*+/+ AML cells are more heavily concentrated in the PM, while *Dpp4*−/− AML cells are more evenly distributed throughout all BM areas. For the MA9 model, *n* = 5 BM sections from five mice were analysed. *Dpp4*+/+ BM sections were collected at days 14, 15, 15, 16 and 17, whereas *Dpp4*−/− BM sections were collected at days 28, 29, 30, 31, 31 and 33.

For AE9-transplanted mice, *n* = 5 BM sections from five mice were analysed. *Dpp4*+/+ BM sections were collected at days 34, 35, 35, 36 and 40, whereas *Dpp4*−/− BM sections were collected at days 65, 66, 67, 70 and 73. Scale bars, 100 μm. **e,f**, LSC population (GFP+IL-7R−Lin−Sca-1−c-Kit+CD34+FcγRII/III+, GMP-like leukaemic cells, L-GMP), measured by flow cytometry in the BM of *Dpp4*−/− and *Dpp4*+/+ AML mice at 2 weeks after transplantation (*n* = 5 biologically independent mice per group), where panel **f** shows the quantification of panel **e**. **g**, Comparison of the CFU capability of *Dpp4*+/+ and *Dpp4*−/− L-GMP cells during serial replating (500 cells per well, *n* = 6 biologically independent wells per group). **h,i**, Cell cycle analysis of MLL-AF9 L-GMP in *Dpp4*+/+ and *Dpp4*−/− mice (*n* = 5 biologically independent mice per group) by flow cytometry, where panel **i** shows the quantification of panel **h. j,k**, Flow cytometry analysis of early (detected as Annexin V+/7-AAD− staining) and late (detected as Annexin V+/7-AAD+ staining) apoptotic L-GMP from the BM of MLL-AF9 mice (*n* = 5 biologically independent mice per group), where panel **k** shows the quantification of panel **j**. Data are presented as mean ± s.e.m. Statistical significance was determined using a two-sided unpaired Student's *t*-test. Exact *P* values are provided in the graphs. Icon in **c** created in BioRender; Kang, X. https://biorender.com/34u8lmg (2026).

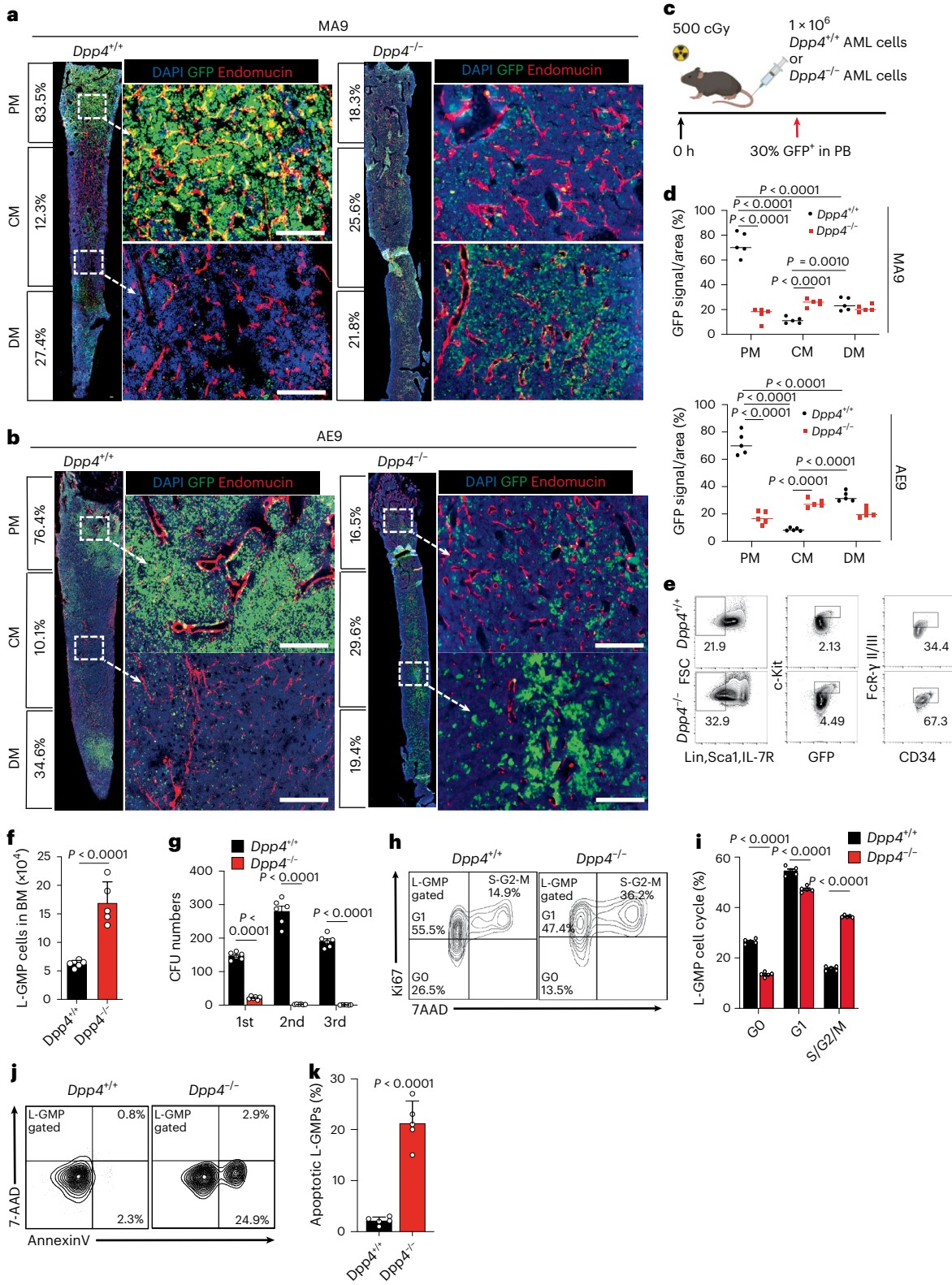

possessed higher stemness (Fig. 2f and Extended Data Fig. 2). Together, these findings indicate that the PM and DM, as opposed to the CM, constitute specialized LSC-supporting niches that promote LSC maintenance and function, as demonstrated by their in vivo leukemogenic capacity.

### *Dpp4* deficiency confines AML cells within the BM through reversed BM-to-PB CXCL12 gradient

Another hallmark finding in *Dpp4*-KO AML mice is the dramatic decrease in peripheral AML cells (GFP⁺) and progenitors (Lin⁻GFP⁺),

which correlates with their increased enrichment within the BM after *Dpp4* deletion in both MA9 and AE9 models (Fig. 3a and Extended Data Fig. 3a,b). Deleting Dpp4 consistently reduced leukaemic blast infiltration in the spleen, preserving its normal architecture. This contrasts with the extensive disruption of the spleen seen in wild-type (WT) AML mice (Extended Data Fig. 3c,d). Complete blood count assay showed that Dpp4⁻/⁻ AML-transplanted mice preserved normal haemoglobin, leukocyte and platelet levels, indistinguishable from healthy controls (Extended Data Fig. 3e). To investigate whether

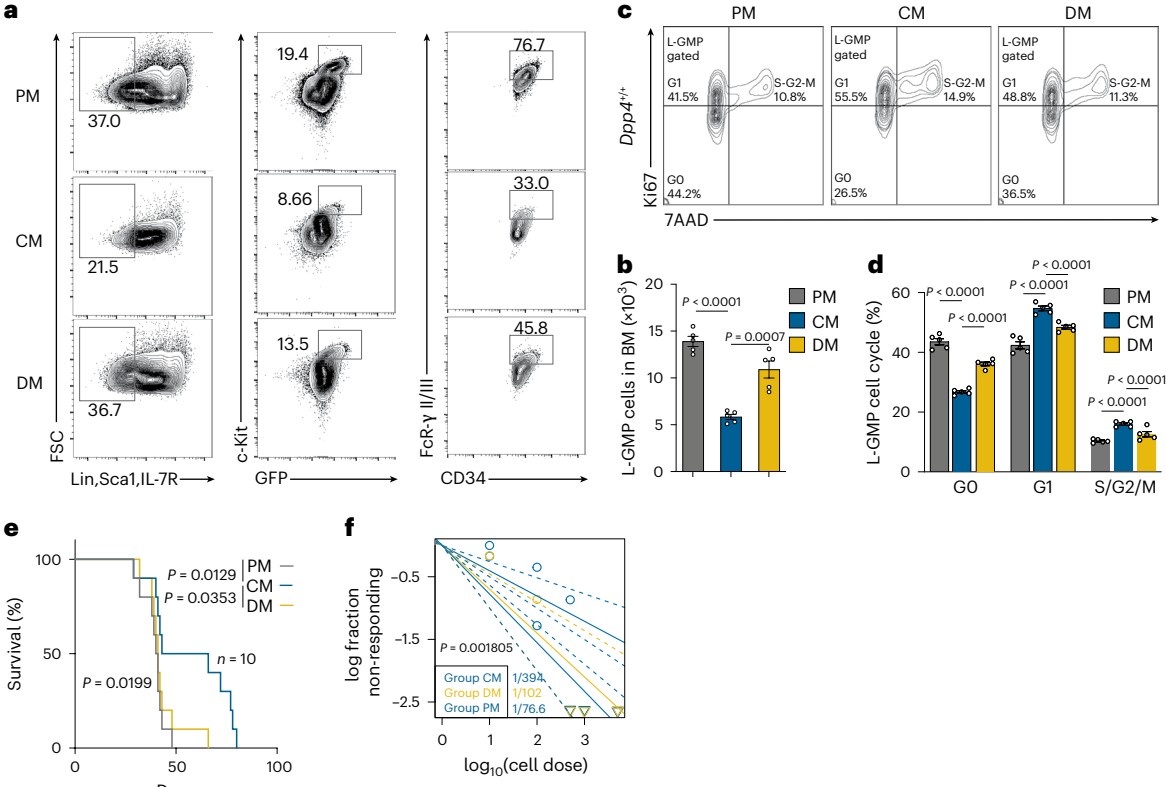

**Fig. 2 | The PM and DM mark distinct niches versus CM for maintaining LSCs.**
**a,b**, L-GMP measured by flow cytometry in the PM, CM and DM areas of $Dpp4^{+/+}$ AML mice at 2 weeks after $1 \times 10^6$ MA9 cell transplantation as shown in Fig. 1c (mean ± s.e.m., $n = 5$ biologically independent mice per group), where panel **b** shows the quantification of panel **a. c,d**, Cell cycle analysis of MA9 L-GMPs in PM, CM and DM areas from $Dpp4^{+/+}$ mice ($n = 5$ biologically independent mice per group) by flow cytometry, where panel **d** shows the quantification of panel **c. e**, Kaplan–Meier survival curves of secondary recipient mice ($n = 10$ per group) transplanted with 500 L-GMPs sorted from the PM, CM and DM of $Dpp4^{+/+}$ AML donor mice. Survival distributions were compared using the Mantel–Cox (log-

rank) test ($\chi^2 = 7.835$, d.f. 2, $P = 0.0199$). **f**, log–log plot showing the fraction of non-responding mice versus the number of transplanted cells ($\log_{10}$ scale). Points represent the proportion of leukaemia-free mice at each dose, and the fitted curves are derived from limiting dilution analysis (LDA). The tumour-initiating cell (TIC) frequency for each group is shown (PM: 1/76.6, CM: 1/394, DM: 1/102), with statistical comparison between groups indicating a significant difference (CM versus PM $P = 0.0001811$; CM versus DM $P = 0.008248$). Data are presented as mean ± s.e.m. Statistical significance was determined using a two-sided unpaired Student's $t$-test. Exact $P$ values are provided in the graphs.

$Dpp4$-KO AML cells exhibit impaired homing or show reduced peripheralization despite successful homing, we compared the homing efficiency of control and $Dpp4$-KO AML cells 16 h after transplantation. Our findings indicate comparable homing of control and $Dpp4$-KO AML cells to all haematopoietic organs (Fig. 3b), suggesting that $Dpp4$ deletion does not impair homing but instead restricts AML cells within the host BM. This restriction persists even in the late stage of AML development (Fig. 3a) and resulted in prolonged mouse survival (Fig. 3c).

However, this finding challenges the existing knowledge that the CM, rich in sinusoids, serves as a crucial niche for LSCs and haematopoietic stem cells (HSCs) trafficking to the PB[7,8]. To explore the mechanism behind this intriguing observation, we next focused on the MA9 AML model. We first hypothesized that $Dpp4$ KO might impair AML cells' access to blood vessels. However, immunofluorescence analysis revealed that both GFP-labelled control and $Dpp4$ KO AML cells were equidistant from endomucin-labelled vessels throughout the BM, indicating unimpeded vascular access[21,22] (Fig. 3d–f). This prompted us to hypothesize that altered cytokine/chemokine gradients might be responsible for the observed AML cell confinement. We focused on DPP4 substrates and cytokines or chemokines implicated in AML cell trafficking[23–27]. Among the factors tested, including CCL11, CCL22, CXCL12, IL-6, IL-15 and so on, only CXCL12 exhibited a reversed gradient from plasma to BM in $Dpp4$-KO mice (Fig. 3g–j). Despite prior reports implicating G-CSF and NPY in haematopoietic stem and progenitor cells (HSPC) mobilization[28–30], their levels were comparable between BM extracellular fluid (BMEF) and plasma in our model, excluding their

role in the observed phenotype. Exogenous CXCL12 administration mobilized both control and $Dpp4$-KO AML cells at high doses (Fig. 3k), but $Dpp4$-KO cells showed tolerance at lower doses. This was not due to altered CXCR4 expression (Fig. 3l), or migration capacity (Fig. 3m,n), suggesting that the reversed gradient itself—not cell-intrinsic changes—drives confinement. Overall, our data show that $Dpp4$ loss traps AML cells in the BM by generating a reversed CXCL12 gradient, rather than by impairing homing, vascular access or chemokine responsiveness.

### Niche-specific CXCL12 production by N-cad⁺ MSCs dictates LSC spatial distribution

Our previous findings demonstrated that a reversed BM-to-PB CXCL12 gradient confines $Dpp4$-KO AML cells to the BM. To dissect the mechanism driving AML redistribution, we quantified CXCL12 levels across BM subregions. Strikingly, $Dpp4$-KO mice exhibited a reversed PM/DM–CM CXCL12 gradient, with low CXCL12 in the CM of controls but elevated levels in $Dpp4$-KO mice (Fig. 4a,b). This suggested compartmentalized CXCL12 regulation by DPP4, prompting us to identify its cellular source.

To identify the specific niche cells secreting CXCL12, we performed single-cell RNA sequencing (scRNA-seq) on non-haematopoietic PM and DM cells from AML mice 3 weeks after transplantation. Through the analysis of 11,512 cells (median of 19,367.5 molecules and 4,391 genes per cell), we identified 14 distinct clusters, spanning MSCs (marked by $Lepr$, $Cdh2$ and $Cxcl12$[16,31,32]), osteolineage cells (OLCs, marked by $Bglap$), chondrocytes (Chondro) and chondrocytes of possible transitional states (c/r, cycling/resting; pro, progenitor, marked by $Acan$ and $Col2a1$),

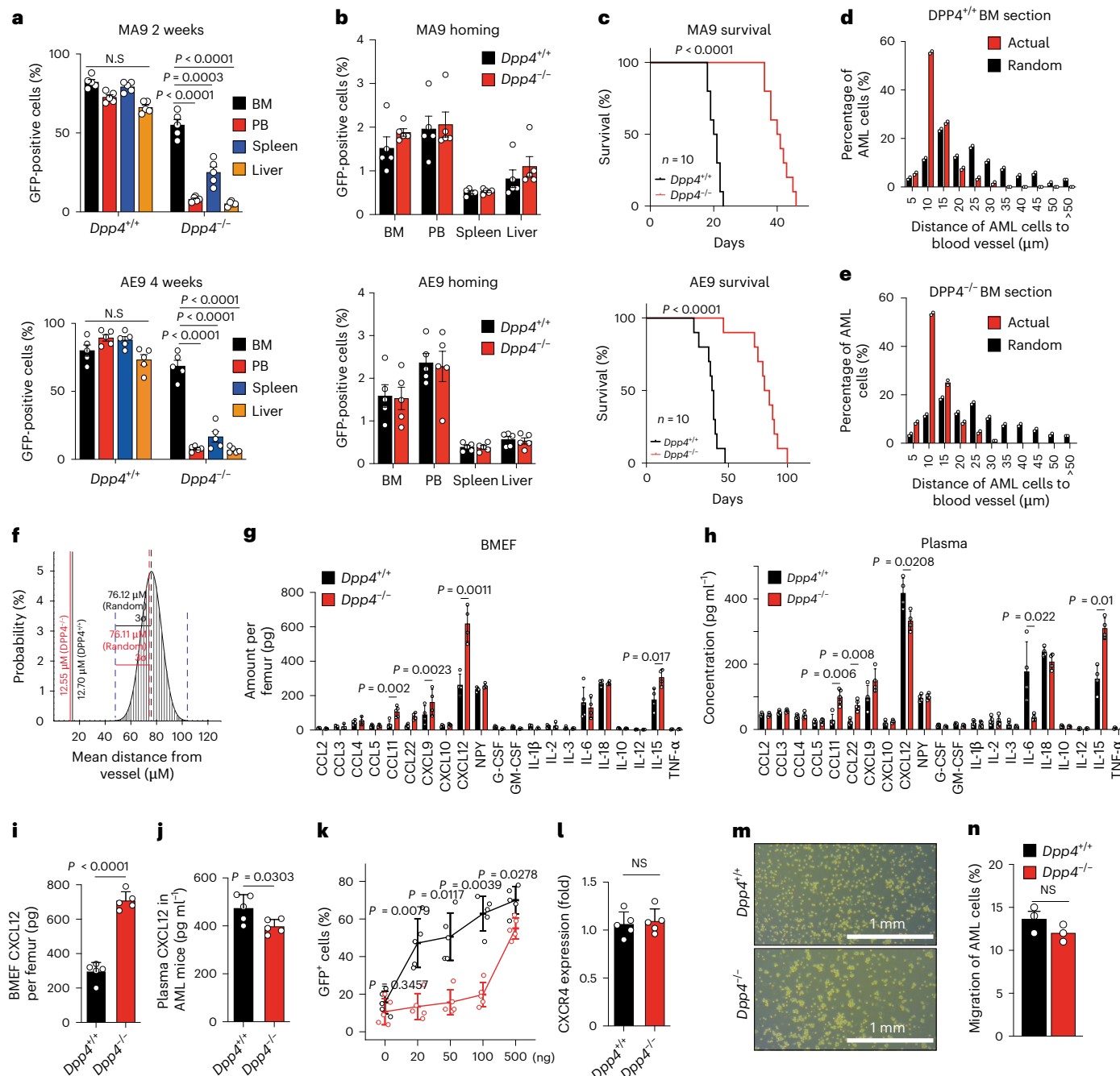

fibroblasts (Fibro, marked by *S1004a*), endothelial cells (ECs, marked by *Cdh5*), pericytes (marked by *Acta2*) (Fig. 4c and Extended Data Fig. 4). Consistent with prior reports, *Cxcl12* was predominantly expressed by the MSC cluster[1] (Fig. 4d). MSCs are heterogeneously characterized by several reported markers, such as LepR[+], N-cad[+] (*Cdh2*), Prx-1[+], Osx[+] (*Sp7*) and Nes[+] MSCs[16,32–37]. To identify the major source of CXCL12 in the PM/DM, we compared the association of *Cxcl12* with MSCs transcriptome and found *Cdh2* and *Lepr* had the highest association with *Cxcl12* expression among the aforementioned MSC subpopulations (Fig. 4e). LepR[+] MSCs are known as a heterogeneous population of MSCs spanning across the BM, including metaphysis (but mainly perivascular localization in metaphysis[32]) and CM region. The N-cad protein is predominantly detected in the endosteum of metaphysis[17]. Other MSC subpopulations, such as Prx-1[+], Osx[+] and Nes[+] MSCs, are minimally found within our study population (Fig. 4f), which is consistent with prior studies that Prx-1[+] MSCs enriched in periosteal region[38,39], Osx[+] enriched in the bone tissue[40] and Nes[+] enriched in perivascular region[37].

Critically, despite prior reports implicating Nestin[+] MSCs in AML chemoresistance[41], their *Cxcl12* expression was negligible (Fig. 4f). To rule out functional redundancy, we deleted *Cxcl12* in Nestin[+] cells (*Nestin-CreER; Cxcl12^{fl/fl}*), and observed no impact on AML progression or BM retention (Extended Data Fig. 5a,b), confirming that Nestin[+] MSCs are dispensable for CXCL12-mediated LSC maintenance in our model.

N-cad[+] MSCs co-expressed *Cxcl12* with other LSC-supportive factors[7,42–44] (for example, *Gas6*, *Angpt1* and *Kitl*; Fig. 4g). Given reports that SCF (*Kitl*) promotes AML adhesion and survival[45], we rigorously tested its role. Conditional deletion of *Scf* in N-cad[+] cells (*N-cad-CreER; Scf^{fl/fl}*) yielded no difference in mice survival, AML cell engraftment and distribution in the BM (Extended Data Fig. 6a–e). Consistently, SCF levels were unchanged across distinct marrow regions in *N-cad-CreER; Scf^{fl/fl}* mice compared with controls (Extended Data Fig. 6f). This negative result underscores that the observed niche effects are CXCL12-specific and not confounded by SCF.

**Fig. 3 | *Dpp4* deficiency confines AML cells within the BM through reversed BM-to-PB CXCL12 gradient. a**, Engraftment efficiency measured by comparison of the proportions of MA9 and AE9 GFP⁺ *Dpp4*⁺/⁺ or *Dpp4*⁻/⁻ AML cells in BM, PB, spleen and liver 2 weeks (MA9) and 4 weeks (AE9) after transplantation (*n* = 5 biologically independent mice per group). **b**, Homing efficiency measured by comparison of the proportions of MA9 and AE9 GFP⁺ *Dpp4*⁺/⁺ or *Dpp4*⁻/⁻ AML cells in BM, PB, spleen and liver 16 h after transplantation (*n* = 5 biologically independent mice per group). **c**, Survival curve of secondary transplanted mice receiving 1 million MA9 or AE9 GFP⁺ *Dpp4*⁺/⁺ or *Dpp4*⁻/⁻ AML cells (*n* = 10 mice; *P* < 0.001, log-rank test). **d,e**, Distance between *Dpp4*⁺/⁺ (**d**) or *Dpp4*⁻/⁻ (**e**) AML cells and blood vessels. The numbers on the *x* axis indicate intervals of 5 μm (5 indicates the interval 0–5; 10 indicates 5–10, and so on). *P* value (*Dpp4*⁺/⁺) = 3.1 × 10⁻¹⁶, *P* value (*Dpp4*⁻/⁻) = 3.7 × 10⁻¹⁶ by the two-sample Kolmogorov–Smirnov (KS) test. *n* = 2 biologically independent mice per group. For each mouse, three fields of view were quantified and averaged to obtain a single value per mouse. **f**, The observed mean distances of *Dpp4*⁺/⁺ or *Dpp4*⁻/⁻ MA9 AML cells to the nearest BM vessel are similar to each other. Probability distribution of the mean distances between GFP⁺ AML cells and vessels derived from simulations of randomly positioned GFP⁺ AML cells and actual vessels on maps of BM. Mean distances observed in situ for both *Dpp4*⁺/⁺ AML cells (black line) and *Dpp4*⁻/⁻ AML cells (red line) are shown in relation to the grand mean (mean of the means) ± 3 s.d. (3*σ*), (dotted black and dotted red lines, respectively). The observed mean distances of both *Dpp4*⁺/⁺ AML cells (12.70 mM) or *Dpp4*⁻/⁻ AML cells (12.55 mM) were statistically different from the mean distance of randomly placed *Dpp4*⁺/⁺ AML cells (76.12 mM) or randomly placed *Dpp4*⁻/⁻ AML cells (76.11 mM) to vessels. *n* = 500 AML cells (*Dpp4*⁺/⁺ and *Dpp4*⁻/⁻), respectively, from three mice in three independent experiments. **g,h**, Cytokine protein levels were measured in BMEF (**g**) and plasma (**h**) of *Dpp4*⁺/⁺ and *Dpp4*⁻/⁻ AML cells transplanted mice using the LEGEND plex Muti-Analyte Flow Assay Kit (*n* = 3 biologically independent mice per group). **i,j**, Quantitation of CXCL12 in BMEF (**i**) or plasma (**j**) of *Dpp4*⁺/⁺ and *Dpp4*⁻/⁻ AML cells transplanted mice by ELISA (*n* = 5 biologically independent mice per group). **k**, Mobilization of circulating GFP⁺ AML cells in PB was measured 17 h after injection by flow cytometry. Dpp4 KO (black) and WT (red) AML mice were intravenously injected with increasing doses of recombinant CXCL12 (0–500 ng) (*n* = 5 biologically independent mice per group). **l**, CXCR4 expression on GFP⁺ Mac-1⁺ Kit⁺ *Dpp4*⁺/⁺ and *Dpp4*⁻/⁻ AML-SCs as determined by flow cytometry (*n* = 5 biologically independent mice per group). **m**, Representative images of the lower wells of the migration assays. For the in vitro migration assay, the upper wells contained *Dpp4*⁺/⁺ or *Dpp4*⁻/⁻ AML cells; all lower wells contained 100 ng ml⁻¹ CXCL12; and the concentration of CXCL12 in the upper wells is 0 ng ml⁻¹. **n**, Quantification of AML cell migration after 4 h of incubation (performed in **m**) at 37 °C and 5% CO₂ (*n* = 3 biologically independent wells per group). Data are presented as mean ± s.e.m. *n* numbers are provided in the corresponding graphs and legends. Statistical significance was determined using a two-sided unpaired Student's *t*-test. Survival differences were assessed using a two-sided log-rank test. Exact *P* values are provided in the graphs. NS, not significant.

To spatially assess the relationship between LSCs and N-cad⁺ cells, we transplanted control L-GMPs into N-cad-tdTomato (N-cad-TdT) reporter mice, where Tomato⁺ cells mark *N-cadherin* expression. AML cells (GFP⁺) and *N-cad*⁺ stromal cells were co-enriched in the PM (Fig. 4h). Strikingly, LSCs (GFP⁺Kit⁺) were significantly more likely to reside within 5 μm of *N-cad*⁺ cells (45.3% ± 5.4%) compared with differentiated AML cells (GFP⁺Kit⁻; 10.6% ± 2.2%) (Fig. 4h, bottom right), suggesting active niche–LSC crosstalk.

To test whether N-cad⁺-derived CXCL12 drives LSC retention, we conditionally deleted *Cxcl12* in N-cad⁺ cells (*N-cad-CreER; Cxcl12*ᶠˡ/ᶠˡ) (Fig. 4i) and transplanted control L-GMPs (Fig. 4j). N-cad; *Cxcl12*⁻/⁻ mice recapitulated the *Dpp4* KO phenotype, with AML cells redistributed from PM/DM to CM (Fig. 4k,l).

Together, these results demonstrate that CXCL12 produced by N-cadherin⁺ MSCs is the key niche signal that anchors LSCs in the metaphysis and dictates their spatial distribution within the BM.

## Shared LSC behaviour and transcriptomic signatures in N-cad; *Cxcl12*⁻/⁻ and *Dpp4*-KO AML mice

The redistribution of LSCs from PM/DM to CM in N-cad; *Cxcl12*⁻/⁻ mice mirrored the phenotype observed in *Dpp4*-KO AML, prompting us to investigate whether these models shared LSC properties and underlying molecular mechanisms. CM-localized AML cells from *N-cad; Cxcl12*⁻/⁻ mice exhibited hallmark features of LSC exhaustion, including increased cell cycle activity and apoptosis, alongside impaired self-renewal capacity (Fig. 5a–d). These phenotypic changes were accompanied by delayed AML progression, as indicated by prolonged survival and reduced organ infiltration (Fig. 5e,f).

To determine whether the shared spatial redistribution phenotype reflected convergent molecular reprogramming, we performed transcriptomic analyses. Principal component analysis (PCA) revealed striking similarity between *Dpp4*-KO AML cells and control AML cells from *N-cad; Cxcl12*⁻/⁻ recipients, with both populations clustering distinctly from their respective controls (Fig. 5g). Heatmap inspection showed a similar pattern in both settings: cell cycle and metabolic genes were increased, whereas gene sets linked to stemness, exhaustion and migration were reduced (Fig. 5h,i and Extended Data Fig. 7a,b). To systematically evaluate pathway enrichment, we performed gene set enrichment analysis (GSEA) across curated hallmark and pathway gene sets. GSEA showed negative enrichment of IL6–JAK–STAT3, MAPK and NF-κB signalling, and positive enrichment of metabolic processes, in both comparisons (*Dpp4*⁻/⁻ versus *Dpp4*⁺/⁺ and *N-cad; Cxcl12*⁻/⁻ versus *N-cad; Cxcl12*⁺/⁺) (Fig. 5j). These data point to a convergent attenuation of cytokine-responsive JAK/STAT, MAPK and NF-κB signalling when DPP4 activity or niche-derived CXCL12 is disrupted. To confirm these pathway changes at the protein level, we performed immunoblotting

**Fig. 4 | Niche-specific CXCL12 production by N-cad⁺ MSCs dictates LSC spatial distribution. a,b**, Cartoon of BM structure anatomical partition (**a**) and summary (**b**) of CXCL12 levels measured in each BM area (PM, CM and DM) from *Dpp4*⁻/⁻ and *Dpp4*⁺/⁺ AML mice (*n* = 5 biologically independent mice per group). **c**, scRNA-seq of 11,512 non-haematopoietic cells from PM and DM regions 3 weeks after AML transplantation revealed 14 distinct clusters, including MSCs, OLCs, chondrocytes (and transitional states), fibroblasts, ECs and pericytes. **d**, Violin plot showing *Cxcl12* expression among identified cell clusters. **e**, Correlation analysis of *Cxcl12* expression level and MSC subtypes marked by *Lepr*, *Cdh2* (N-cadherin), *Sp7* (Orx), *Prx* and *Nes* (Nestin). Cor., Pearson correlation coefficient. **f**, UMAP feature plots showing expression of *Cxcl12* across MSC subtypes marked by *Lepr*, *Cdh2*, *Sp7* (Orx), *Prx* and *Nes* (Nestin). **g**, UMAP plots showing the spatial expression of Cdh2 (green) and selected niche factors (Cxcl12, Angpt1, Kitl, Gas6, Tgfb1 and Vcam1; red), with merged signals in yellow. Overlapping expression patterns are observed within the MSC compartment. **h**, Representative images of BM sections of *Dpp4*⁺/⁺ AML cell localization in the BM of N-cad-TdT mice at day 15 after transplantation. Green, GFP⁺ AML cells; red, N-cad⁺ MSCs; white, Kit; blue, DAPI for nuclei. The dashed box indicates the area of focus. Scale bars, 100 μm. Bottom right: relative distance between GFP⁺Kit⁺ AML cells and GFP⁺Kit⁻ AML cells to N-cad⁺ cells (*n* = 100 GFP⁺Kit⁺ AML cells, *n* = 80 GFP⁺Kit⁻ AML cells; *n* = 3 biologically independent mice per group). **i**, CXCL12 levels were measured in each BM area (PM, CM and DM) from N-cad; *Cxcl12*⁺/⁺ or N-cad; *Cxcl12*⁻/⁻ mice (*n* = 3 biologically independent mice per group). **j**, Schematic representation of *Cxcl12* conditional KO in the N-cad⁺ cell mouse model. In total, 500 primary L-GMPs were transplanted into sublethally irradiated recipients 14 days after tamoxifen injection. **k,l**, Immunostaining and statistical analysis of L-GMP markers in the BM section of N-cad; *Cxcl12*⁺/⁺ (top) and N-cad; *Cxcl12*⁻/⁻ (bottom) AML mice at day 14 and day 31, respectively, after transplantation. The calculated proportion of L-GMPs/GFP⁺ cells in each of the three anatomical BM areas is listed to the left of the images. The dashed box indicates the area of focus. Arrowheads point to the L-GMPs. Scale bars, 100 μm (*n* = 5 biologically independent mice per group), where panel **l** shows the quantification of panel **k**. Data are presented as mean ± s.e.m. *n* numbers are provided in the corresponding panels and legends. Statistical significance was determined using a two-sided unpaired Student's *t*-test; exact *P* values are provided in the graphs. Icons in **j** created in BioRender; Kang, X. https://biorender.com/34u8lmg (2026).

of AML cells, which showed reduced phosphorylation of STAT3, ERK1/2, p38 and NF-κB p65 (Fig. 5k and Extended Data Fig. 8a–e), aligning with the transcriptional suppression of the corresponding pathways. Together, these findings demonstrate that disruption of the CXCL12–DPP4 axis induces convergent transcriptomic rewiring that drives LSC exhaustion and impairs disease progression.

### N-cad+ cells support LSCs through GPC3-mediated attraction

While we have shown that DPP4, by deactivating CXCL12, specifically creates an intra-BM CXCL12 gradient that favours LSC localization to the PM/DM for maintenance, two key questions remain unanswered:

(1) Why does DPP4 selectively deactivate CXCL12 in the CM while preserving CXCL12 in the PM and DM in control AML mice (Fig. 4b)? (2) How do control LSCs achieve close proximity to N-cad+ niche cells (Fig. 4h)?

Prior studies have identified CXCL12 hotspots as critical HSC niches that attract HSCs to close proximity for their maintenance[46]. Consistent with this, our scRNA-seq analysis revealed that N-cad+ cells express high levels of *Cxcl12* (Fig. 4e,f). To further investigate, we compared the spatial relationship between LSC-enriched populations (GFP+Kit+) and N-cad+ cells in control and *Dpp4*-KO AML mice. Interestingly, *Dpp4*-KO LSCs resided significantly farther from N-cad+ cells than control LSCs, with only 10.5% ± 2.1% of *Dpp4*-KO LSCs located

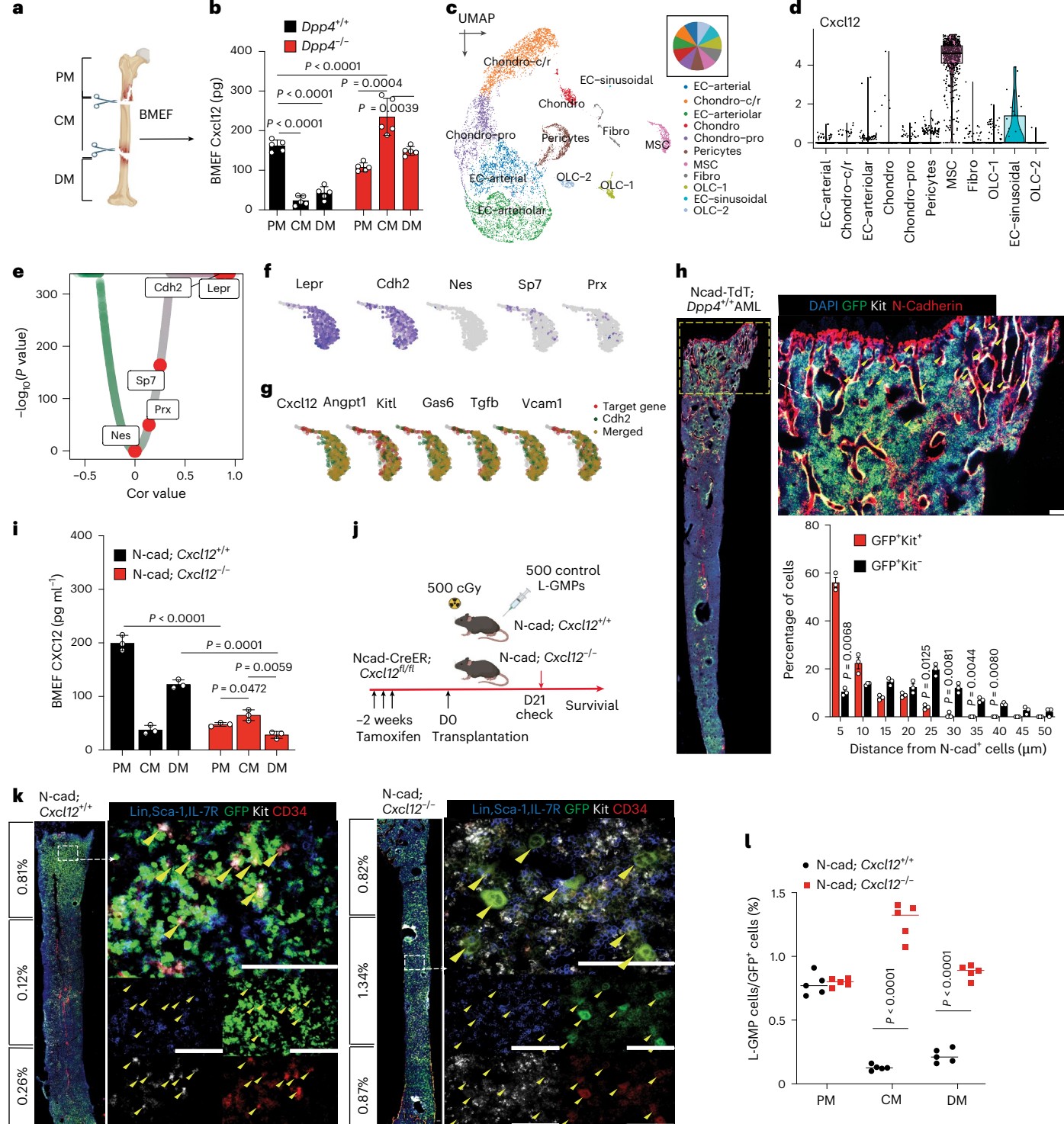

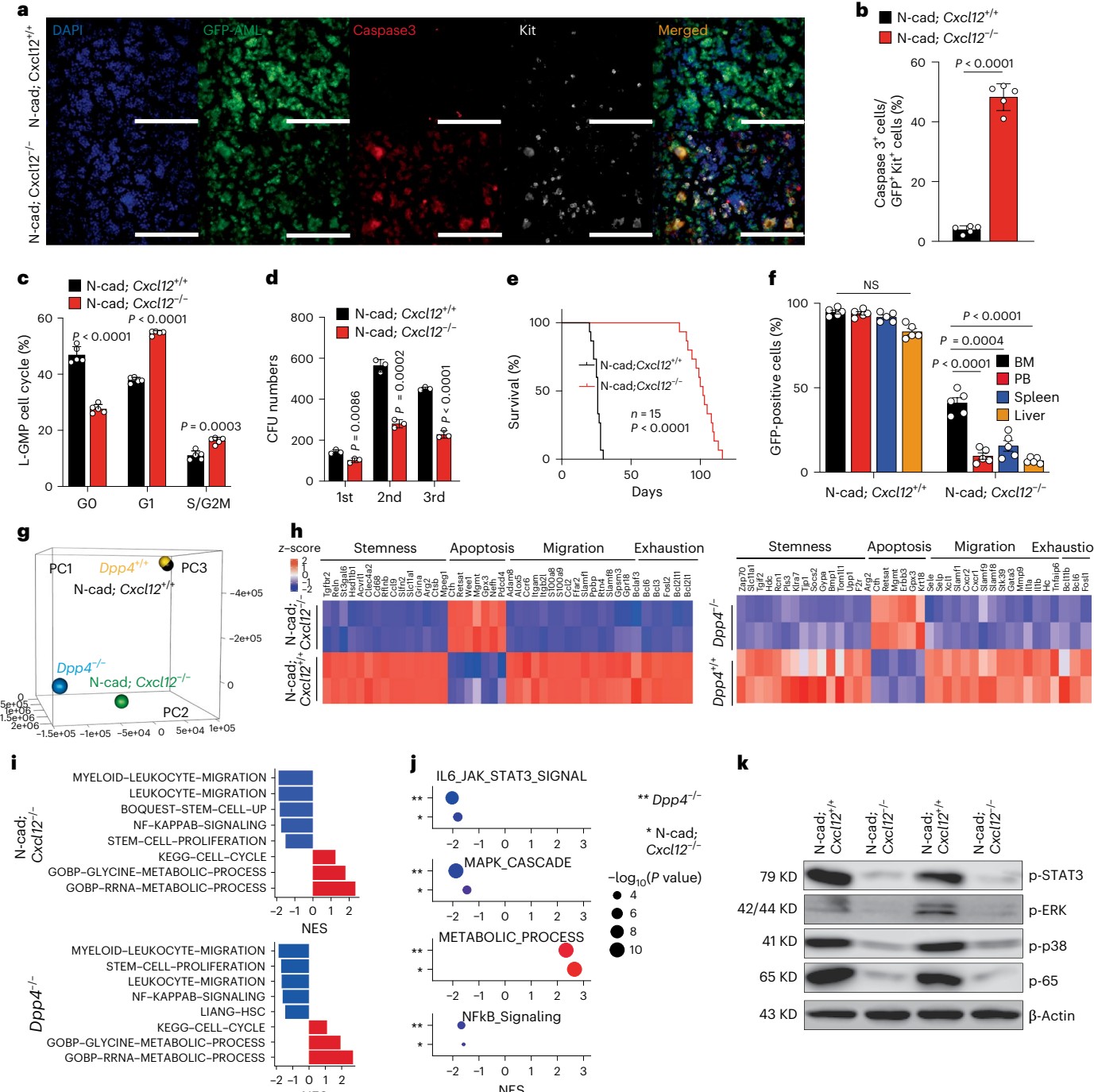

**Fig. 5 | Shared LSC behaviour and transcriptomic signatures in N-cad;** ***Cxcl12***$^{-/-}$ **and** ***Dpp4***-**KO AML mice. a**, Immunofluorescent image showing the levels of cleaved caspase-3 in the BM of N-cad; *Cxcl12*$^{+/+}$ and N-cad; *Cxcl12*$^{-/-}$ AML mice at day 14 and day 31, respectively, after transplantation. Green, GFP$^+$ AML cells; red, cleaved-caspase3, bule, DAPI; white, Kit$^+$ cells; scale bars, 100 μm. **b**,**c**, Apoptosis (**b**) and cell cycle (**c**) analysis of MA9 AML L-GMPs in N-cad; *Cxcl12*$^{+/+}$ and N-cad; *Cxcl12*$^{-/-}$ AML mice (*n* = 5 biologically independent mice per group) by flow cytometry. **d**, Comparison of CFU capability of N-cad; *Cxcl12*$^{+/+}$ and N-cad; *Cxcl12*$^{-/-}$ GFP$^+$/Gr1$^+$ AML cells during serial replating (2,000 cells per well, *n* = 3 biologically independent wells per group). **e**, Survival curve of 500 control L-GMPs transplanted recipient N-cad; *Cxcl12*$^{+/+}$ or N-cad; *Cxcl12*$^{-/-}$ mice (*n* = 15 mice; *P* < 0.0001, log-rank test). **f**, Comparison of the proportions of GFP$^+$ N-cad; *Cxcl12*$^{+/+}$ or N-cad; *Cxcl12*$^{-/-}$ AML cells in BM, PB, spleen and liver 6 weeks after transplantation (*n* = 5 biologically independent mice per group). **g**, PCA analysis for AML cells from N-cad; *Cxcl12*$^{-/-}$ and *Dpp4*$^{-/-}$ AML mice and their pairing N-cad; *Cxcl12*$^{+/+}$ and *Dpp4*$^{+/+}$ AML mice. **h**, Heatmap showing differentially expressed

genes in AML cells from N-cad; *Cxcl12*$^{+/+}$ and N-cad; *Cxcl12*$^{-/-}$ mice, as well as from *Dpp4*$^{+/+}$ and *Dpp4*$^{-/-}$ mice. **i**, Bar plot of GSEA results illustrating biological processes associated with AML cells of N-cad; *Cxcl12*$^{-/-}$ mice and *Dpp4*$^{-/-}$ mice shows the significant alteration of migration, stemness, proliferation and the cell cycle. **j**, Highly consistent GSEA change between *Dpp4*$^{-/-}$ versus *Dpp4*$^{+/+}$ (**) and N-cad; *Cxcl12*$^{-/-}$ versus N-cad; *Cxcl12*$^{+/+}$ (*), including downregulation of STAT3, MAPK and NF-kB pathways and increased metabolic processes. NES, normalized enrichment score; FDR, false discovery rate. **k**, Comparison of p-STAT3, p-ERK, p-P38 and p-p65 protein levels in N-cad; *Cxcl12*$^{+/+}$ and N-cad; *Cxcl12*$^{-/-}$ MLL-AF9 BM cells from each of two AML mice. KD, kilodaltons (kDa). Data are presented as mean ± s.e.m. *n* numbers are provided in the corresponding panels and legends. Statistical significance was determined using a two-sided unpaired Student's *t*-test. Survival differences were assessed using a two-sided log-rank test. Exact *P* values are provided in the graphs. The immunoblotting experiment was independently repeated three times with similar results.

within 5 μm of N-cad⁺ cells, compared with 58.6% ± 7.3% of control LSCs (Figs. 4h and 6a,b). Moreover, in the PM/DM, *Dpp4*-KO L-GMPs exhibited exhaustion-like behaviour (Fig. 6c), underscoring the importance of close proximity to N-cad⁺ cells for LSC maintenance. As such, we hypothesized that an N-cad⁺ cell-derived factor interacts with DPP4 to establish the CXCL12 gradient at both the macroscopic (PM/DM–CM) and the microscale level. To test this, we conducted transcriptional profiling of ten known CXCL12 or DPP4 regulators (for example, DPP8, elastase, MMPs, cathepsin G, TFPI and GPC3)[14,47,48] in both AML cells and N-cad⁺ cells from the PM/DM. Notably, GPC3 was highly expressed in N-cad⁺ cells compared with other factors (Fig. 6d,e). GPC3 is known as an inhibitor of DPP4, suggesting its potential role in suppressing DPP4 activity and preserving CXCL12 near N-cad⁺ cells. Indeed, in vitro experiments demonstrated that GPC3 binds to DPP4 on AML cells and inhibits DPP4 enzymatic activity (Fig. 6f,g). Immunostaining further revealed significantly higher GPC3 expression in N-cad⁺ cells (79.8% ± 3.3%) compared with N-cad⁻ cells (10.5% ± 2.1%) in the PM region (Fig. 6h–i), with 73.5% of GPC3 and DPP4 colocalized between N-cad⁺ cells and LSCs (Fig. 6j). These data indicate that GPC3, expressed by N-cad⁺ cells, inhibits DPP4 and sustains CXCL12 at the microscale, thereby attracting LSCs to CXCL12 hotspots for their maintenance. In addition, at the macroscopic level, GPC3-mediated DPP4 inhibition contributes to the intra-BM CXCL12 gradient between the PM/DM and CM, further facilitating LSC localization to supportive niche environments.

### *Gpc3* KO in N-cad⁺ redistributes AML cells and impairs LSCs

To functionally assess the role of the GPC3–DPP4 interaction in LSC maintenance, we conditionally deleted *Gpc3* from N-cad⁺ cells (N-cad; *Gpc3*⁻/⁻) and transplanted LSCs into both N-cad; *Gpc3*⁻/⁻ and control mice (N-cad; *Gpc3*⁺/⁺) (Fig. 7a). Loss of GPC3 in N-cad⁺ cells led to a significant reduction in CXCL12 levels in the PM and DM (Fig. 7b), resulting in an altered AML cell distribution pattern similar to that observed in *Dpp4*-KO and N-cad; *Cxcl12*⁻/⁻ mice (Fig. 7c). This redistribution was accompanied by LSC exhaustion, as indicated by increased L-GMP division and apoptosis (Fig. 7d,e). The disruption of the CXCL12 gradient and loss of proximity to supportive N-cad⁺ cells impaired LSC maintenance, ultimately prolonging mouse survival (Fig. 7f).

Thus, loss of GPC3 in N-cad⁺ MSCs disrupts CXCL12 maintenance, redistributes AML cells away from supportive niches and improves survival.

## Discussion

Our study identified two pivotal findings: the redistribution of LSCs, leading to their exhaustion, and the confinement of AML cells within the BM. Both phenomena are fundamentally mediated by CXCL12, a key chemoattractant for AML cells[9,10] (Extended Data Fig. 9). We show that it is not the absolute levels of CXCL12, but rather its

spatial compartmentalization and gradients within the BM niche, that determine LSC fate. While CXCL12's role in LSC biology has been controversial[2,49–51], our data reconcile these discrepancies by linking LSC exhaustion to altered CXCL12 gradients rather than chemokine abundance per se. In addition, transcriptional profiling indicates that Dpp4 loss is accompanied by coordinated metabolic changes, which may further contribute to the intrinsic exhaustion phenotype of Dpp4-deficient LSCs.

A key advance of our study is the identification of N-cadherin⁺ MSCs as the functionally relevant CXCL12 source for LSC maintenance. This contrasts with previous work focusing on Prx⁺ periosteal[38,39] or Tek⁺ endothelial niches[49], which our scRNA-seq showed to be minimally involved in metaphyseal LSC maintenance (Fig. 4f). Other well-known MSC markers, such as Lepr, Nestin and Osx, are either widespread expressed throughout the BM[32], enriched in perivascular regions[37] or restricted to bone tissue[40]. Importantly, we confirmed through genetic deletion that neither Nestin⁺ MSC-derived CXCL12 (despite their reported role in chemoresistance) nor N-cad⁺ MSC-derived SCF contribute meaningfully to the phenotypes we observed (Extended Data Figs. 5 and 6) underscoring the specificity of the N-cadherin⁺ MSC-CXCL12 axis in our model.

The discovery of a reverse PM/DM–CM CXCL12 gradient in control AML mice provides a mechanistic basis for LSC niche specificity. This gradient depends on regional regulation of DPP4 activity: while DPP4 degrades CXCL12 in the CM, GPC3 from N-cad⁺ MSCs inhibits DPP4 in the PM/DM, creating protected CXCL12 microdomains. This explains why earlier high-resolution mapping studies in normal hematopoiesis failed to detect long-range CXCL12 gradients[46]—pathological DPP4 activity in AML imposes spatial chemokine asymmetry not obvious under steady-state conditions[20]. Collectively, our work identifies the CXCL12–DPP4–GPC3 axis as a master regulator of LSC niche interactions.

Importantly, these findings complement rather than replace prior niche-targeting paradigms. Structural and vascular remodelling have been implicated in leukaemia progression. For example, Hawkins et al. showed that T cell acute lymphoblastic leukaemia (T-ALL) cells exhaust niches by physical displacement[52], and Duarte et al. reported that AML remodels vascular niches[6]. Other landmark studies have shown that targeting CXCR4–CXCL12 interactions, E-selectin-mediated adhesion or endosteal vascular remodelling can significantly alter leukaemia–niche dynamics[3,53–57]. Our work contributes a distinct regulatory layer by defining how protease-mediated chemokine inactivation, counterbalanced by stromal inhibition of DPP4, shapes LSC positioning and survival.

The MLL-AF9 and AML-ETO9a models used here recapitulate aggressive human AML subtypes (KMT2A-rearranged and t(8;21)), which are associated with high leukaemic burden and poor prognosis. In both models, genetic ablation of Dpp4 confined AML cells within the BM, depleted LSCs from protective metaphyseal niches and reduced

---

**Fig. 6 | N-cad⁺ cells support LSCs through GPC3-mediated attraction.**
**a**, Representative images of BM sections of *Dpp4*⁻/⁻ AML cell localization in the BM of N-cad-TdT mice at day 15 after transplantation. Green, GFP⁺ AML cells; red, N-cad⁺ MSCs; white, Kit; blue, DAPI for nuclei. The dashed box indicates the area of focus. Scale bars, 100 μm. **b**, Distance between *Dpp4*⁺/⁺ (Fig. 4h) or *Dpp4*⁻/⁻ (Fig. 5a) LSC-enriched populations (GFP⁺Kit⁺) and N-cad⁺ cells. The numbers on the x axis indicate intervals of 5 μm (5 indicates the interval 0–5; 10 indicates 5–10, and so on; in total, 200 AML cells for each group were calculated). n = 3 biologically independent mice per group. **c**, Cell cycle analysis of MA9 L-GMPs in PM, CM and DM areas from *Dpp4*⁻/⁻ mice (n = 5 biologically independent mice per group) by flow cytometry. **d**, Bar graphs compare the expression levels of the indicated transcripts in N-cad⁺ MSCs and MA9 AML cells. All transcripts' levels were normalized to levels of actin expression (n = 3 biologically independent wells). **e**, Contour plots show that N-cad⁺ BM cells have significantly higher cell surface expression of GPC3 than N-cad⁻ BM cells. GPC3 expression level (median fluorescence intensity, MFI) in N-cad⁺ and N-cad⁻ populations have

been indicated. n = 5 biologically independent mice per group. **f**, Flow cytometry analysis of recombinant GPC3 (100 nM) binding to DPP4⁺ BM AML cells, MFIs are indicated. **g**, Comparison of the DPP4 enzyme activity of 1 × 10⁶ mouse AML cells measured by fluorescence assay with the indicated treatment and timepoints. Results are reported as RLUs (n = 3 biologically independent wells). **h**, Representative BM section images of GFP-labelled *Dpp4*⁺/⁺ AML cells in the N-cad-TdT AML mouse trabecular bone region at day 15 after transplantation. The dashed box indicates the area of focus. The dashed line indicates the bone structure; scale bar, 20 μM. Selected three-dimensional image of DPP4 and GPC3 interaction. Green, GFP⁺ AML cells; red, N-cad⁺ s; blue, DPP4; yellow, GPC3. **i**, Statistical analysis of expression and localization of GPC3. n = 3 biologically independent mice per group. **j**, Percentage of DPP4⁺ AML cells overlap with GPC3⁺ N-cad⁺ MSCs. n = 4 biologically independent mice. Data are presented as mean ± s.e.m. n numbers are provided in the corresponding panels and legends. Statistical significance was determined using a two-sided unpaired Student's t-test. Exact P values are provided in the graphs.

peripheral dissemination—a clinically relevant effect given that leukocytosis and organ infiltration are major drivers of AML morbidity[58,59]. Thus, targeting the DPP4–CXCL12–GPC3 axis may simultaneously impair LSC maintenance and reduce systemic disease burden.

This axis offers immediate translational promise. DPP4 inhibitors (for example, sitagliptin and linagliptin), already approved by the US Food and Drug Administration for diabetes[13,60], could be repurposed to: (1) reverse BM–PB CXCL12 gradients and confine AML cells,

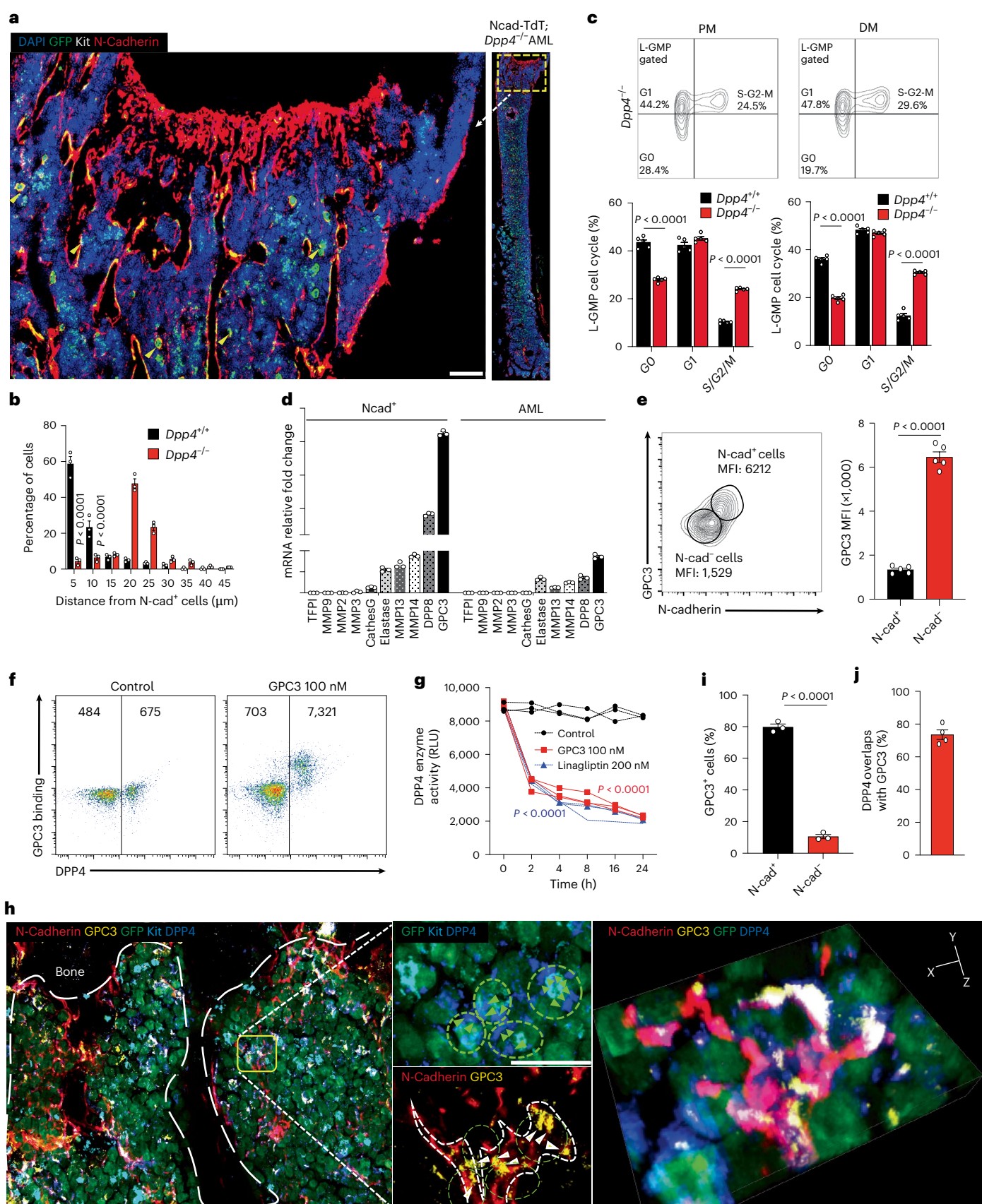

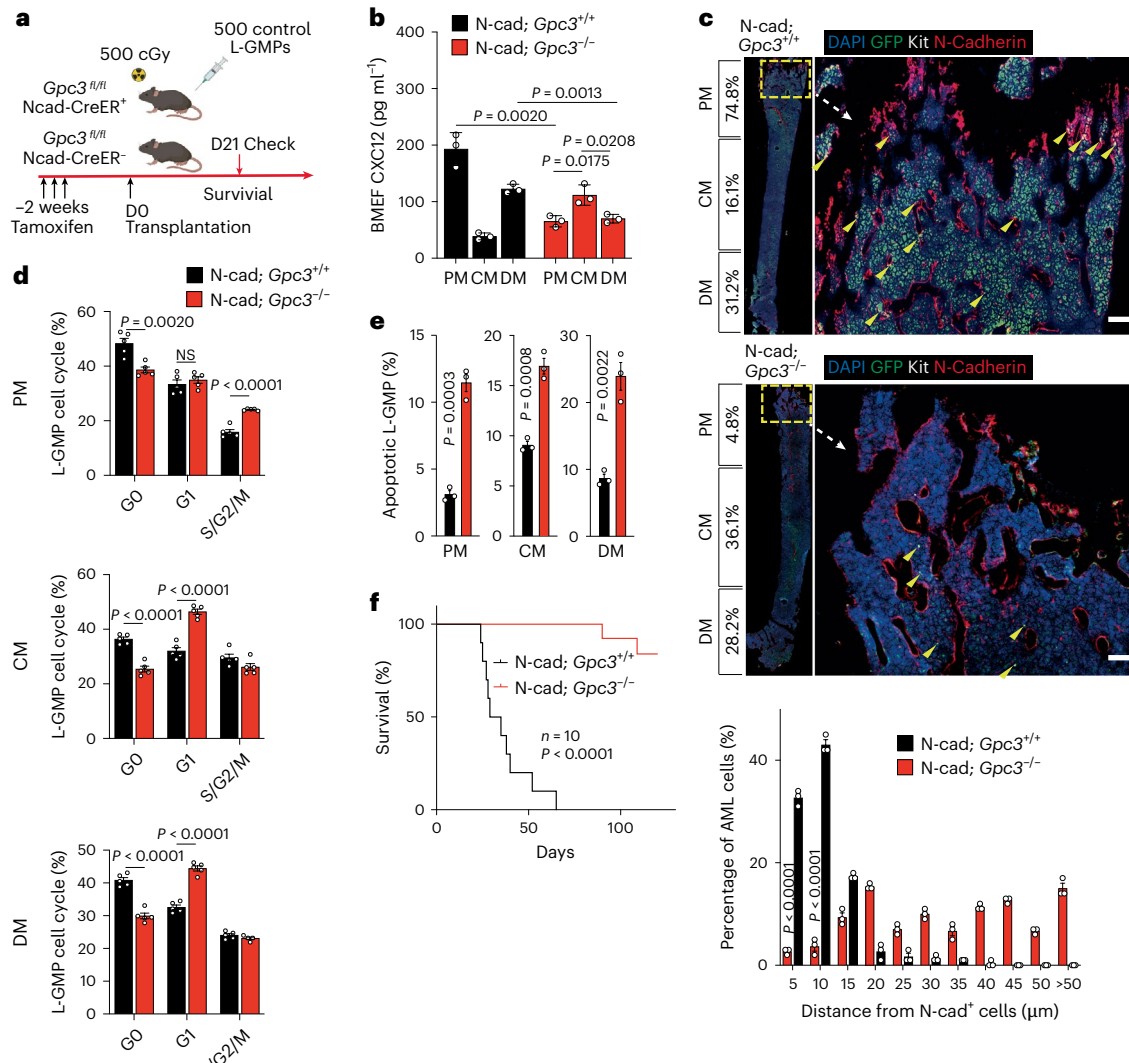

**Fig. 7 | *Gpc3* KO in N-cad⁺ redistributes AML cells and impairs LSCs.**
**a**, Schematic representation of *Gpc3* conditional KO in the N-cad⁺ cell mouse model. In total, 500 primary L-GMPs were transplanted into sublethally irradiated recipients 14 days after tamoxifen injection. **b**, Cxcl12 levels were measured in each BM area (PM, CM and DM) from N-cad; *Gpc3*⁺/⁺ or N-cad; *Gpc3*⁻/⁻ mice (n = 5 biologically independent mice per group). **c**, Representative images of BM sections of N-cad; *Gpc3*⁺/⁺ (top) and N-cad; *Gpc3*⁻/⁻ (middle) AML cell localization in the BM of N-cad-TdT mice at day 15 and day 28, respectively, after transplantation. Green, GFP⁺ AML cells; red, N-cad⁺ MSCs; white, Kit⁺; blue, DAPI for nuclei. The dashed box indicates the area of focus. Scale bars, 100 µm. Bottom: distance between AML cells and N-cad⁺ cells. The numbers on the *x* axis indicate intervals of 5 µm (5 indicates the interval 0–5; 10 indicates 5–10, and so

on; n = 3 biologically independent mice per group). **d**, Cell cycle analysis of MA9 L-GMPs in PM, CM and DM areas in N-cad; *Gpc3*⁺/⁺ or N-cad; *Gpc3*⁻/⁻ mice by flow cytometry (n = 5 biologically independent mice per group). **e**, Apoptosis analysis of MA9 L-GMPs in PM, CM and DM areas in N-cad; *Gpc3*⁺/⁺ or N-cad; *Gpc3*⁻/⁻ mice (n = 3 biologically independent mice per group) by flow cytometry. **f**, Survival curve of control L-GMPs transplanted recipient N-cad; *Gpc3*⁺/⁺ or N-cad; *Gpc3*⁻/⁻ mice (n = 10 mice; P < 0.0001, log-rank test). Data are presented as mean ± s.e.m. *n* numbers are provided in the corresponding panels and legends. Statistical significance was determined using a two-sided unpaired Student's *t*-test. Survival differences were assessed using a two-sided log-rank test. Exact *P* values are provided in the graphs. Icons in **a** created in BioRender; Kang, X. https://biorender.com/34u8lmg (2026).

reducing complications of hyperleukocytosis[61,62]; (2) dislodge LSCs from protective niches to sensitize them to chemotherapy; and (3) induce LSC exhaustion. Building on these mechanistic insights, we have initiated preclinical testing of combination strategies, including CXCR4 antagonists (for example, AMD3100)[63], NF-κB inhibitors (for example, QNZ)[64] and N-cadherin antagonist (ADH-1)[65], to disrupt stromal anchoring, survival pathways and chemokine gradients in parallel. At the same time, future work will prioritize functional validation of the survival and metabolic gene modules identified in our genomic analyses to clarify which pathways most directly govern LSC fitness. Such efforts are aligned with the expanding field of BM niche-targeting therapies in haematopoietic malignancies[3,52–54,66], exemplified by the E-selectin antagonist uproleselan, which remodels stromal interactions and enhances haematopoietic recovery[57]. These collective advances

reinforce that the BM microenvironment is a drug-responsive component of haematologic malignancies.

Important limitations remain. We did not evaluate whether additional DPP4 substrates contribute to LSC regulation, and while murine models capture many aspects of human AML, validation in patient-derived xenografts will be essential to address interpatient heterogeneity. Human AML exhibits substantial genetic, epigenetic and functional diversity, with leukaemia-initiating cell activity distributed across both CD34⁺ and CD34⁻ fractions, depending on subtype[67,68]. Moreover, quiescence and stemness do not invariably overlap; some LSCs are deeply dormant and therapy-resistant, whereas others retain self-renewal despite active cycling[7,69]. This diversity underscores the need to determine whether the DPP4–CXCL12–GPC3 axis governs these distinct leukaemia-initiating cell states in a similar manner.

Nevertheless, our findings provide a mechanistic and translational framework that integrates protease activity, stromal signalling and chemokine gradients as interdependent regulators of leukaemic stemness. Collectively, the DPP4–CXCL12–GPC3 axis emerges as a clinically actionable stromal checkpoint, establishing rationale for niche-targeted therapies in AML (Extended Data Fig. 9).

## Online content

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

## Methods

### Mice

All animal experiments were performed in accordance with institutional guidelines and were approved by the Institutional Animal Care and Use Committee (IACUC) of the University of Missouri (protocol number 65384). C57BL/6NCrl mice (*Mus musculus*) were obtained from Charles River Laboratories. *Dpp4*[flox/flox] mice were generated by breeding targeted C57Bl/6NTac-DPP4tm1a Wtsi/Ics mice (European Mouse Mutant Cell Repository, EUCOMM) with 129S4/Bl6-Gt(ROSA)26Sortm2(FLP*)Sor/J (stock no. 012930, The Jackson Laboratory). The offspring were further crossed with Vav-iCre mice (stock no. 018968, The Jackson Laboratory)[70], to generate *Dpp4*[fl/fl] ; Vav-Cre mice[13]. N-cad-tdTomato (Cdh2-tdTamato), N-cad-CreER (Cdh2-CreEr) and *Gpc3*[fl/fl] strains were generated by L.L. at Stowers Institute for Medical Research[16,18]. *Cxcl12*[fl/fl] (stock no. 022457), *Scf*[fl/fl] (stock no. 017861) and Nestin-CreER (stock no. 016261) mice were purchased from The Jackson Laboratory. To induce expression of Cre-ER recombinase, mice received tamoxifen via intraperitoneal injection (Sigma, 75 mg tamoxifen per kilogram body weight) as described[71]. All mouse strains used in this study had a C57BL/6 genetic background. Both male and female mice aged 6–10 weeks were used unless otherwise specified. Animals were randomly assigned to experimental groups based on genotyping results. Investigators were blinded to group allocation during data analysis but not during experimental procedures. Sample sizes for each experiment are detailed in figure legends. Mice were housed in a specific-pathogen-free facility under a 12-h light/12-h dark cycle at an ambient temperature of 20–24 °C and relative humidity of 40–60%, with ad libitum access to food and water.

### AML transplantation

For Fig. 1a, we transplanted infected *Dpp4*-KO or control Lin⁻ cells ($2 \times 10^5$; infection efficiency consistently 40–50%) into lethally irradiated (1,000 cGy) C57/B6 mice (6–8 weeks old) and evaluated 2–50 weeks after transplantation. For other *Dpp4* KO versus control AML models, $1 \times 10^6$ primary AML cells were transplanted C57/B6 recipients. For the homing assay, GFP⁺ AML cells were evaluated at 16 h. Engraftment was assessed 2 and 4 weeks after transplantation for MLL-AF9 and AML-ETO9a models, respectively. For control AML cells transplantation into N-cad⁺ mice with or without *Cxcl12* or *Gpc3*, 500 L-GMPs were transplanted into sublethally irradiated (500 cGy) mice and engraftment assessed 2–8 weeks after transplantation. To preserve niche cell integrity, all mice were sublethally irradiated (500 cGy) before transplantation. The maximal leukaemia burden permitted under the approved IACUC protocol (#65384) was defined according to institutional humane endpoint criteria. Animals were monitored regularly and euthanized upon reaching predefined signs of morbidity or distress. The maximal leukaemia burden permitted by the ethics committee was not exceeded in any experiment.

### Flow cytometry

PB, BM, spleen and liver haematopoietic cells were labelled with the following antibodies (all from BioLegend unless otherwise noted): anti-CD3e-PE/Cyanine5 (clone 17A2, #100310, 1:200), anti-Ly6G/Ly6C (Gr-1)-PE/Cyanine5 (clone RB6-8C5, #108410, 1:200), anti-CD11b-PE/Cyanine5 (clone M1/70, #101210, 1:200), anti-CD45R-PE/Cyanine5 (clone RA3-6B2, #103210, 1:200), anti-Ter-119-PE/Cyanine5 (clone TER-119, #116210, 1:200), anti-CD117 (c-Kit)-APC (clone 2B8, #105812, 1:200), anti-Sca-1-PE-Cy7 (clone D7, #108114, 1:200), anti-CD150-PE (clone TC15-12F12.2, #115904, 1:200), anti-CD48-APC/Cyanine7 (clone HM48-1, #103432, 1:200), anti-Ki67-FITC (clone 16A8, #652410, 1:200), Hoechst 34580 (BD Pharmingen, #565877), anti-CD16/32-PE (clone 93, #101308, 1:200), anti-CD34-FITC (clone RAM34, eBioscience, #11-0341-82, 1:200), anti-CD127-APC/Cyanine7 (clone A7R34, #135040, 1:200), anti-CD135-Brilliant Violet 421 (clone A2F10, #135314, 1:200), anti-CD45-APC (clone I3/2.3, #147708, 1:200), anti-Ter-119-PE

(clone TER-119, #116208, 1:200), anti-CD31-PerCP/Cyanine5.5 (clone W18222B, #160206, 1:200), Annexin V (#640941, 1:20) and propidium iodide (#421301, 1:50). Intracellular staining was performed using the Foxp3/Transcription Factor Staining Kit (eBioscience) according to the manufacturer's protocol. Flow cytometry analyses were performed independently in triplicate, with biological replicates from at least five mice per condition. Technical replicates were included for measurement accuracy.

### Immunofluorescence staining and quantification

Femurs were perfused with phosphate-buffered saline (PBS), fixed with 4% paraformaldehyde and subjected to frozen sectioning. Antigen retrieval was performed with 1 µg ml⁻¹ proteinase K in TE buffer (100 mM Tris–HCl, pH 8.0, 50 mM EDTA) at 37 °C for 30 min. Sections were blocked with Universal Blocking Reagent (BioGenex) and incubated overnight at 4 °C with primary antibodies, including anti-Endomucin (goat polyclonal, R&D Systems, #AF4666, 1:100), Biotin anti-mouse Lineage Panel (clone 145-2C11; RB6-8C5; RA3-6B2; Ter-119; M1/70 BioLegend, #133307, 1:200), Biotin anti-mouse IL-7Rα (clone A7R34, BioLegend, #135006, 1:200), Biotin anti-mouse Sca-1 (clone D7, BioLegend, #108104, 1:200), phycoerythrin (PE) anti-mouse CD34 (clone HM34, BioLegend, #128610, 1:200), Alexa Fluor 647-anti mouse CD117(c-kit) (clone 2B8, BioLegend, #105818, 1:200) and anti-GPC3 (rabbit polyclonal, Abcam, #ab216606, 1:200). Secondary antibodies included donkey anti-goat Alexa Fluor 555 (Invitrogen; 1:500), goat anti-rabbit Alexa Fluor 750 (Invitrogen; 1:500) and Brilliant Violet 421-conjugated streptavidin (BioLegend; 1:500) at room temperature for 1 h. A 4′,6-diamidino-2-phenylindole (DAPI) stock solution was diluted to 300 nM in PBS, and 300 ml was added to the coverslip preparation for 1 min. Sections were rinsed three times in PBS, excess buffer drained from the coverslip and mounted with Shandon Immu-Mount (Fisher Scientific). Image stitching was done to capture the entire specimen at high magnification and seamlessly create a single high-resolution image. Sections were imaged using a Keyence BZ-X800 fluorescence microscope at 20× magnification (resulting in 200× magnification) and 60× magnification (resulting in 600× magnification). Quantification was performed using Keyence BZ-X800 analyser software, assessing GFP⁺ AML cell distribution across BM areas (PM, CM and DM), L-GMP localization, and apoptosis. Distances between GFP⁺ AML cells and N-cad⁺ cells were measured using a minimum of 100 GFP⁺Kit⁺ and 80 GFP⁺Kit⁻ AML cells per dataset. A minimum of five mice per condition were used for each quantification dataset. Detailed light microscopy acquisition parameters are provided in Supplementary Table 1.

### Cytokine analyses

Cytokine quantification in plasma and BMEF were determined using the LEGENDplex Multi-Analyte Flow Assay Kit (BioLegend), a bead-based immunoassay that quantifies multiple cytokines simultaneously via flow cytometry. In brief, a custom mouse cytokine and chemokine panel was used to measure the concentrations of the designed cytokines/chemokines. The LSRFortessa X-20 Cell Analyzer (BD Biosciences) was used for data acquisition, and results were analysed using the LEGENDplex Data Analysis software. Assays were performed in 96-well plates following manufacturer protocols, with data recorded using a Fisherbrand microplate photometer. BMEF and plasma were used for enzyme-linked immunosorbent assay (ELISA) collected using the mouse SDF-1 alpha ELISA Kit (Invitrogen) following the manufacturer's protocol.

### Migration assay

In vitro, the Transwell migration assay was utilized to assess cell migration. DPP4⁺/⁺ and DPP4⁻/⁻ AML cells were cultured and then seeded in serum-free medium into the upper chamber of Transwell inserts (Corning) with an 8-µm pore size, Matrigel-coated membrane.

The upper wells contained CXCL12 at concentrations of 0 ng ml$^{-1}$. The lower chambers were filled with culture medium containing 100 ng ml$^{-1}$ CXCL12. After a 4-h incubation at 37 °C and 5% CO$_2$, non-migratory cells on the upper membrane surface were removed using a cotton swab. Migratory cells on the lower membrane surface were visualized and quantified in multiple random fields under a microscope. Migration rates and statistical significance were analysed accordingly.

In vivo, mice ($n = 5$) were intravenously injected with PBS or CXCL12 (0–500 ng g$^{-1}$ per mouse as indicated in Fig. 3k). The percentage of GFP$^+$ AML cells in PB was measured before and 17 h after injection.

## Colony assays

Mouse AML cells were diluted to the indicated concentration in Iscove's modified Dulbecco's medium with 2% fetal bovine serum and were then seeded into methylcellulose medium M3534 (STEMCELL Technologies) for myeloid colony formation analysis[72]. For serial CFU assays, L-GMP cells were re-isolated by fluorescence-activated cell sorting (FACS) from collected colonies and replated into fresh methylcellulose medium for subsequent rounds of colony formation. Each CFU assay was performed in at least triplicate using independent biological replicates derived from distinct AML mice. Consistent colony-forming capacity was observed across all replicates.

## DPP4 activity assay

DPP4 activity was measured in plasma-EDTA, BMEF and cell lysates. For each assay, 20 µl of serum or BMEF, or 100 nM of GPC3 protein, was diluted in DPP4 assay buffer (Tris–HCl (pH 8.0), 150 mM NaCl and protease inhibitor cocktail) in a black 96-well plate to a final volume of 50 µl. An equal volume (50 µl) of 200 mM H-Ala-Pro-AFC substrate (I-1680; Bachem Americas) was added, and the plate was incubated for 10 min at room temperature in the dark. Fluorescence was measured using a Synergy Microplate Reader at excitation/emission wavelengths of 405/535 nm, and results were reported as relative light units (RLUs).

## Ligand binding assay

Recombinant His-tagged GPC3 binding to $Dpp4^{+/+}$ AML cells was assessed as similarly described[73]. In brief, $1 \times 10^6$ $Dpp4^{+/+}$ AML cells were incubated with or without His-GPC3 (100 nM) in 200 ml PBS/1% BSA for 3 h at 25 °C. Non-specific binding was subtracted. After incubation, cells were washed twice by centrifugation, resuspended in ice-cold PBS/1% BSA, and stained with Alexa Fluor 488 anti-His Tag Antibody for flow cytometry analysis.

## Quantitative RT–qPCR

Reverse-transcription quantitative PCR (RT–qPCR) was performed using 5 ng total RNA, gene-specific primers and a QIAGEN One Step RT-PCR kit (210210; Qiagen) following the manufacturer's instructions. 18S rRNA was used as an internal control for normalization. The primer sequences used are listed below: TFPI: forward(5′-GGG CTC CGT TCT TGG TCT C-3′) and reverse(5′-TTG AAT CTG CGG CAC TTT TGC-3′), MMP9: forward(5′-CTG GAC AGC CAG ACA CTA AAG-3′) and reverse (5′-CTC GCG GCA AGT CTT CAG AG-3′), MMP2: forward(5′-CAA GTT CCC CGG CGA TGT C-3′) and reverse (5′-TTC TGG TCA AGG TCA CCT GTC-3′), MMP3: forward(5′-ACA TGG AGA CTT TGT CCC TTT TG-3′) and reverse (5′-TTG GCT GAG TGG TAG AGT CCC-3′), MMP13: forward(5′-CTT CTT CTT GTT GAG CTG GAC TC-3′) and reverse (5′-CTG TGG AGG TCA CTG TAG ACT-3′), MMP14: forward(5′-CAG TAT GGC TAC CTA CCT CCA G-3′) and reverse (5′-GCC TTG CCT GTC ACT TGT AAA-3′), cathepsin G: forward(5′-AGG GTT TCT GGT GCG AGA AG-3′) and reverse (5′-GTT CTG CGG ATT GTA ATC AGG AT-3′), Elastase: forward(5′-AGC AGT CCA TTG TGT GAA CGG-3′) and reverse (5′-CAC AGC CTC CTC GGA TGA AG-3′), DPP8: forward (5′-GGG AAA TGG TGA ATC ACA GGA C-3′) and reverse (5′-ATG TAG CCG TGG TAT TTT CTG G-3′), GPC3: forward (5′-CAG CCC GGA CTC AAA TGG G-3′) and reverse (5′-CAG CCG TGC TGT TAG TTG GTA-3′).

## RNA-seq analysis

BM AML cells were FACS-sorted from two independent Dpp4$^{+/+}$ and Dpp4$^{-/-}$ leukaemia-bearing mice, as well as from N-cad-Cre; Cxcl12$^{+/+}$ and N-cad-Cre; Cxcl12$^{-/-}$ mice. Total RNA was extracted using the miRNeasy Mini Kit (QIAGEN) according to the manufacturer's instructions. RNA concentration was quantified using the Qubit RNA HS Assay Kit (Invitrogen) on a Qubit 4 Fluorometer, and RNA integrity was assessed using the Fragment Analyzer automated capillary electrophoresis system (Agilent Technologies). For library preparation, 1 µg of total RNA was used. Polyadenylated mRNA was isolated, fragmented and reverse-transcribed to generate double-stranded cDNA. Adapter ligation and library amplification were performed using the TruSeq Stranded mRNA Library Prep Kit (Illumina) following the manufacturer's protocol. Amplified libraries were purified using AxyPrep Mag PCR Clean-Up beads (Axygen). The final library quality and fragment size distribution were validated using the Fragment Analyzer, and concentrations were determined using the Qubit™ dsDNA HS Assay Kit (Invitrogen). Sequencing-ready libraries were diluted and pooled according to Illumina's standard protocol and paired-end sequencing was performed on an Illumina NextSeq 500 platform at the University of Missouri DNA Core Facility.

## Cell preparation for scRNA-seq

For tissue collection, femurs were collected after euthanasia and immediately placed in ice-cold PBS. The PM and DM regions of the bone were isolated. BM was extracted by crushing the bones with a mortar and pestle, followed by enzymatic digestion with collagenase/dispase at 37 °C for 45 min. The resulting cell suspension was washed, lysed and filtered through a 100-µm strainer. FACS isolation of non-haematopoietic cells: BM single-cell suspensions were stained with Anti-CD45-APC (clone I3/2.3, #147708, 1:200), anti-Ter-119-PE (clone TER-119, #116208, 1:200) and Anti-CD31-PerCP/Cyanine5.5 (clone W18222B, #160206, 1:200) to exclude haematopoietic cells and ECs. Live/dead discrimination was performed using DAPI. Viable CD45$^-$Ter-119$^-$CD31$^-$ (triple-negative) cells were sorted on a BD FACSAria II cell sorter.

## Single-cell sequencing

Single cells were encapsulated into emulsion droplets using Chromium Controller (10x Genomics). scRNA-seq libraries were constructed using Chromium Single Cell 3′ v2 Reagent Kit according to the manufacturer's protocol. In brief, the post-sorting sample volume was decreased, and cells were examined under a microscope and counted with a haemocytometer. Cells were then loaded in each channel with a target output of ~4,000 cells. Reverse transcription and library preparation were performed on C1000 Touch Thermal cycler with 96-Deep Well Reaction Module (Bio-Rad). Amplified cDNA and final libraries were evaluated on an Agilent Bioanalyzer using a High Sensitivity DNA Kit (Agilent Technologies). Individual libraries were diluted to 4 nM and pooled for sequencing. Pools were sequenced with 75 cycle run kits (26-bp Read1, 8-bp Index1 and 55-bp Read2) on the Novaseq 5000 Sequencing System (Illumina).

## scRNA-seq analysis

Single-cell transcriptomic profiling was performed as previously described with modifications[1]. In brief, raw sequencing data were processed using the Cell Ranger Single Cell Software Suite (v3.0.2, 10x Genomics) for quality control, sample demultiplexing, barcode assignment and 3′ gene counting. Reads were aligned to the mouse reference transcriptome (refdata-cellranger-mm39-3.0.0) with default parameters. Gene–barcode matrices were subsequently imported into the Seurat v3 package (R software). Quality control filters were applied to exclude low-quality or apoptotic cells: cells with >5% mitochondrial unique molecular identifier content and <1,500 detected genes were removed. Genes expressed in fewer than two cells were also excluded. Following normalization and scaling, highly variable

genes were identified and used for dimensionality reduction by PCA. Cell clustering was performed using the Seurat 'FindNeighbors' and 'FindClusters' functions with the first 20 principal components and a resolution parameter of 0.5. Uniform Manifold Approximation and Projection (UMAP) was applied for visualization in two-dimensional space. Differentially expressed genes (cluster-specific markers) were determined using the 'FindMarkers/FindAllMarkers' function with the Wilcoxon rank-sum test. To define cell identities, we compared cluster-specific gene signatures with canonical markers reported in prior literature[1]. For MSCs, markers including Lepr, Cdh2 (N-cadherin), Prx1, Osx (Sp7) and Nestin were examined. OLCs were identified by Bglap expression; fibroblasts by S100a4; chondrocytes by Acan and Col2a1; ECs by Cdh5; and pericytes by Acta2. Marker selection was based on well-established stromal lineage-defining studies and enabled robust annotation of BM niche subpopulations. All plots were generated in R using Seurat visualization functions.

## Spleen histology

The spleen was fixed in 4% phosphate-buffered formalin, dehydrated, embedded in frozen section compound 22 blue (Leica), sectioned at 6 μm and placed onto coated Superfrost Plus Microscope Slides (Fisher Scientific). Sections were stained with haematoxylin and eosin using the Thermo Scientific Shandon Rapid-Chrome H&E Frozen Section Staining Kit. Images were acquired using a BZ-8000 fluorescence microscope with 20× and 60× objectives (yielding 200× and 600× magnifications).

## Western blotting

Cells were lysed in Laemmli sample buffer (Sigma-Aldrich) supplemented with protease inhibitor cocktail (Roche Diagnostics). Samples were separated on SDS–PAGE gels (Bio-Rad) and transferred to nitrocellulose membranes (Bio-Rad) for protein detection. Primary antibodies were obtained from Cell Signaling Technology: Phospho-p44/42 MAPK (Erk1/2) (Thr202/Tyr204) (clone 20G11, Cell Signaling Technology, #75796S, 1:1,000), Phospho-NF-κB p65 (clone 93H1, Cell Signaling Technology, #3039S, 1:1,000), Phospho-STAT3 (Tyr705) (clone D3A7, Cell Signaling Technology, #9145S, 1:2,000), Phospho-p38 MAPK (clone D3F9, Cell Signaling Technology, #4092S, 1:1,000) and β-actin (clone C4, Santa Cruz Biotechnology, #sc-47778, 1:2,000). Secondary detection was performed with HRP-conjugated anti-rabbit and anti-mouse antibodies (R&D Systems, #HAF007 and #HAF005, respectively, 1:1,000), and protein bands were visualized using a chemiluminescent substrate (Invitrogen).

## Complete blood count assay

PB was collected from mice by retro-orbital bleeding into EDTA-coated microtubes to prevent coagulation. Complete blood counts, including haemoglobin concentration, total leukocyte count and platelet count, were measured using an automated haematology analyser (Hemavet 950FS, Drew Scientific) according to the manufacturer's instructions (performed by Comparative Clinical Pathology Services LLC). Each group contained five mice ($n = 5$), and results are presented as mean ± standard error of the mean (s.e.m.).

## Statistics and reproducibility

Data are expressed as mean ± s.e.m. Statistical analyses were performed using GraphPad Prism Version 9.0 (GraphPad Prism Software). For continuous variables, normality and homogeneity of variance were assessed using the Shapiro–Wilk and Brown–Forsythe tests, respectively. After confirming homogeneous variances and normality, two-group comparisons for means were performed using the two-sided Student $t$-test, and multigroup comparisons for means were performed using two-way analysis of variance with Holm–Šidák multiple comparison test. For data that did not pass either normality or equal variance test, two-group comparisons were performed using the Mann–Whitney

rank-sum test, and multigroup comparisons were performed using the Kruskal–Wallis one-way analysis of variance on ranks test with the Dunn post-hoc test. $P < 0.05$ was considered statistically significant. Animals were randomly assigned to experimental groups based on genotyping results using a simple randomization approach. Where applicable, littermates were distributed across treatment groups to minimize bias. Investigators were blinded to group allocation during data analysis but not during experimental procedures. No animals were excluded from the analyses. Data points were excluded only if predefined technical criteria were not met (for example, sample processing failure), before statistical analysis.

## Sample size determination

No statistical methods were used to predetermine sample sizes. Sample sizes were selected on the basis of prior experience with the AML transplantation and BM niche models and are comparable to those reported in previous publications using similar experimental systems[13,74]. The chosen sample sizes are consistent with established standards in the field and were sufficient to detect biologically meaningful differences with appropriate statistical tests.

## Reporting summary

Further information on research design is available in the Nature Portfolio Reporting Summary linked to this article.

## Data availability

Sequencing data generated in this study are publicly available via the NCBI Sequence Read Archive (SRA) under BioProject accession PRJNA1077712. RNA-seq data from the $Dpp4^{+/+}$ and $Dpp4^{-/-}$ AML model used in this study are available under SRA accession SRP323430. Source data are provided with this paper. All other data supporting the findings of this study are available from the corresponding author on reasonable request.

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

## Acknowledgements

We thank the University of Missouri Genomics Technology Core for expert support in generating the scRNA-seq data. We also gratefully acknowledge H. Qian (Karolinska Institute) for valuable discussions and insights that contributed to this project. X.K. is supported by the National Cancer Institute grant (R37CA241603, R01CA300057) and the American Cancer Society (RSG-23-1152630); L.L. is supported by Stowers Institute for Medical Research (SIMR-1004) and NCI grant (P30 CA168524).

## Author contributions

X.K. conceived and supervised the study; C.W. performed animal experiments and analyzed data; C.W. and W.Z. performed scRNA-seq and bulk RNA-seq; Y.P., C.W. and X.K. performed bioinformatics and

statistical analyses; C.W. performed immunofluorescence staining and analyzed imaging data; C.W., Y.P., W.Z., X.M. and X.K. verified the reproducibility of results; R.D.H. contributed to bone marrow imaging data analysis; L.L., R.D. and R.N. provided technical assistance and contributed to data analysis; C.W., Y.P. and X.K. wrote the original draft.

## Competing interests

The authors declare no competing interests.

## Additional information

**Extended data** is available for this paper at https://doi.org/10.1038/s41556-026-01939-3.

**Correspondence and requests for materials** should be addressed to XunLei Kang.

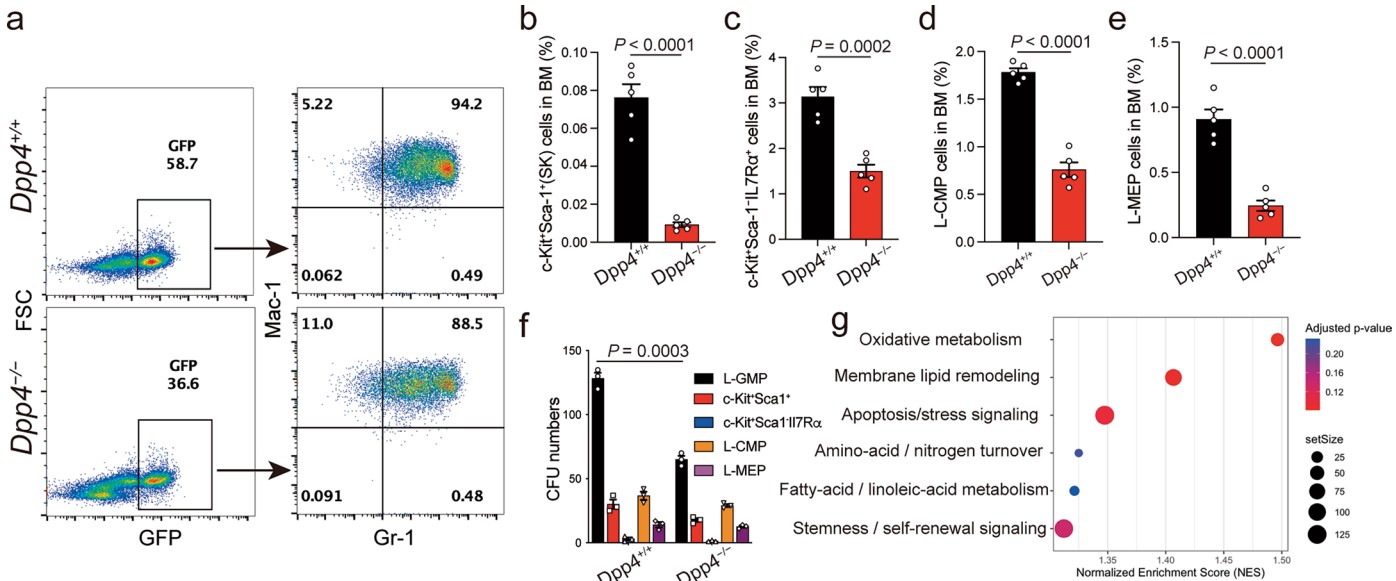

**Extended Data Fig. 1 | Decreased L-CMP and L-MEP frequencies and colony-forming capacity in Dpp4−/− bone marrow. (a)** Representative flow cytometry plots showing GFP+ leukemic cells co-expressing myeloid markers Gr-1 and Mac-1, confirming that GFP+ cells are predominantly myeloid blasts. Flow cytometric quantification of c-Kit+Sca-1+ (**b**), c-Kit+Sca-1−IL-7Rα+ cells (CLP-like progenitors, **c**), L-CMP (**d**) and L-MEP (**e**) like cells in the bone marrow of *Dpp4+/+* and *Dpp4−/−* mice (mean ± SEM, n = 5 biologically independent mice per group). (**f**) Colony-forming unit (CFU) assays were performed on sorted leukemic cell populations from the bone marrow of *Dpp4+/+* and *Dpp4−/−* mice, including L-GMP,

c-Kit+Sca-1+, c-Kit+Sca-1−IL-7Rα+, L-CMP, and L-MEP subsets(500 cells/well, mean ± SEM, n = 3 biologically independent wells). (**g**) Bubble plot showing KEGG pathway enrichment across selected metabolic and stress-response programs in *Dpp4−/−* compared with *Dpp4+/+* AML cells. The six major pathway groups include oxidative metabolism, membrane lipid remodeling, amino-acid/nitrogen turnover, fatty-acid/linoleic-acid metabolism, apoptosis/stress signaling, and stemness/self-renewal signaling. Bubble size reflects the number of genes within each pathway, and color represents the −log10(FDR) of enrichment.

| Quantification of CRUs | Survival ratio(latency days) | | |
|---|---|---|---|
| Transplanted cells | **PM** | **CM** | **DM** |
| 10 | 6/7 | 7/7 | 6/7 |
| 100 | 2/7 | 5/7 | 3/7 |
| 500 | 0/7 | 3/7 | 0/7 |
| 1000 | 0/7 | 0/7 | 0/7 |
| 5000 | 0/7 | 0/7 | 0/7 |
| Frequency of tumor initiating cells | 1:76.6 | 1:394 | 1:102 |
| 95% confidence interval | 34.1-173 | 212-731 | 46-226 |

**Extended Data Fig. 2 | Limiting dilution analysis of leukemia-initiating cell frequency across metaphysis, central marrow, and distal marrow regions.** Table summarizing the survival ratio (number of leukemic mice / total recipients) at varying transplanted cell doses (10–5000 cells) in PM, CM, and DM groups. The frequency of tumor-initiating cells (CRUs) was calculated using LDA (Limiting Dilution Analysis), with 95% confidence intervals provided.

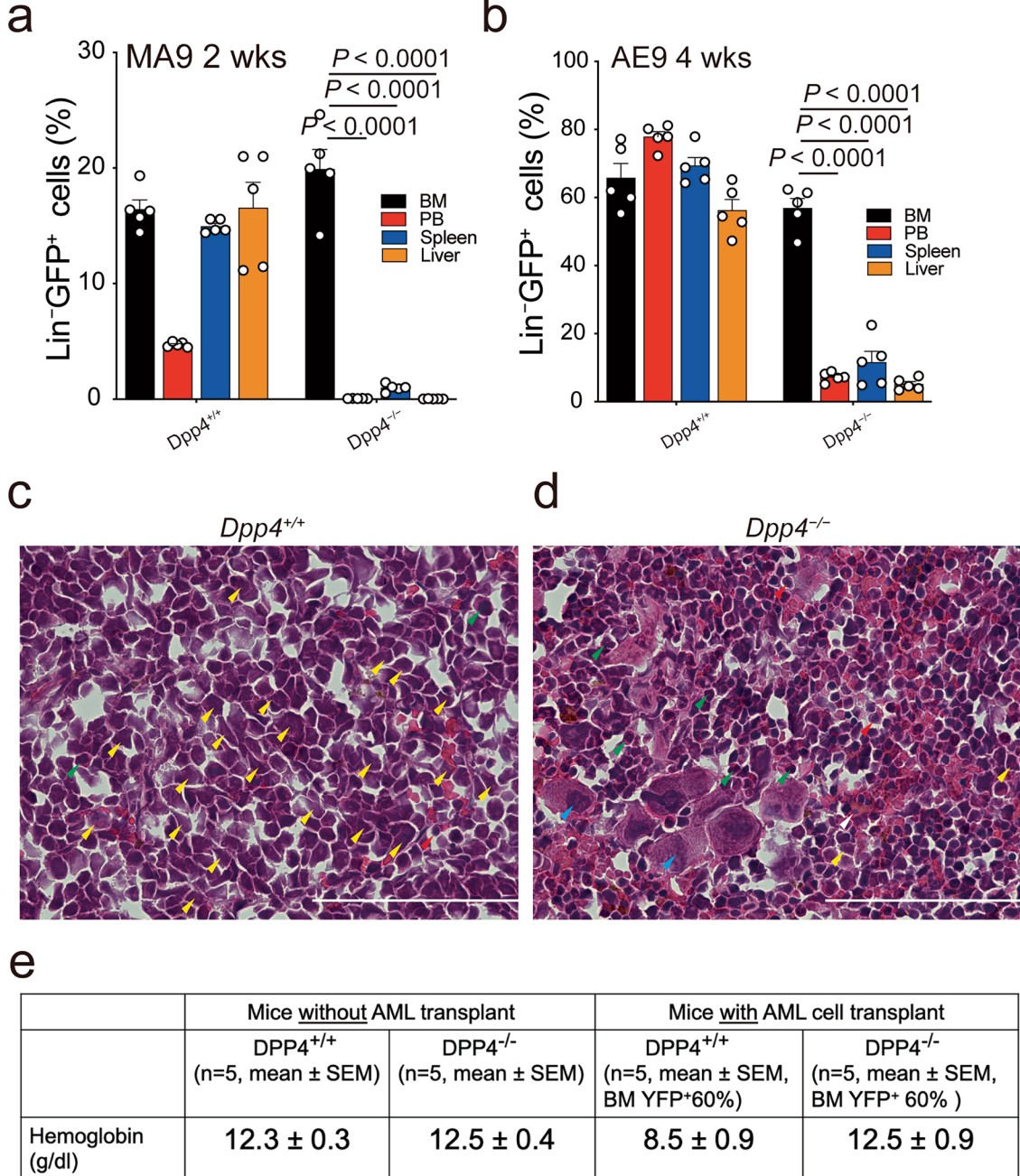

**Extended Data Fig. 3 | Reduced leukemia burden and preserved hematopoiesis in Dpp4-deficient AML mice.** (**a-b**) Engraftment efficiency measured by comparison of the proportions of MA9 Lin−GFP+ $Dpp4^{+/+}$ and $Dpp4^{-/-}$ AML cells in BM, PB, spleen, and liver 2 weeks (MA9) and 4 weeks (AE9) post transplantation. (mean ± SEM, n = 5 biologically independent mice per group.) Representative hematoxylin and eosin (H&E)−stained spleen sections from $Dpp4^{+/+}$ (**c**) and $Dpp4^{-/-}$ (**d**) AML mouse model. Cell populations, as annotated by a clinical pathologist, are indicated by specific arrowheads: yellow, blast cells; red, lymphocytes; green, granulocytes; blue, megakaryocytes; and white, macrophages. Scale bar: 100 μm. (**e**) Table of complete blood count (CBC) of DPP4+/+ and DPP4−/− AML cells transplanted mice (4-week and 13-week post-transplantation, respectively), at time points with similar leukemia cell percentages ( ~ 60%) in the BM. Data are presented as mean ± SEM. n numbers are provided in the corresponding panels and legends.

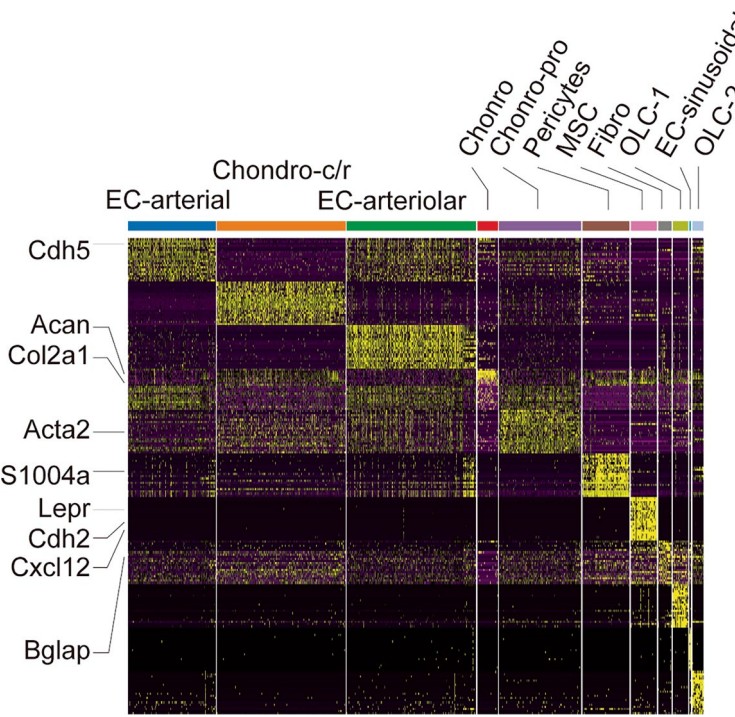

**Extended Data Fig. 4 | Scaled expression of representative marker genes across bone marrow niche cell populations.** Heatmap displays scaled expression of representative marker genes across defined cell populations identified in the bone marrow niche. Columns represent single cells grouped by annotated cluster identity (color-coded on top), and rows represent selected marker genes.

a

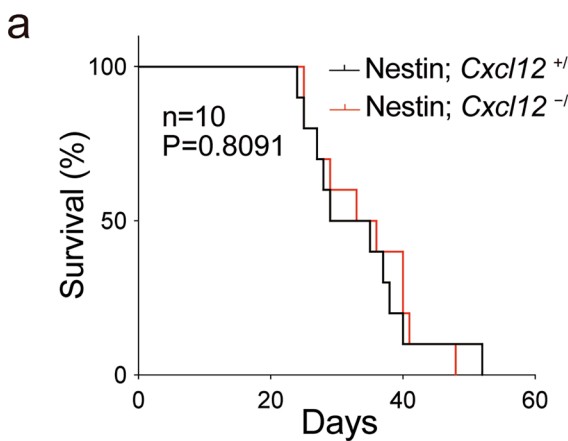

b

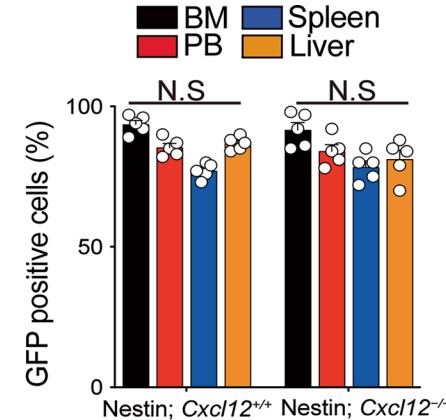

**Extended Data Fig. 5 | Nestin+ MSCs are dispensable for CXCL12-mediated AML retention.** (**a**) Survival curve of 500 control L-GMPs transplanted recipient Nestin; *Cxcl12*+/+ or Nestin; *Cxcl12*−/− mice (n = 10 mice; P = 0.8091, log-rank test).

(**b**) Comparison of the proportions of GFP+ Nestin; *Cxcl12*+/+ or Nestin; *Cxcl12*−/− AML cells in BM, PB, spleen, and liver 5 weeks post transplantation. (mean ± SEM, n = 5 biologically independent mice per group).

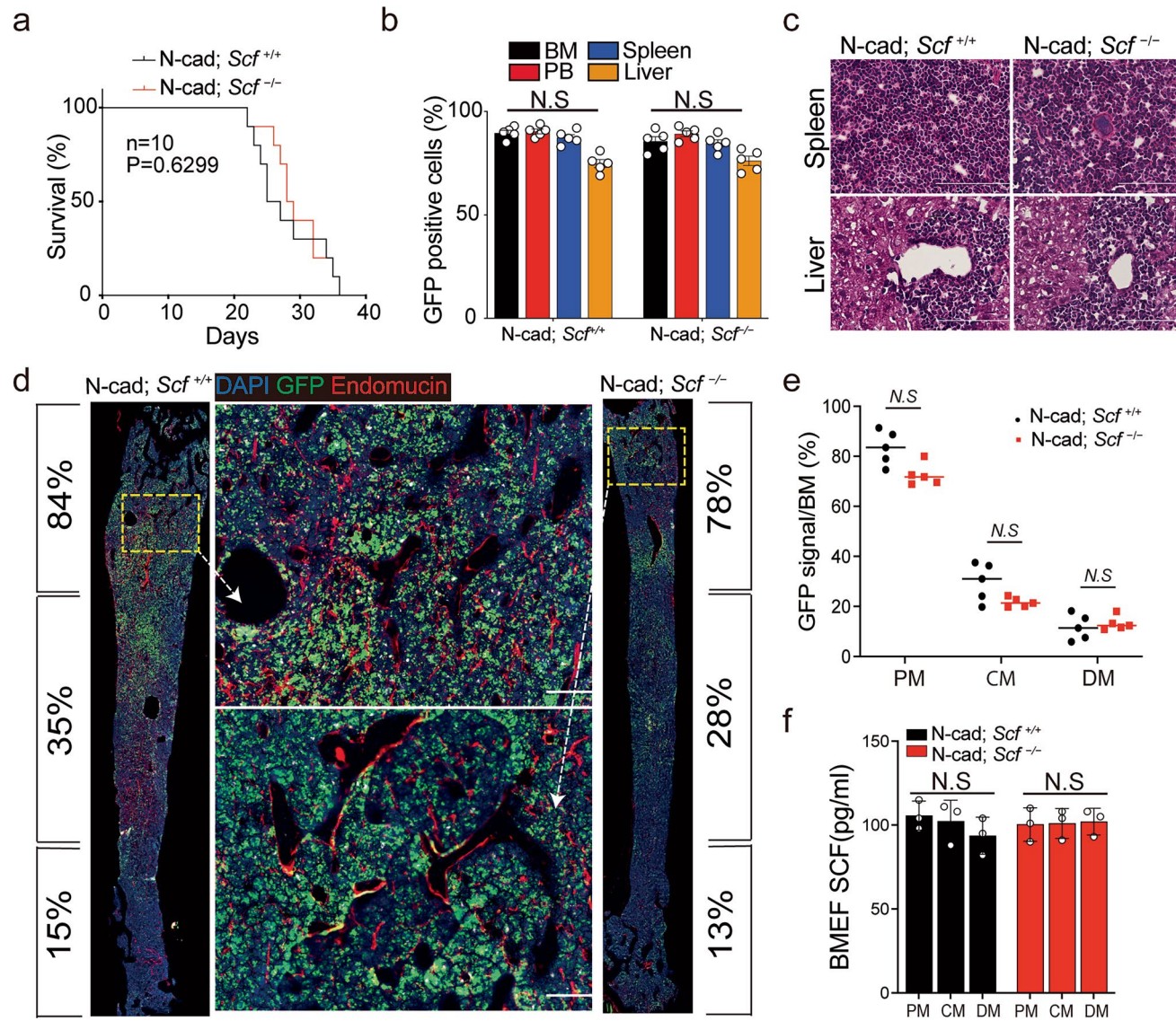

**Extended Data Fig. 6 | SCF from N-cad+ MSCs is dispensable for niche effects.** (**a**) Survival curve of 500 control L-GMPs transplanted recipient N-cad; $Scf^{+/+}$ or N-cad; $Scf^{-/-}$ mice (n = 10 mice; P = 0.6299, log-rank test). (**b**) Comparison of the proportions of GFP⁺-cad; $Scf^{+/+}$ or N-cad; $Scf^{-/-}$ AML cells in BM, PB, spleen, and liver 5 weeks post transplantation (mean ± SEM, n = 5 biologically independent mice per group.) (**c**) H&E staining for spleen and liver at 5-week post transplantation. (**d**) Representative images of BM sections from N-cad; $Scf^{+/+}$ (left) or N-cad; $Scf^{-/-}$ (right) AML mice model at week 6 respectively after

transplantation. Green, GFP + AML cells; red, endomucin staining blood vessels; blue, DAPI for nuclei. The calculated proportion of GFP+ signals in each of the three anatomical BM areas is listed to the left/right of the images. The dashed box indicates the area of focus. Scale bars, 100μm. (**e**) Summary of GFP⁺ signals in each BM area (PM, CM, DM). n = 5, BM sections from 5 mice, BM sections were collected at day 28,30,32,34,36. (**f**) SCF concentrations were measured in the proximal metaphysis (PM), central marrow (CM), and distal metaphysis (DM) regions. (mean ± SEM, n = 3 biologically independent mice per group).

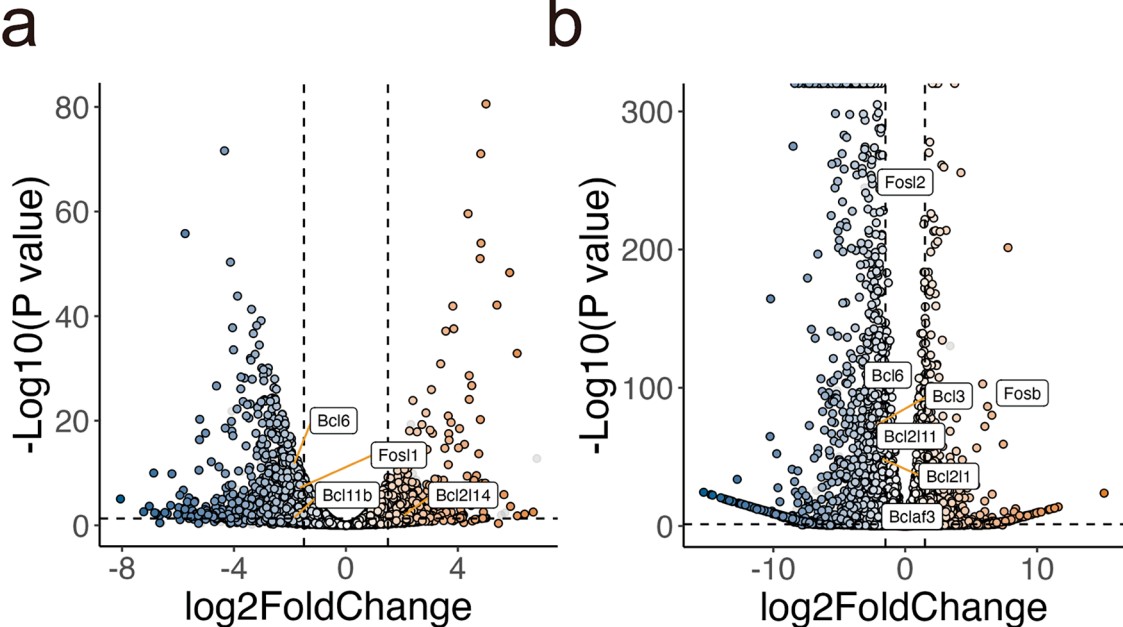

**Extended Data Fig. 7 | Downregulation of stemness and survival genes upon Dpp4 or Cxcl12 loss in AML. (a-b)** The volcano plots and heatmap illustrate differentially expressed genes in AML cells, comparing Dpp4$^{-/-}$ and N-cad; Cxcl12$^{-/-}$ against their respective controls. In both models, key survival and stemness-related genes (annotated) are significantly downregulated, suggesting an exhaustion-like phenotype driven by DPP4 depletion in AML cells or CXCL12 loss in N-cadherin MSCs.

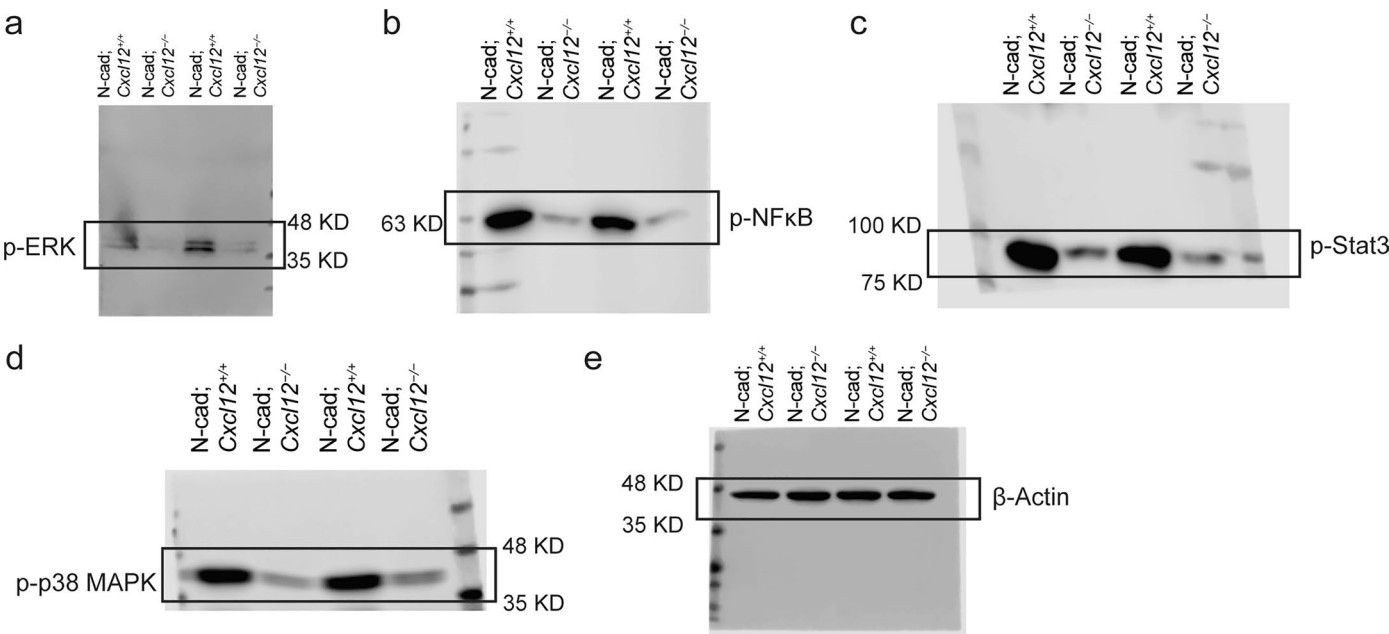

**Extended Data Fig. 8 | Reduced phosphorylation of STAT3, ERK1/2, p38, and NF-κB in N-cad; Cxcl12 / mice.** Western blot analysis showing reduced phosphorylation of ERK1/2 (**a**), NF-κB p65 (**b**), STAT3 (**c**), p38 MAPK (**d**), and in AML cells from N-cad; Cxcl12[−/−] mice compared to controls. β-Actin (**e**) served as a loading control. The immunoblotting experiments were independently repeated three times with similar results.

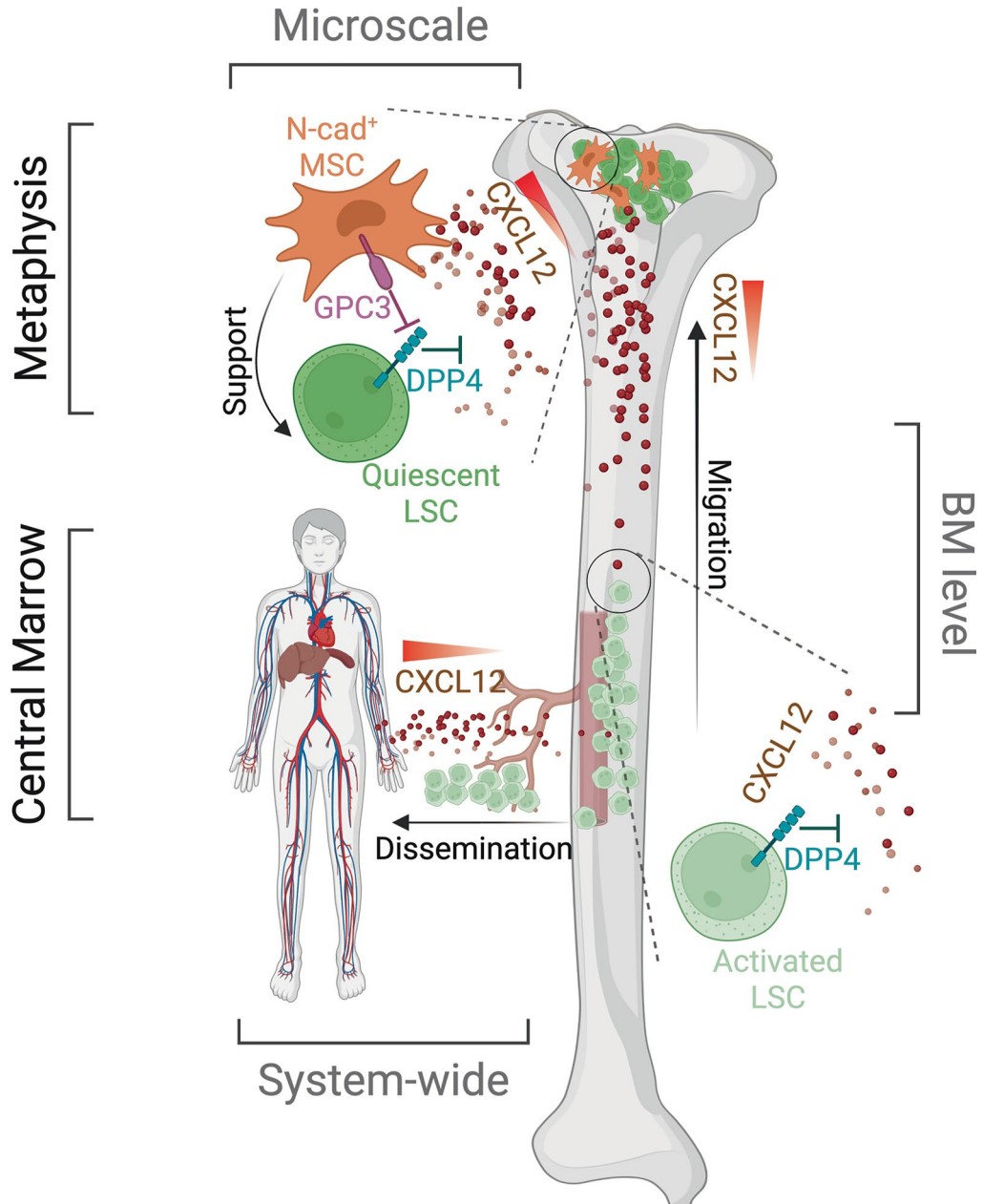

**Extended Data Fig. 9 | Model of the CXCL12–DPP4–GPC3 axis regulating leukemic stem cell localization within longitudinal bone marrow niches.** In the metaphysis, N-cad⁺ MSCs support quiescent leukemic stem cells (LSCs) by creating CXCL12-rich niches. These MSCs express GPC3, which inhibits DPP4 on LSCs, preserving local CXCL12 levels and reinforcing LSC retention and quiescence. In the central marrow, the absence of N-cad⁺ MSCs allows DPP4 to degrade CXCL12, forming a CXCL12 gradient from the central marrow to the metaphysis. This gradient drives LSC migration toward the metaphysis, where they become quiescent, while central marrow LSCs remain active and proliferative. Systemically, low CXCL12 in the marrow and high levels in peripheral tissues (for example, liver, spleen) promote LSC dissemination into circulation. Figure created in BioRender; Kang, X. https://biorender.com/6rzxcb (2026).

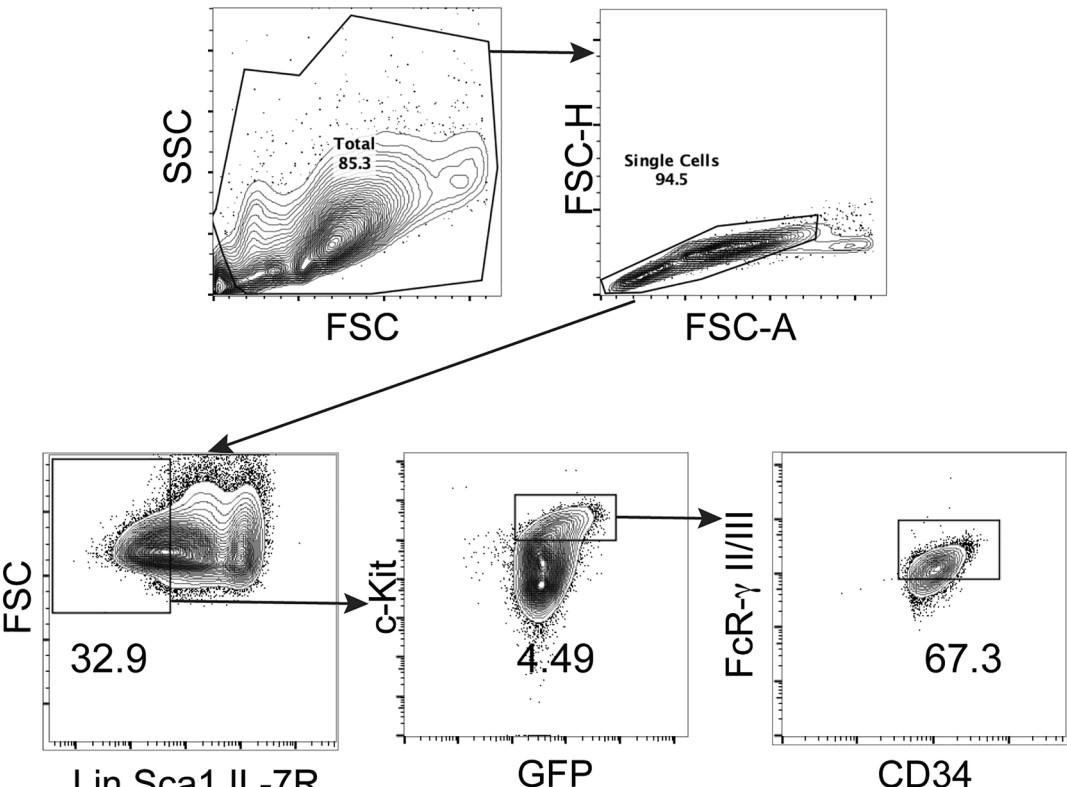

**Extended Data Fig. 10 | Representative flow cytometry gating for L-GMP population.** Bone marrow cells from leukemia-bearing mice were first gated on viable cells based on FSC/SSC properties, followed by singlet discrimination using FSC-A versus FSC-H. Lineage-negative (Lin⁻) Sca-1⁻ IL-7R⁻ populations were selected and further gated on c-Kit⁺ GFP⁺ AML cells. Within the GFP⁺ c-Kit⁺ compartment, L-GMPs were identified as FcγRII/III⁺ CD34⁺ cells. Numbers shown represent the percentage of cells within the parent gate. Data are representative of at least three independent experiments.

# Reporting Summary

## Statistics

For all statistical analyses, confirm that the following items are present in the figure legend, table legend, main text, or Methods section.

| n/a | Confirmed | |
|---|---|---|
| ☐ | ☒ | The exact sample size (*n*) for each experimental group/condition, given as a discrete number and unit of measurement |
| ☐ | ☒ | A statement on whether measurements were taken from distinct samples or whether the same sample was measured repeatedly |
| ☐ | ☒ | The statistical test(s) used AND whether they are one- or two-sided<br>*Only common tests should be described solely by name; describe more complex techniques in the Methods section.* |
| ☒ | ☐ | A description of all covariates tested |
| ☐ | ☒ | A description of any assumptions or corrections, such as tests of normality and adjustment for multiple comparisons |
| ☐ | ☒ | A full description of the statistical parameters including central tendency (e.g. means) or other basic estimates (e.g. regression coefficient) AND variation (e.g. standard deviation) or associated estimates of uncertainty (e.g. confidence intervals) |
| ☐ | ☒ | For null hypothesis testing, the test statistic (e.g. *F*, *t*, *r*) with confidence intervals, effect sizes, degrees of freedom and *P* value noted<br>*Give P values as exact values whenever suitable.* |
| ☒ | ☐ | For Bayesian analysis, information on the choice of priors and Markov chain Monte Carlo settings |
| ☒ | ☐ | For hierarchical and complex designs, identification of the appropriate level for tests and full reporting of outcomes |
| ☒ | ☐ | Estimates of effect sizes (e.g. Cohen's *d*, Pearson's *r*), indicating how they were calculated |

*Our web collection on statistics for biologists contains articles on many of the points above.*

## Software and code

Policy information about availability of computer code

| | |
|---|---|
| Data collection | Data collection was performed using commercial software only. No custom code was used.<br>• Flow cytometry raw data were acquired on an LSRFortessa X-20 cell analyzer (BD Biosciences) using FACSDiva v8.0 acquisition software.<br>• Immunofluorescence images were captured on a Keyence BZ-X800 fluorescence microscope using BZ-X Viewer v1.3.1.<br>• Single-cell library generation and sequencing were performed on a 10x Genomics Chromium Controller (Single Cell 3' v2 kit) and an Illumina NovaSeq 5000 (75-cycle kit). Raw base calls were converted to FASTQ files using bcl2fastq v2.20.<br>• Total RNA from FACS-sorted BM AML cells was used to prepare TruSeq stranded poly(A)+ libraries, which were sequenced on an Illumina NextSeq 500. |
| Data analysis | Data analysis was performed using commercially available and open-source software only. No custom code was used.<br>• RNA-Seq data:Reads were quality-checked, trimmed, aligned to mm10 with STAR, counted by featureCounts, and differential expression was called using DESeq2.<br>• scRNA-seq preprocessing (demultiplexing, alignment, UMI counting) with Cell Ranger Single Cell Software Suite v3.0.2 (10x Genomics).<br>• Downstream single-cell analysis in R v4.0.5 using the Seurat v3.2 package (normalization, clustering, TSNE).<br>• Statistical analyses (t-tests, ANOVA, nonparametric tests) and plotting in GraphPad Prism v9.0.<br>• Cytokine bead assay data were analyzed with LEGENDplex Data Analysis software v8.0 (BioLegend).<br>• Imaging quantification was carried out with Keyence BZ-X800 analyzer software v1.2.<br>• Flow cytometry data were analyzed using FlowJo v10 (BD Biosciences). |

For manuscripts utilizing custom algorithms or software that are central to the research but not yet described in published literature, software must be made available to editors and reviewers. We strongly encourage code deposition in a community repository (e.g. GitHub). See the Nature Portfolio guidelines for submitting code & software for further information.

## Data

Policy information about availability of data

All manuscripts must include a data availability statement. This statement should provide the following information, where applicable:
- Accession codes, unique identifiers, or web links for publicly available datasets
- A description of any restrictions on data availability
- For clinical datasets or third party data, please ensure that the statement adheres to our policy

Sequencing data generated in this study are publicly available in the NCBI Sequence Read Archive (SRA) under BioProject accession PRJNA1077712. RNA-seq data from the Dpp4$^{+/+}$ and Dpp4$^{-/-}$ AML model used in this study are available under SRA accession SRP323430. Source data are provided with this study. All other data supporting the findings of this study are available within the paper and its Supplementary Information files.

## Research involving human participants, their data, or biological material

Policy information about studies with human participants or human data. See also policy information about sex, gender (identity/presentation), and sexual orientation and race, ethnicity and racism.

| | |
|---|---|
| Reporting on sex and gender | N/A |
| Reporting on race, ethnicity, or other socially relevant groupings | N/A |
| Population characteristics | N/A |
| Recruitment | N/A |
| Ethics oversight | All animal experiments were approved by the Institutional Animal Care and Use Committee (IACUC) of the University of Missouri (protocol 65384). |

Note that full information on the approval of the study protocol must also be provided in the manuscript.

# Field-specific reporting

Please select the one below that is the best fit for your research. If you are not sure, read the appropriate sections before making your selection.

☒ Life sciences   ☐ Behavioural & social sciences   ☐ Ecological, evolutionary & environmental sciences

For a reference copy of the document with all sections, see nature.com/documents/nr-reporting-summary-flat.pdf

# Life sciences study design

All studies must disclose on these points even when the disclosure is negative.

| | |
|---|---|
| Sample size | No statistical methods were used to predetermine sample sizes. Sample sizes were selected based on prior experience with the AML transplantation and bone marrow niche models and are comparable to those reported in previous publications using similar experimental systems (Wang et al., Cell Reports, 2023; PMID: 36807138). The chosen sample sizes are consistent with established standards in the field and were sufficient to detect biologically meaningful differences with appropriate statistical tests. |
| Data exclusions | No data or animals were excluded from any analyses. All collected samples and experimental measurements are included in the reported results. |
| Replication | All key findings were reproduced in at least three independent experiments using distinct biological replicates (separate cohorts of mice or independent cell preparations). Consistent results were obtained across these replicates. |
| Randomization | After genotyping, mice were randomly assigned to experimental or control groups. Randomization was performed by a laboratory member not involved in data acquisition, using a random-number generator. |
| Blinding | Investigators performing data acquisition and image quantification were not blinded to genotype when handling animals (genotype-specific treatments required identifiable handling), but all data analyses (flow-cytometry gating, image quantification and statistical testing) were conducted with the analyst blinded to group allocation. |

# Reporting for specific materials, systems and methods

We require information from authors about some types of materials, experimental systems and methods used in many studies. Here, indicate whether each material, system or method listed is relevant to your study. If you are not sure if a list item applies to your research, read the appropriate section before selecting a response.

## Materials & experimental systems

| n/a | Involved in the study |
|-----|----------------------|
| ☐ | ☒ Antibodies |
| ☒ | ☐ Eukaryotic cell lines |
| ☒ | ☐ Palaeontology and archaeology |
| ☐ | ☒ Animals and other organisms |
| ☒ | ☐ Clinical data |
| ☒ | ☐ Dual use research of concern |
| ☒ | ☐ Plants |

## Methods

| n/a | Involved in the study |
|-----|----------------------|
| ☒ | ☐ ChIP-seq |
| ☐ | ☒ Flow cytometry |
| ☒ | ☐ MRI-based neuroimaging |

# Antibodies

| | |
|---|---|
| Antibodies used | Anti-CD3e-PE/Cyanine5 (clone 17A2, BioLegend, #100310, 1:200, flow cytometry), Anti-Ly6G/Ly6C (Gr-1)-PE/Cyanine5 (clone RB6-8C5, BioLegend, #108410, 1:200, flow cytometry), Anti-CD11b-PE/Cyanine5 (clone M1/70, BioLegend, #101210, 1:200, flow cytometry), Anti-CD45R-PE/Cyanine5 (clone RA3-6B2, BioLegend, #103210, 1:200, flow cytometry), Anti-Ter-119-PE/Cyanine5 (clone TER-119, BioLegend, #116210, 1:200, flow cytometry), Anti-CD117 (c-Kit)-APC (clone 2B8, BioLegend, #105812, 1:200, flow cytometry), Anti-Sca-1-PE-Cy7 (clone D7, BioLegend, #108114, 1:200, flow cytometry), Anti-CD150-PE (clone TC15-12F12.2, BioLegend, #115904, 1:200, flow cytometry), Anti-CD48-APC/Cyanine7 (clone HM48-1, BioLegend, #103432, 1:200, flow cytometry), Anti-Ki67-FITC (clone 16A8, BioLegend, #652410, 1:200, flow cytometry), Anti-CD16/32-PE (clone 93, BioLegend, #101308, 1:200, flow cytometry), Anti-CD34-FITC (clone RAM34, eBioscience, #11-0341-82, 1:200, flow cytometry), Anti-CD127-APC/Cyanine7 (clone A7R34, BioLegend, #135040, 1:200, flow cytometry), Anti-CD135-Brilliant Violet 421 (clone A2F10, BioLegend, #135314, 1:200, flow cytometry), Annexin V (BioLegend, #640941, 1:20, apoptosis assay), propidium iodide (BioLegend, #421301, 1:50, apoptosis assay), and Hoechst 34580 (BD Pharmingen, #565877, nuclear staining). For immunofluorescence, anti-Endomucin (goat polyclonal, R&D Systems, #AF4666, 1:100), Biotin anti-mouse Lineage Panel (clone 145-2C11; RB6-8C5; RA3-6B2; Ter-119; M1/70 BioLegend # 133307, 1:200), Biotin anti-mouse IL-7Rα (clone A7R34, BioLegend,#135006, 1:200), Biotin anti-mouse Sca-1 (clone D7, BioLegend,#108104, 1:200), PE-conjugated anti-CD34 (clone RAM34, BioLegend, 1:200), Alexa Fluor 647-conjugated anti-c-Kit (clone 2B8, Invitrogen, 1:200), and anti-GPC3 (rabbit polyclonal, Abcam, #ab216606, 1:200) were used. Secondary antibodies included donkey anti-goat Alexa Fluor® 555 (Invitrogen, 1:500), goat anti-rabbit Alexa Fluor® 750 (Invitrogen, 1:500), and Brilliant Violet 421®-conjugated streptavidin (BioLegend, 1:500). For western blotting, Phospho-p44/42 MAPK (Erk1/2) (Thr202/Tyr204) (clone 20G11, Cell Signaling Technology, #75796S, 1:1000), Phospho-NF-κB p65 (clone 93H1, Cell Signaling Technology, #3039S, 1:1000), Phospho-STAT3 (Tyr705) (clone D3A7, Cell Signaling Technology, #9145S, 1:2000), Phospho-p38 MAPK (clone D3F9, Cell Signaling Technology, #4092S, 1:1000), and β-actin (clone C4, Santa Cruz Biotechnology, #sc-47778, 1:2000) were used. HRP-conjugated anti-rabbit and anti-mouse secondary antibodies (R&D Systems, #HAF007 and #HAF005, respectively, 1:1000) were used for detection with a chemiluminescent substrate (Invitrogen). |
| Validation | All primary antibodies used in this study were commercially obtained and validated by the manufacturers for the indicated applications (flow cytometry, immunofluorescence, or western blot). Validation data, including specificity testing and performance validation, are available on the manufacturers' websites. In addition, the specificity of key antibodies was supported by expected molecular weight detection (for western blot) or appropriate staining patterns consistent with known biological expression profiles. |

# Animals and other research organisms

Policy information about studies involving animals; ARRIVE guidelines recommended for reporting animal research, and Sex and Gender in Research

| | |
|---|---|
| Laboratory animals | All animal experiments were performed in accordance with institutional guidelines and were approved by the Institutional Animal Care and Use Committee (IACUC) of the University of Missouri (protocol number 65384). C57BL/6NCrl mice (Mus musculus) were obtained from Charles River Laboratories, Inc. Dpp4 flox/flox mice were generated by breeding targeted C57Bl/6NTac-DPP4tm1a Wtsi/Ics mice (European Mouse Mutant Cell Repository, EUCOMM) with 129S4/Bl6-Gt (ROSA) 26Sortm2(FLP*) Sor/J (stock #012930, The Jackson Laboratory, Bar Harbor, ME). The offspring were further crossed with Vav-iCre mice (stock #018968, The Jackson Laboratory, Bar Harbor, ME), to generate Dpp4 fl/fl ; Vav-Cre mice. N-cad-tdTomato (N-cad-TdT), N-cad-CreER and Gpc3 fl/fl strains were generated by Dr. Linheng Li's lab. Cxcl12 fl/fl (stock #022457), Scf fl/fl (stock #017861) and Nestin-CreER (stock #016261) mice were purchased from Jackson Lab. To induce expression of Cre-ER recombinase, mice received tamoxifen via intraperitoneal injection (Sigma, 75 mg tamoxifen/kg body weight) as described. |
| Wild animals | The study did not involve wild-caught or non–laboratory animals. |
| Reporting on sex | Both sexes were included. No sex-specific differences were observed, and sex was not treated as a variable in statistical analyses. |
| Field-collected samples | The study did not use any field-collected tissues or organisms. |
| Ethics oversight | All mouse strains used in this study had a C57BL/6 genetic background. Both male and female mice aged 6–10 weeks were used unless otherwise specified. Animals were randomly assigned to experimental groups based on genotyping results. Investigators were blinded to group allocation during data analysis but not during experimental procedures. Sample sizes for each experiment are |

detailed in figure legends. Mice were housed in a specific pathogen–free facility under a 12-hour light/12-hour dark cycle at an ambient temperature of 20–24 °C and relative humidity of 40–60%, with ad libitum access to food and water.

Note that full information on the approval of the study protocol must also be provided in the manuscript.

# Plants

**Seed stocks**

N/A

**Novel plant genotypes**

N/A

**Authentication**

N/A

# Flow Cytometry

## Plots

Confirm that:

☒ The axis labels state the marker and fluorochrome used (e.g. CD4-FITC).

☒ The axis scales are clearly visible. Include numbers along axes only for bottom left plot of group (a 'group' is an analysis of identical markers).

☒ All plots are contour plots with outliers or pseudocolor plots.

☒ A numerical value for number of cells or percentage (with statistics) is provided.

## Methodology

**Sample preparation**

Peripheral blood (EDTA), bone marrow, spleen and liver were harvested and filtered through a 70 μm strainer. Red cells were lysed with ACK buffer, and remaining cells washed and resuspended in PBS + 2% FBS + 2 mM EDTA.

**Instrument**

Data were acquired on a BD LSRFortessa X-20 cell analyzer (BD Biosciences).

**Software**

Acquisition: FACSDiva v8.0
Analysis: FlowJo v10, using consistent gating templates across all replicates.

**Cell population abundance**

Each panel reports the percentage of marker-positive cells (mean ± SEM) from ≥ 5 biologically independent mice per group (with technical duplicates).

**Gating strategy**

1. FSC/SSC to exclude debris.
2. Singlet gate (FSC-A vs FSC-H).
3. Live/dead exclusion (PI- or Annexin V-negative).
4. Lineage (CD3ε, Gr-1, CD11b, B220, Ter-119) to identify Lin⁻ cells.
5. L-GMP:GFP+IL-7R−Lin−Sca-1−c-Kit+CD34+FcRII/III+
6. Bone marrow niche cells: 7AAD-GFP-CD45-

☒ Tick this box to confirm that a figure exemplifying the gating strategy is provided in the Supplementary Information.

