## [Peer Review File · Nature Cell Biology]

Longitudinal Localization of Leukemia Stem Cells Between Metaphysis and Central Marrow Governs Leukemic Stem Cell Behavior

Corresponding Author: Dr Xunlei kang

Version 0:

Decision Letter:

*Please delete the link to your author homepage if you wish to forward this email to co-authors.

Dear Dr Kang,

I am writing on behalf of my colleague Dr. Stylianos Lefkopoulos, who is out of the office.

Your manuscript, "Longitudinal Localization of Leukemia Stem Cells Between Metaphysis and Central Marrow Governs Leukemic Stem Cell Behavior", has now been seen by 2 referees, who are experts in leukemia stem cells, single cell genomics (referee 1); and bone marrow, leukemia, scRNA-seq (referee 2). As you will see from their comments (attached below) they find this work of potential interest, but have raised substantial concerns, which in our view would need to be addressed with considerable revisions before we can consider publication in Nature Cell Biology.

Nature Cell Biology editors discuss the referee reports in detail within the editorial team, including the chief editor, to identify key referee points that should be addressed with priority, and requests that are overruled as being beyond the scope of the current study. To guide the scope of the revisions, I have listed these points below. We are committed to providing a fair and constructive peer-review process, so please feel free to contact me if you would like to discuss any of the referee comments further.

In particular, it would be essential to:

A) Address the potential inconsistency in data from patient specimens versus mouse models:

Reviewer 1

"In patient specimens, leukemia-initiating function is not always enriched in phenotypically immature CD34+ fractions such as LMPPs, CMPs or GMPs; Even phenotypically mature CD34(-) cells will initiate leukemia in immune-compromised mice in some cases. Please present changes in frequencies of non-L-GMP cell fractions such as c-Kit+Sca1+ cells, c-Kit+Sca1(-)Il7Ra+ cells with or without DPP4 targeting in Fig. 1."

B) Strengthen the working mechanism as questioned by Reviewer 2:

Reviewer 2

"The exact mechanism of how DPP4 deletion alters the CXCL12 gradient remains elusive. Why would deletion of DPP4 in AML cells, which presumably infiltrate the entire BM niche, lead to such regional differences in levels of CXCL12? What happens to the CXCL12 gradient in other organs?"

C) All other referee concerns pertaining to strengthening existing data, providing controls, methodological details, clarifications and textual changes as applicable should also be addressed.

D) Finally please pay close attention to our guidelines on statistical and methodological reporting (listed below) as failure to do so may delay the reconsideration of the revised manuscript. In particular please provide:

We would be happy to consider a revised manuscript that would satisfactorily address these points, unless a similar paper is published elsewhere, or is accepted for publication in Nature Cell Biology in the meantime.

- ensure that it conforms to our format instructions and publication policies (see below and www.nature.com/nature/authors/).
- provide a point-by-point rebuttal to the full referee reports verbatim, as provided at the end of this letter.
- provide the completed Editorial Policy Checklist (found here <https://www.nature.com/authors/policies/Policy.pdf>), and Reporting Summary (found here <https://www.nature.com/authors/policies/ReportingSummary.pdf>). This is essential for reconsideration of the manuscript and these documents will be available to editors and referees in the event of peer review. For more information see <http://www.nature.com/authors/policies/availability.html> or contact me.

Nature Cell Biology is committed to improving transparency in authorship. As part of our efforts in this direction, we are now requesting that all authors identified as 'corresponding author' on published papers create and link their Open Researcher and Contributor Identifier (ORCID) with their account on the Manuscript Tracking System (MTS), prior to acceptance. ORCID helps the scientific community achieve unambiguous attribution of all scholarly contributions. You can create and link your ORCID from the home page of the MTS by clicking on 'Modify my Springer Nature account'. For more information please visit <http://www.springernature.com/orcid>.

Link Redacted

We would like to receive a revised submission within six months. We would be happy to consider a revision even after this timeframe, however if the resubmission deadline is missed and the paper is eventually published, the submission date will be the date when the revised manuscript was received.

We hope that you will find our referees' comments, and editorial guidance helpful. Please do not hesitate to contact me if there is anything you would like to discuss.

Best wishes,

Zhe Wang

Zhe Wang, PhD
Senior Editor
Nature Cell Biology

Tel: +44 (0) 207 843 4924
email: zhe.wang@nature.com

Reviewers' Comments:

Reviewer #1 (Remarks to the Author):

Using two commonly-used mouse AML models MLL-AF9 and AML-ETO, the authors showed, DPP4 conditional targeting dramatically changed multiple properties of leukemic stem cells (L-GMP): Quiescence, location, migration from BM to PB, and disease-initiating capacity. Since altered location of AML cells after DPP4 deletion was striking, the authors further examined interaction between L-GMPs and stromal cells resulting in identification of GPC3 as an essential molecule expressed by N-cad+ MSCs that interact with DPP4+ leukemic cells. Sophisticated experimental approaches including imaging, functional assessment and transplantation were taken throughout the manuscript.

The group previously reported DPP4 as a potential treatment target (Wang et al., Cell Reports, 2023). A different group described a possible role of DPP4 in CAF in breast cancer (PMID 38561380). No significant overlap was found between the submitted work and previously published papers, meaning that novelty nor impact of the submitted paper were negatively affected.

Here are some points that the authors may address to make the quality of the work even higher.

1. In patient specimens, leukemia-initiating function is not always enriched in phenotypically immature CD34+ fractions such as LMPPs, CMPs or GMPs; Even phenotypically mature CD34(-) cells will initiate leukemia in immune-compromised mice in some cases. Please present changes in frequencies of non-L-GMP cell fractions such as c-Kit+Sca1+ cells, c-Kit+Sca1(-)Il7Ra+ cells with or without DPP4 targeting in Fig. 1.
2. The authors used terms such as "exhaustion" phenotype in L-GMPs in Dpp4KO. Some key survival markers/genes such as Fos, Fosb and Bcl2 family members are shared between memory T lymphocytes, HSCs and tumor cells. I found DEGs and GSEA in Figure 5 to be rather superficial. Which genes or molecules in AML cells are critical in cell survival through interaction with Cxcl12 expressed by Nestin+ stromal cells? If the authors address these points, please discuss potential utility of targeting leukemic cell-stromal cell interactions as a new therapeutic option.

3. Bone structures are distinct between PM/DM and CM, with PM/DM having increased endosteal regions and possibility for a greater contact with hematopoietic cells. How do those locations contribute to normal HSC function? Effect of Dpp4 targeting appeared to be larger in PM than in DM. If this is statistically significant, please describe authors' interpretation. Please also explain what caused differences between PM and DM in multiple figure panels.
4. Clinically, MLL/KMT2A leukemia exhibit aggressive clinical course with abundant tumor cells both in BM and in circulation. Dpp4KO provides a biologically intriguing phenomenon in two AML mouse models. Discussion of how these models recapitulate patient leukemia and contribute to clinical translation by targeting DPP4-GPC3 interaction would strengthen this point.
5. scRNAseq demonstrated molecular interaction between the niche and AML. Please provide detailed information regarding cell population annotation in scRNAseq analysis. List of genes used to annotate each cell population and heat map representation of those genes in each defined cell population would be helpful. For instance, how was the MSC cluster defined? Which genes were chosen as the most characteristic genes in MSCs alongside CXCL12?
6. Kaplan-Meier curve presented in Figure 5h strongly supports the authors' claim. Since few AML cells were detected in PB, I presume leukemic infiltration in spleen, liver or other organs were also scant. When leukemic cells disturb normal hematopoiesis in BM, extra-medullary hematopoiesis frequently occurs in spleen. Please show histology of DPP4-wt and DPP4-/- spleen and distribution of normal and malignant blood cells in the organs. CBC (WBC, Hb, platelets) of the two mice would also be very helpful to understand discrepant distribution of AML cells in BM and PB in the model.
7. How do normal HSCs and L-GMPs share DPP4-GPC3 axis and how does CXCL12 gradients affect homeostasis of normal hematopoietic stem/progenitor cells?

Reviewer #2 (Remarks to the Author):

In this manuscript by Wang et al. the authors show that the distribution of AML cells in the bone marrow influences biological behavior of leukemia cells. Specifically, the authors highlight the influence of location in longitudinal niches of the bone marrow on leukemic stem cell (LSC) survival and proliferation. LSC seem to localize to the metaphysis, showing reduced stemness and aggressiveness when mobilized to the central marrow region. The authors deleted DPP4 in AML cells altered the CXCL12 gradient within the bone marrow niche. DPP4 is thought to deactivate CXCL12, while GPC3, found in N-cad+ cells, inhibits DPP4. The manuscript lacks clarity in writing and the data are not convincing. The manuscript also makes several overstatements and is inaccurate in several places. The aspect of niche location correlating with aggressivity is not novel (see Pereira et al., Kumar et al.).

Major points are:

1. The exact mechanism of how DPP4 deletion alters the CXCL12 gradient remains elusive. Why would deletion of DPP4 in AML cells, which presumably infiltrate the entire BM niche, lead to such regional differences in levels of CXCL12? What happens to the CXCL12 gradient in other organs?
2. The statements in the introduction that 'no effective treatments has been developed to restrict leukemia dissemination or eradicate LSC' is very simplified and partially inaccurate. In addition, many publications have studied the bidirectional interactions between leukemia cells and their niche. Hence reference 3 is only of limited value. A more objective and comprehensive view on the published studies is necessary.
3. Lines 74-77 and corresponding Figures 1-b. The sentences do not describe what is supposed to be seen in the figures. The abbreviations in the figure are not spelled out.
4. A homing experiment of DPP4 KO cells after 18 hours is necessary.
5. Please put your manuscript in relation to PMID: 27750279 and PMID: 29276143.
6. Figures 2a-b: How can the authors be sure to have isolated the LSC only from the CM, PM and DM? How was this performed?
7. A key experiment would be to delete CXCL12 specifically in the CM, PM and DM regions. Have the authors attempted an experiment along these lines? It is appreciated that this is difficult.
8. Line 253: Clotting abnormalities and stroke are usually not complications of AML (apart from APML).
9. Leukemia cells should be labelled by GFP and Gr1, e.g. Figure 1a-b.
10. Figure 1: What images represent 2 weeks, which 12 weeks?
11. Figure 1a legend: primary AML mouse models: Do you mean recipients of AML cells?
12. Figure 1d: Bone sections need to be collected on the same day for recipients of WT and KO cells!
13. Figure 1g: The authors mean WT versus KO L-GMP cells, not mice, correct? Please be accurate.
14. Fig. 2 e: It is hard to believe that the survival of mice is 50 days after transplantation of 500 L-GMP and that the p value for these survival curves is <0.0001.
15. Fig. 3a-b: Please compare WT versus KO, not different organs within one cohort.
16. Fig. 3a-b: Please compare Lin- GFP+ cells, not just GFP+ cells.
17. Fig. 3g-h: How do the authors explain the discrepant results of CXCL12 levels in BM versus plasma of recipients of WT versus KO AML cells?
18. Please show the levels of SCF in different parts of the bone.

Minor points:

1. English syntax and grammar need to be corrected.
2. Figure 1c: Spelling of cells.
3. Please spell out abbreviations, e.g. TIC etc.
4. Fig 3k: What is the label on the x-axis?

Methods should be written concisely, but should contain all elements necessary to allow interpretation and replication of the results. As a guideline, Methods sections typically do not exceed 3,000 words. The Methods should be divided into subsections listing reagents and techniques. When citing previous methods, accurate references should be provided and any alterations should be noted. Information must be provided about: antibody dilutions, company names, catalogue numbers and clone numbers for monoclonal antibodies; sequences of RNAi and cDNA probes/primers or company names and catalogue numbers if reagents are commercial; cell line names, sources and information on cell line identity and authentication. Animal studies and experiments involving human subjects must be reported in detail, identifying the committees approving the protocols. For studies involving human subjects/samples, a statement must be included confirming that informed consent was obtained. Statistical analyses and information on the reproducibility of experimental results should be provided in a section titled "Statistics and Reproducibility".

All Nature Cell Biology manuscripts submitted on or after March 21 2016 must include a Data availability statement at the end of the Methods section. For Springer Nature policies on data availability see <http://www.nature.com/authors/policies/availability.html>; for more information on this particular policy see <http://www.nature.com/authors/policies/data/data-availability-statements-data-citations.pdf>. The Data availability statement should include:

- Accession codes for primary datasets (generated during the study under consideration and designated as "primary accessions") and secondary datasets (published datasets reanalysed during the study under consideration, designated as "referenced accessions"). For primary accessions data should be made public to coincide with publication of the manuscript. A list of data types for which submission to community-endorsed public repositories is mandated (including sequence, structure, microarray, deep sequencing data) can be found here <http://www.nature.com/authors/policies/availability.html#data>.
- Unique identifiers (accession codes, DOIs or other unique persistent identifier) and hyperlinks for datasets deposited in an approved repository, but for which data deposition is not mandated (see here for details <http://www.nature.com/sdata/data-policies/repositories>).
- At a minimum, please include a statement confirming that all relevant data are available from the authors, and/or are included with the

manuscript (e.g. as source data or supplementary information), listing which data are included (e.g. by figure panels and data types) and mentioning any restrictions on availability.

- If a dataset has a Digital Object Identifier (DOI) as its unique identifier, we strongly encourage including this in the Reference list and citing the dataset in the Methods.

We recommend that you upload the step-by-step protocols used in this manuscript to protocols.io. More details can found at <https://www.protocols.io/help/publish-articles>.

All imaging data should be accompanied by scale bars, which should be defined in the legend.

Cropped images of gels/blots are acceptable, but need to be accompanied by size markers, and to retain visible background signal within the linear range (i.e. should not be saturated). The boundaries of panels with low background have to be demarked with black lines. Splicing of panels should only be considered if unavoidable, and must be clearly marked on the figure, and noted in the legend with a statement on whether the samples were obtained and processed simultaneously. Quantitative comparisons between samples on different gels/blots are discouraged; if this is unavoidable, it should only be performed for samples derived from the same experiment with gels/blots were processed in parallel, which needs to be stated in the legend.

The total number of Supplementary Figures (not including the "unprocessed scans" Supplementary Figure) should not exceed the number of main display items (figures and/or tables (see our Guide to Authors and March 2012 editorial <http://www.nature.com/ncb/authors/submit/index.html#supinfo>; <http://www.nature.com/ncb/journal/v14/n3/index.html#ed>). No restrictions apply to Supplementary Tables or Videos, but we advise authors to be selective in including supplemental data.

GUIDELINES FOR EXPERIMENTAL AND STATISTICAL REPORTING

REPORTING REQUIREMENTS – To improve the quality of methods and statistics reporting in our papers we have recently revised the reporting checklist we introduced in 2013. We are now asking all life sciences authors to complete two items: an Editorial Policy Checklist (found here https://www.nature.com/authors/policies/Policy.pdf](https://www.nature.com/authors/policies/Policy.pdf)) that verifies compliance with all required editorial policies and a reporting summary (found here https://www.nature.com/authors/policies/ReportingSummary.pdf](https://www.nature.com/authors/policies/ReportingSummary.pdf))

that collects information on experimental design and reagents. These documents are available to referees to aid the evaluation of the manuscript. Please note that these forms are dynamic 'smart pdfs' and must therefore be downloaded and completed in Adobe Reader. We will then flatten them for ease of use by the reviewers. If you would like to reference the guidance text as you complete the template, please access these flattened versions at http://www.nature.com/authors/policies/availability.html](http://www.nature.com/authors/policies/availability.html).

STATISTICS – Wherever statistics have been derived the legend needs to provide the n number (i.e. the sample size used to derive statistics) as a precise value (not a range), and define what this value represents. Error bars need to be defined in the legends (e.g. SD, SEM) together with a measure of centre (e.g. mean, median). Box plots need to be defined in terms of minima, maxima, centre, and percentiles. Ranges are more appropriate than standard errors for small data sets. Wherever statistical significance has been derived, precise p values need to be provided and the statistical test used needs to be stated in the legend. Statistics such as error bars must not be derived from n<3. For sample sizes of n<5 please plot the individual data points rather than providing bar graphs. Deriving statistics from technical replicate samples, rather than biological replicates is strongly discouraged. Wherever statistical significance has been derived, precise p values need to be provided and the statistical test stated in the legend.

Version 1:

Decision Letter:

*Please delete the link to your author homepage if you wish to forward this email to co-authors.

Dear Xunlei,

Your manuscript, "Longitudinal Localization of Leukemia Stem Cells Between Metaphysis and Central Marrow Governs Leukemic Stem Cell Behavior", has now been seen by the two original reviewers. As you will see from their comments (attached below), there is overall an improvement, but they still raise some important points. Although we are also interested in this study, we believe that their concerns should be addressed before we can consider publication in Nature Cell Biology.

Specifically, we would require that you please address the following:

(A) The concerns related to exhaustion claims, by performing serial replating assays. This was a point raised by reviewer #2 and which we discussed with reviewer #1 as well (see last comment in their report):

Reviewer #1:

"Regarding the comment by reviewer #2 on the need for transplantation assays, indeed serial transplantation assay is the most powerful method to prove "stemness" and "self-renewal capacity" of stem cells. But asking the authors to provide the in vivo assay would take too long. Instead, serial replating is a reasonable alternative good for the authors to strengthen their claim."

Reviewer #2:

"The only way to determine cell exhaustion is to perform serial transplantation assays, or at least serial replating assays."

(B) Address the other remaining point of reviewer #1:

"I thank the authors for providing me with information on a report showing consistently high expression of DPP4 in CD34+ human AML cells. CD34(-) LICs are present in some patients. Rather than relevance between DPP4 expression and leukemia initiation, I want the authors to touch upon heterogeneous leukemia in patients and to what extent the dynamics between LSCs and niche can be applied to human and to therapeutic indication in the future. In human, AML initiating cells and cell cycle quiescent AML cells may not perfectly overlap. This is one minor point that I want the authors to work on."

(C) Textually address the following points by reviewer #2:

"...not all edits were introduced into the manuscript, as far as this reviewer can tell. It also seems as if not all the manuscript's edits were clearly highlighted in another colour. This makes it difficult to evaluate the changes."

"It would be helpful to add to the abstract and the introduction, how DPP4 regulates CXCL12 and not just cite a publication (in the introduction)."

"The LSC-intrinsic effect in DPP4 KO AML cells could be -at least partly- due to metabolic changes. This is not even discussed. Along the same lines, the conclusions do not consider alternative explanations, e.g. at the end of the text for figure 1."

"The end of the manuscript sounds as if niche-targeting therapies in AML and other hematological malignancies has not been attempted, which is not true. The discussion as a whole is not balanced and does not put this manuscript into perspective of published work."

"Colony assays: How many progenitor cells were plated? A very difficult assay to do in view of the risk of contamination of these cells in such small number during the sorting process."

"Figures 1f and 1g: Why are there these differences? Some groups believe the L-GMP are the LSC in this model, while others believe that the c-Kit+ Sca-1+ fraction harbors the LSC. Therefore, how are these differences possible?"

"LSC do not kill the mice in this model. It is the leukemic blasts. Therefore, a conclusion about aggressivity of disease cannot be based on the number of LSC (in Figure 2f)."

"As mentioned, two markers need to be shown for leukemic cells, i.e. GFP and Gr1 or another myeloid marker. It is not sufficient to include the information in the rebuttal. The reader needs to be convinced too."

"What do the lanes in Supplementary Figure 7 represent?"

(D) Perform the following formatting changes:

- * Please rename your supplementary figures to extended data figures and reference them as such in the text.
- * Please make sure to either add also data on main figure 8 (to not have the model occupy a full main figure alone) or if you prefer to keep the model as a single figure alone, then you need to present it in an extended data figure. Please note that we allow from 5 to 8 main figures and up to 10 extended data figures.
- * Please move the table presented on figure 2 to an extended data figure.
- * Please provide a data availability statement and make sure to deposit your genome-wide datasets on a public repository. An example "Data Availability" statement follows, but please modify as appropriate to ensure that it applies to the data included in this study. Including this information is mandatory for publication of the manuscript:
"Sequencing data that support the findings of this study have been deposited in the Gene Expression Omnibus (GEO) under accession code GSEXXX. Previously published X data that were re-analysed here are available under accession code GSEZZZZ [AU: please complete the statement if applicable]. Source data are provided with this study. All other data supporting the findings of this study are available from the corresponding author on reasonable request."
- * Please ensure that all figures adhere to a maximum page size of roughly 180mm wide x 200mm high to fit standard page format and use a font size of no smaller than 7pt Arial or Helvetica throughout the figures

(E) Finally, please pay close attention to our guidelines on statistical and methodological reporting (listed below) as failure to do so may delay the reconsideration of the revised manuscript. In particular please provide:

- unprocessed images of all gels/blots in the form of a multi-page pdf file. Please ensure that blots/gels are labeled and the sections presented in the figures are clearly indicated.

- all numerical source data in Excel format, with data for different figures provided as different sheets within a single Excel file. The file should include source data giving rise to graphical representations and statistical descriptions in the paper and for all instances where the figures present representative experiments of multiple independent repeats, the source data of all repeats should be provided.

In contrast, although we agree with referee #2 regarding this specific point and we believe it would provide valuable insights, we consider it, as raised at this stage, to be beyond the scope of the present study. Thus, addressing it experimentally will not be necessary for reconsideration of the manuscript at this journal:

"Why don't the authors try the effect of DPP4 inhibitors in their model system to make a point that their findings are of translational value?"

We therefore invite you to take these points into account when revising the manuscript. In addition, when preparing the revision please:

- ensure that it conforms to our format instructions and publication policies (see below and www.nature.com/nature/authors/).

- provide a point-by-point rebuttal to the full referee reports verbatim, as provided at the end of this letter.

Nature Cell Biology is committed to improving transparency in authorship. As part of our efforts in this direction, we are now requesting that all authors identified as 'corresponding author' on published papers create and link their Open Researcher and Contributor Identifier (ORCID) with their account on the Manuscript Tracking System (MTS), prior to acceptance. ORCID helps the scientific community achieve unambiguous attribution of all scholarly contributions. You can create and link your ORCID from the home page of the MTS by clicking on 'Modify my Springer Nature account'. For more information please visit www.springernature.com/orcid.

Link Redacted

We would like to receive the revision within 4-5 weeks. If submitted within this time period, reconsideration of the revised manuscript will not be affected by related studies published elsewhere, or accepted for publication in Nature Cell Biology in the meantime. We would be happy to consider a revision even after this timeframe, but in that case we will consider the published literature at the time of resubmission when assessing the file.

We hope that you will find our referees' comments, and editorial guidance helpful. Please do not hesitate to contact me if there is anything you would like to discuss.

Best wishes,

Stelios

Dr. Stylianos Lefkopoulos
He/him/his
Senior Editor, Nature Cell Biology
Springer Nature
Heidelberger Platz 3, 14197 Berlin, Germany

E-mail: stylianos.lefkopoulos@springernature.com
Bluesky: [@slefkopoulos.bsky.social](https://bsky.app/profile/@slefkopoulos.bsky.social)
LinkedIn: www.linkedin.com/in/stylianos-lefkopoulos-81b007a0

Reviewers' Comments:

Reviewer #1 (Remarks to the Author):

In the revised manuscript, interaction through DPP4, GPC3 and CXCL12 was more convincingly proven by robust in vivo experiments and data analyses with a state-of-art technique. As I made comments in the first round of review, to what degree findings obtained from the two mouse models the authors used (MLL-AF9 and AML-ETO) are helpful in understanding and targeting human leukemia with more heterogeneity and complexity has been a critical key to assess impact and quality of the paper.

I thank the authors for providing me with information on a report showing consistently high expression of DPP4 in CD34+ human AML cells. CD34(-) LICs are present in some patients. Rather than relevance between DPP4 expression and leukemia initiation, I want the authors to touch upon heterogenous leukemia in patients and to what extent the dynamics between LSCs and niche can be applied to human and to therapeutic indication in the future. In human, AML initiating cells and cell cycle quiescent AML cells may not perfectly overlap. This is one minor point that I want the authors to work on.

Genomic analyses in Response Figure 2 picked up several survival and metabolic genes differentially expressed. It would be nice to

validate in the future which genes or gene sets are mainly involved, but I do not ask the authors to include the validation in the revision.

I am satisfied with response and response figures for other comments of mine such as how normal HSCs and L-GMPs share DPP4-GPC3-CXCL12 axis and leukemic infiltration in non-BM organs.

Again, in the revised ms, I feel it convincing that bone structural differences and differential niche availability between PM/DM and CM regions result in quite dynamic distinction of stem cell properties of leukemic cells. Critical molecules between niche and L-GMPs and mechanistic insights into the biology were nicely provided.

Regarding the comment by reviewer #2 on the need for transplantation assays, indeed serial transplantation assay is the most powerful method to prove "stemness" and "self-renewal capacity" of stem cells. But asking the authors to provide the in vivo assay would take too long. Instead, serial replating is a reasonable alternative good for the authors to strengthen their claim.

Reviewer #2 (Remarks to the Author):

In this revised manuscript the authors have addressed the reviewers' questions mostly in the rebuttal letter, but not all edits were introduced into the manuscript, as far as this reviewer can tell. It also seems as if not all the manuscript's edits were clearly highlighted in another colour. This makes it difficult to evaluate the changes. Issues with the manuscript remain:

1. It would be helpful to add to the abstract and the introduction, how DPP4 regulates CXCL12 and not just cite a publication (in the introduction).
2. The LSC-intrinsic effect in DPP4 KO AML cells could be -at least partly- due to metabolic changes. This is not even discussed. Along the same lines, the conclusions do not consider alternative explanations, e.g. at the end of the text for figure 1. The only way to determine cell exhaustion is to perform serial transplantation assays, or at least serial replating assays. In some cases a conclusion is missing altogether in the results section of some figures.
3. Why don't the authors try the effect of DPP4 inhibitors in their model system to make a point that their findings are of translational value?
4. The end of the manuscript sounds as if niche-targeting therapies in AML and other hematological malignancies has not been attempted, which is not true. The discussion as a whole is not balanced and does not put this manuscript into perspective of published work.
5. Colony assays: How many progenitor cells were plated? A very difficult assay to do in view of the risk of contamination of these cells in such small number during the sorting process.
6. Figures 1f and 1g: Why are there these differences? Some groups believe the L-GMP are the LSC in this model, while others believe that the c-Kit+ Sca-1+ fraction harbors the LSC. Therefore, how are these differences possible?
7. LSC do not kill the mice in this model. It is the leukemic blasts. Therefore, a conclusion about aggressivity of disease cannot be based on the number of LSC (in Figure 2f).
8. As mentioned, two markers need to be shown for leukemic cells, i.e. GFP and Gr1 or another myeloid marker. It is not sufficient to include the information in the rebuttal. The reader needs to be convinced too.
9. What do the lanes in Supplementary Figure 7 represent?

GUIDELINES FOR SUBMISSION OF NATURE CELL BIOLOGY ARTICLES

ARTICLE FORMAT

ABSTRACT – should not exceed 150 words and should be unreferenced. This paragraph is the most visible part of the paper and should briefly outline the background and rationale for the work, and accurately summarize the main results and conclusions. Key genes, proteins and organisms should be specified to ensure discoverability of the paper in online searches.

TEXT – the main text consists of the Introduction, Results, and Discussion sections and must not exceed 3500 words including the abstract. The Introduction should expand on the background relating to the work. The Results should be divided in subsections with subheadings, and should provide a concise and accurate description of the experimental findings. The Discussion should expand on the findings and their implications. All relevant primary literature should be cited, in particular when discussing the background and specific findings.

REFERENCES – are limited to a total of 70 in the main text and Methods combined,. They must be numbered sequentially as they appear in the main text, tables and figure legends and Methods and must follow the precise style of Nature Cell Biology references. References only cited in the Methods should be numbered consecutively following the last reference cited in the main text. References only associated with Supplementary Information (e.g. in supplementary legends) do not count toward the total reference limit and do not need to be cited in numerical continuity with references in the main text. Only published papers can be cited, and each publication cited should be included in the numbered reference list, which should include the manuscript titles. Footnotes are not permitted.

Methods should be written concisely, but should contain all elements necessary to allow interpretation and replication of the results. As a guideline, Methods sections typically do not exceed 3,000 words. The Methods should be divided into subsections listing reagents and techniques. When citing previous methods, accurate references should be provided and any alterations should be noted. Information must be provided about: antibody dilutions, company names, catalogue numbers and clone numbers for monoclonal antibodies; sequences of RNAi and cDNA probes/primers or company names and catalogue numbers if reagents are commercial; cell line names, sources and information on cell line identity and authentication. Animal studies and experiments involving human subjects must be reported in detail, identifying the committees approving the protocols. For studies involving human subjects/samples, a statement must be included confirming that informed consent was obtained. Statistical analyses and information on the reproducibility of experimental results should be provided in a section titled "Statistics and Reproducibility".

All Nature Cell Biology manuscripts submitted on or after March 21 2016, must include a Data availability statement as a separate section after Methods but before references, under the heading "Data Availability". For Springer Nature policies on data availability see <http://www.nature.com/authors/policies/availability.html>; for more information on this particular policy see <http://www.nature.com/authors/policies/data/data-availability-statements-data-citations.pdf>. The Data availability statement should include:

- Accession codes for primary datasets (generated during the study under consideration and designated as "primary accessions") and secondary datasets (published datasets reanalysed during the study under consideration, designated as "referenced accessions"). For primary accessions data should be made public to coincide with publication of the manuscript. A list of data types for which submission to community-endorsed public repositories is mandated (including sequence, structure, microarray, deep sequencing data) can be found here <http://www.nature.com/authors/policies/availability.html#data>.
- Unique identifiers (accession codes, DOIs or other unique persistent identifier) and hyperlinks for datasets deposited in an approved repository, but for which data deposition is not mandated (see here for details <http://www.nature.com/sdata/data-policies/repositories>).
- At a minimum, please include a statement confirming that all relevant data are available from the authors, and/or are included with the manuscript (e.g. as source data or supplementary information), listing which data are included (e.g. by figure panels and data types) and mentioning any restrictions on availability.
- If a dataset has a Digital Object Identifier (DOI) as its unique identifier, we strongly encourage including this in the Reference list and citing the dataset in the Methods.

We recommend that you upload the step-by-step protocols used in this manuscript to [protocols.io](https://www.protocols.io). More details can found at <https://www.protocols.io/help/publish-articles>.

DISPLAY ITEMS – main display items are limited to 6-8 main figures and/or main tables. For Supplementary Information see below.

FIGURES – Colour figure publication costs \$395 per colour figure. All panels of a multi-panel figure must be logically connected and arranged as they would appear in the final version. Unnecessary figures and figure panels should be avoided (e.g. data presented in small tables could be stated briefly in the text instead).

All imaging data should be accompanied by scale bars, which should be defined in the legend.

Cropped images of gels/blots are acceptable, but need to be accompanied by size markers, and to retain visible background signal within the linear range (i.e. should not be saturated). The boundaries of panels with low background have to be demarked with black lines. Splicing of panels should only be considered if unavoidable, and must be clearly marked on the figure, and noted in the legend with a statement on whether the samples were obtained and processed simultaneously. Quantitative comparisons between samples on different gels/blots are discouraged; if this is unavoidable, it has be performed for samples derived from the same experiment with gels/blots were

processed in parallel, which needs to be stated in the legend.

- For line art, graphs, charts and schematics we prefer Adobe Illustrator (.AI), Encapsulated PostScript (.EPS) or Portable Document Format (.PDF). Files should be saved or exported as such directly from the application in which they were made, to allow us to restyle them according to our journal house style.
- We accept PowerPoint (.PPT) files if they are fully editable. However, please refrain from adding PowerPoint graphical effects to objects, as this results in them outputting poor quality raster art. Text used for PowerPoint figures should be Helvetica (preferred) or Arial.
- We do not recommend using Adobe Photoshop for designing figures, but we can accept Photoshop generated (.PSD or .TIFF) files only if each element included in the figure (text, labels, pictures, graphs, arrows and scale bars) are on separate layers. All text should be editable in 'type layers' and line-art such as graphs and other simple schematics should be preserved and embedded within 'vector smart objects' - not flattened raster/bitmap graphics.
- Some programs can generate Postscript by 'printing to file' (found in the Print dialogue). If using an application not listed above, save the file in PostScript format or email our Art Editor, Allen Beattie for advice (a.beattie@nature.com).

Regardless of format, all figures must be vector graphic compatible files, not supplied in a flattened raster/bitmap graphics format, but should be fully editable, allowing us to highlight/copy/paste all text and move individual parts of the figures (i.e. arrows, lines, x and y axes, graphs, tick marks, scale bars etc). The only parts of the figure that should be in pixel raster/bitmap format are photographic images or 3D rendered graphics/complex technical illustrations.

Unprocessed scans of all key data generated through electrophoretic separation techniques need to be presented in a supplementary figure that should be labeled and numbered as the final supplementary figure, and should be mentioned in every relevant figure legend. This figure does not count towards the total number of figures and is the only figure that can be displayed over multiple pages, but should be provided as a single file, in PDF or TIFF format. Data in this figure can be displayed in a relatively informal style, but size markers and the figures panels corresponding to the presented data must be indicated.

The total number of Supplementary Figures (not including the "unprocessed scans" Supplementary Figure) should not exceed the number of main display items (figures and/or tables (see our Guide to Authors and March 2012 editorial <http://www.nature.com/ncb/authors/submit/index.html#suppinfo>; <http://www.nature.com/ncb/journal/v14/n3/index.html#ed>). No restrictions apply to Supplementary Tables or Videos, but we advise authors to be selective in including supplemental data.

GUIDELINES FOR EXPERIMENTAL AND STATISTICAL REPORTING

REPORTING REQUIREMENTS – To improve the quality of methods and statistics reporting in our papers we have recently revised the reporting checklist we introduced in 2013. We are now asking all life sciences authors to complete two items: an Editorial Policy Checklist (found here <https://www.nature.com/authors/policies/Policy.pdf>) that verifies compliance with all required editorial policies and a Reporting Summary (found here <https://www.nature.com/authors/policies/ReportingSummary.pdf>) that collects information on experimental design and reagents. These documents are available to referees to aid the evaluation of the manuscript. Please note that these forms are dynamic 'smart pdfs' and must therefore be downloaded and completed in Adobe Reader. We will then flatten them for ease of use by the reviewers. If you would like to reference the guidance text as you complete the template, please access these flattened versions at <http://www.nature.com/authors/policies/availability.html>.

Version 2:

Decision Letter:

*Please delete the link to your author homepage if you wish to forward this email to co-authors.

Dear Xunlei,

As you know, your manuscript, "Longitudinal Localization of Leukemia Stem Cells Between Metaphysis and Central Marrow Governs Leukemic Stem Cell Behavior", has now been seen by the two original reviewers. Additionally, we have communicated to both reviewers your response to the comments of reviewer #2 in this round of review (which you provided to me via email and gave me permission to share with the reviewers) and have now obtained their comments on it. As you will see from their comments (attached below) they continue to raise some important points, which we believe should be addressed before we can consider publication in Nature Cell Biology.

Specifically, it would be essential that you address (including with new experiments and data) the following points:

Reviewer #1:

"... it is a little confusing even to me that transplantation of LSK resulted in Gr1+ AML colonies. LSK stands for Lin(-)Sca1+Kit+. If LICs (leukemia-initiating-cells) express Gr1, a critical lineage marker for mouse hematopoiesis, the authors should lose all the leukemic cells."

Reviewer #2:

"... it is VERY unusual to claim LSC activity or perform CFU-assays with Mac1+ or Gr-1+ cells. I do not agree, that Mac1 or Gr1+ cells are widely believed to have LSC activity. Why did they not use L-GMP or LSK cells, as they claim these to be the LSC in their paper? My suggestion would be that the authors perform a CFU assay with L-GMP and LSK cells to prove their point."

Finally, please pay close attention to our guidelines on statistical and methodological reporting (listed below) as failure to do so may delay the reconsideration of the revised manuscript. In particular please provide:

- unprocessed images of all gels/blots in the form of a multi-page pdf file. Please ensure that blots/gels are labeled and the sections presented in the figures are clearly indicated.

- all numerical source data in Excel format, with data for different figures provided as different sheets within a single Excel file. The file should include source data giving rise to graphical representations and statistical descriptions in the paper and for all instances where the figures present representative experiments of multiple independent repeats, the source data of all repeats should be provided.

We therefore invite you to take these points into account when revising the manuscript. In addition, when preparing the revision please:

- ensure that it conforms to our format instructions and publication policies (see below and <https://www.nature.com/nature/for-authors>).

- provide a point-by-point rebuttal to the full referee reports verbatim, as provided at the end of this letter.

- for any revision that includes light microscopy data, we ask our authors to please include a completed light microscopy reporting table https://www.nature.com/documents/Light_microscopy_reporting_table.xlsx to ensure the methods are described thoroughly. The table will be available to reviewers and ultimately published should the manuscript be accepted at the journal.

- provide the completed Reporting Summary (found here <https://www.nature.com/documents/nr-reporting-summary.pdf>). This is essential for reconsideration of the manuscript and will be available to editors and referees in the event of peer review. For more information see <http://www.nature.com/authors/policies/availability.html> or contact me.

EXTENDED DATA FIGURES

Nature Cell Biology is committed to improving transparency in authorship. As part of our efforts in this direction, we are now requesting that all authors identified as 'corresponding author' on published papers create and link their Open Researcher and Contributor Identifier (ORCID) with their account on the Manuscript Tracking System (MTS), prior to acceptance. ORCID helps the scientific community achieve unambiguous attribution of all scholarly contributions. You can create and link your ORCID from the home page of the MTS by clicking on 'Modify my Springer Nature account'. For more information please visit www.springernature.com/orcid.

This journal strongly supports public availability of data. Please place the data used in your paper into a public data repository, or alternatively, present the data as Supplementary Information. If data can only be shared on request, please explain why in your Data Availability Statement, and also in the correspondence with your editor. Please note that for some data types, deposition in a public repository is mandatory - more information on our data deposition policies and available repositories appears below.

Link Redacted

We would like to receive the revision within 6-8 weeks. We would be happy to consider a revision even after this timeframe, but in that case we will consider the published literature at the time of resubmission when assessing the file.

We hope that you will find our referees' comments, and editorial guidance helpful. Please do not hesitate to contact me if there is anything you would like to discuss.

Best wishes,

Stelios

Dr. Stylianos Lefkopoulos
He/him/his
Senior Editor, Nature Cell Biology
Springer Nature
Heidelberger Platz 3, 14197 Berlin, Germany

E-mail: stylianos.lefkopoulos@springernature.com

Bluesky: @slefkopoulos.bsky.social
LinkedIn: www.linkedin.com/in/stylianios-lefkopoulos-81b007a0

Reviewers' Comments:

Reviewer #1 (Remarks to the Author):

To the authors:

The authors nicely presented importance of DPP4-GPC3-CXCL12 axis and differential availability of niche between PM/DM and CM regions. Potential translation of the findings has been nicely discussed as well. I fully appreciate all the effort by the authors to revise the manuscript.

ADDITIONAL COMMENTS ON THE REMAINING ISSUE BY REVIEWER #2 AND THE AUTHORS' RESPONSE TO IT
Expression of Gr1 in mouse AML models is fine. In particular for MLL-AF9 AML model, in addition to references that authors listed, I found another showing Gr1+ MLL-AF9 AML (PMCID PMC3951989).
"MN1-CMP leukemias showed a high proportion of myeloid blasts in bone marrow (Figure 1D), infiltrated spleen (Figure 1E and F) and liver (data not shown), and had an immature immunophenotype (high c-kit expression) with some expression of myeloid markers Gr-1 and CD11b, similar to MN1 leukemias generated by transduction of 5-FU treated bulk bone marrow cells (Figure 1G)."
For MLL-AF9, we sometimes find CD34(-) LICs in human as well.

However, it is a little confusing even to me that transplantation of LSK resulted in Gr1+ AML colonies. LSK stands for Lin(-)Sca1+Kit+. If LICs (leukemia-initiating-cells) express Gr1, a critical lineage marker for mouse hematopoiesis, the authors should lose all the leukemic cells.

Reviewer #2 (Remarks to the Author):

In this second revision the authors have addressed the reviewers' concerns. However, the authors claim that the LSC fraction is harbored in the LSK or the L-GMP fraction. Then how is it possible that sorted AML cells positive for Gr1, a marker for mature and immature myeloid cells, give rise to colonies in a colony-forming assay, which assesses the function of leukemic stem and progenitor cells?

ADDITIONAL COMMENTS ON THE AUTHORS' RESPONSE TO MY REMAINING ISSUE ABOVE
In their first reference (Krivtsov et al), which is the manuscript which identified the MLL-AF9 model, I found no reference to Gr1 or Mac1 as a marker for leukaemia stem cells (LSC) (as they identify the L-GMP cells as the LSC). In the second manuscript by Somerville et al. I will admit that they mention LSC activity in the Gr1+ or Mac1+ fraction.
Therefore, Wang et al. may be correct. However, it is VERY unusual to claim LSC activity or perform CFU-assays with Mac1+ or Gr-1+ cells. I do not agree, that Mac1 or Gr1+ cells are widely believed to have LSC activity. Why did they not use L-GMP or LSK cells, as they claim these to be the LSC in their paper? My suggestion would be that the authors perform a CFU assay with L-GMP and LSK cells to prove their point.

GUIDELINES FOR SUBMISSION OF NATURE CELL BIOLOGY ARTICLES

ARTICLE FORMAT

ABSTRACT – should not exceed 150 words and should be unreferenced. This paragraph is the most visible part of the paper and should

briefly outline the background and rationale for the work, and accurately summarize the main results and conclusions. Key genes, proteins and organisms should be specified to ensure discoverability of the paper in online searches.

TEXT – the main text consists of the Introduction, Results, and Discussion sections and must not exceed 3500 words including the abstract. The Introduction should expand on the background relating to the work. The Results should be divided in subsections with subheadings, and should provide a concise and accurate description of the experimental findings. The Discussion should expand on the findings and their implications. All relevant primary literature should be cited, in particular when discussing the background and specific findings.

REFERENCES – are limited to a total of 70 in the main text and Methods combined. They must be numbered sequentially as they appear in the main text, tables and figure legends and Methods and must follow the precise style of Nature Cell Biology references. References only cited in the Methods should be numbered consecutively following the last reference cited in the main text. References only associated with Supplementary Information (e.g. in supplementary legends) do not count toward the total reference limit and do not need to be cited in numerical continuity with references in the main text. Only published papers can be cited, and each publication cited should be included in the numbered reference list, which should include the manuscript titles. Footnotes are not permitted.

Methods should be written concisely, but should contain all elements necessary to allow interpretation and replication of the results. As a guideline, Methods sections typically do not exceed 3,000 words. The Methods should be divided into subsections listing reagents and techniques. When citing previous methods, accurate references should be provided and any alterations should be noted. Information must be provided about: antibody dilutions, company names, catalogue numbers and clone numbers for monoclonal antibodies; sequences of RNAi and cDNA probes/primers or company names and catalogue numbers if reagents are commercial; cell line names, sources and information on cell line identity and authentication. Animal studies and experiments involving human subjects must be reported in detail, identifying the committees approving the protocols. For studies involving human subjects/samples, a statement must be included confirming that informed consent was obtained. Statistical analyses and information on the reproducibility of experimental results should be provided in a section titled "Statistics and Reproducibility".

All Nature Cell Biology manuscripts submitted on or after March 21 2016, must include a Data availability statement as a separate section after Methods but before references, under the heading "Data Availability". For Springer Nature policies on data availability see <http://www.nature.com/authors/policies/availability.html>; for more information on this particular policy see <http://www.nature.com/authors/policies/data/data-availability-statements-data-citations.pdf>. The Data availability statement should include:

- Accession codes for primary datasets (generated during the study under consideration and designated as "primary accessions") and secondary datasets (published datasets reanalysed during the study under consideration, designated as "referenced accessions"). For primary accessions data should be made public to coincide with publication of the manuscript. A list of data types for which submission to community-endorsed public repositories is mandated (including sequence, structure, microarray, deep sequencing data) can be found here <http://www.nature.com/authors/policies/availability.html#data>.
- Unique identifiers (accession codes, DOIs or other unique persistent identifier) and hyperlinks for datasets deposited in an approved repository, but for which data deposition is not mandated (see here for details <http://www.nature.com/sdata/data-policies/repositories>).
- At a minimum, please include a statement confirming that all relevant data are available from the authors, and/or are included with the manuscript (e.g. as source data or supplementary information), listing which data are included (e.g. by figure panels and data types) and mentioning any restrictions on availability.
- If a dataset has a Digital Object Identifier (DOI) as its unique identifier, we strongly encourage including this in the Reference list and citing the dataset in the Methods.

We recommend that you upload the step-by-step protocols used in this manuscript to [protocols.io](http://www.protocols.io). More details can be found at <http://www.protocols.io/help/publish-articles>.

DISPLAY ITEMS – main display items are limited to 6-8 main figures and/or main tables. For Supplementary Information see below.

FIGURES – Colour figure publication costs \$395 per colour figure. All panels of a multi-panel figure must be logically connected and arranged as they would appear in the final version. Unnecessary figures and figure panels should be avoided (e.g. data presented in small tables could be stated briefly in the text instead).

All imaging data should be accompanied by scale bars, which should be defined in the legend. Cropped images of gels/blots are acceptable, but need to be accompanied by size markers, and to retain visible background signal within the linear range (i.e. should not be saturated). The boundaries of panels with low background have to be demarked with black lines.

Splicing of panels should only be considered if unavoidable, and must be clearly marked on the figure, and noted in the legend with a statement on whether the samples were obtained and processed simultaneously. Quantitative comparisons between samples on different gels/blots are discouraged; if this is unavoidable, it has to be performed for samples derived from the same experiment with gels/blots were processed in parallel, which needs to be stated in the legend.

Regardless of format, all figures must be vector graphic compatible files, not supplied in a flattened raster/bitmap graphics format, but should be fully editable, allowing us to highlight/copy/paste all text and move individual parts of the figures (i.e. arrows, lines, x and y axes, graphs, tick marks, scale bars etc). The only parts of the figure that should be in pixel raster/bitmap format are photographic images or 3D rendered graphics/complex technical illustrations.

Unprocessed scans of all key data generated through electrophoretic separation techniques need to be presented in a supplementary figure that should be labeled and numbered as the final supplementary figure, and should be mentioned in every relevant figure legend. This figure does not count towards the total number of figures and is the only figure that can be displayed over multiple pages, but should be provided as a single file, in PDF or TIFF format. Data in this figure can be displayed in a relatively informal style, but size markers and the figures panels corresponding to the presented data must be indicated.

The total number of Supplementary Figures (not including the "unprocessed scans" Supplementary Figure) should not exceed the number of main display items (figures and/or tables (see our Guide to Authors and March 2012 editorial <http://www.nature.com/ncb/authors/submit/index.html#suppinfo>; <http://www.nature.com/ncb/journal/v14/n3/index.html#ed>). No restrictions apply to Supplementary Tables or Videos, but we advise authors to be selective in including supplemental data.

GUIDELINES FOR EXPERIMENTAL AND STATISTICAL REPORTING

REPORTING REQUIREMENTS – We ask authors to complete a Reporting Summary that collects information on experimental design and reagents. We hope this will aid in your evaluation of the paper. The Reporting Summary can be found here <https://www.nature.com/documents/nr-reporting-summary.pdf> Please note that these forms are dynamic 'smart pdfs' and must therefore be downloaded and completed in Adobe Reader. We will then flatten them for ease of use. If you would like to reference the guidance text as you complete the template, please access these flattened versions at <http://www.nature.com/authors/policies/availability.html>.

Version 3:

Decision Letter:

9th February 2026

Dear Xunlei,

Thank you for submitting your revised manuscript "Longitudinal Localization of Leukemia Stem Cells Between Metaphysis and Central Marrow Governs Leukemic Stem Cell Behavior" (NCB-A57950C). It has now been seen by the original referees and their comments are below. The reviewers find that the paper has improved in revision, and therefore we'll be happy in principle to publish it in Nature Cell Biology, pending minor revisions to comply with our editorial and formatting guidelines.

If the current version of your manuscript is in a PDF format, please email us a copy of the file in an editable format (Microsoft Word or LaTeX)-- we cannot proceed with PDFs at this stage.

Thank you again for your interest in Nature Cell Biology. Please do not hesitate to contact me if you have any questions.

Best wishes,
Stelios

Dr. Stylianos Lefkopoulos
He/him/his
Senior Editor, Nature Cell Biology
Springer Nature
Heidelberger Platz 3, 14197 Berlin, Germany

E-mail: stylianos.lefkopoulos@springernature.com
Bluesky: [@slefkopoulos.bsky.social](https://bsky.app/profile/@slefkopoulos.bsky.social)
LinkedIn: www.linkedin.com/in/stylianos-lefkopoulos-81b007a0

Reviewer #1 (Remarks to the Author):

Again, I appreciated convincing data of DPP4-GPC3-CXCL12 pathways and high impact of niche in distinct locations (PM/DM and CM).

Regarding a confusion regarding purification of "LSK" for donor cells, I now understand that the authors did not lose leukemia-initiating cells that expressed Gr1+ cells for donor cells by detailed explanation and presentation of sorting strategies of L-GMP and SK cells.

Reviewer #2 (Remarks to the Author):

The authors have now addressed my comments.

Version 4:

Decision Letter:

Dear Xunlei,

I am pleased to inform you that your manuscript, "Longitudinal Localization of Leukemia Stem Cells Between Metaphysis and Central Marrow Governs Leukemic Stem Cell Behavior", has now been accepted for publication in *Nature Cell Biology*.

Over the next few weeks, your paper will be copyedited to ensure that it conforms to *Nature Cell Biology* style. Once your paper is typeset, you will receive an email with a link to choose the appropriate publishing options for your paper and our Author Services team will be in touch regarding any additional information that may be required.

Publication is conditional on the manuscript not being published elsewhere and on there being no announcement of this work to any media outlet until the online publication date in *Nature Cell Biology*.

Please note that *Nature Cell Biology* is a Transformative Journal (TJ). Authors may publish their research with us through the traditional subscription access route or make their paper immediately open access through payment of an article-processing charge (APC). Authors will not be required to make a final decision about access to their article until it has been accepted. [Find out more about Transformative Journals](https://www.springernature.com/gp/open-research/transformative-journals)

Authors may need to take specific actions to achieve compliance with funder and institutional open access mandates. If your research is supported by a funder that requires immediate open access (e.g. according to [Plan S principles](https://www.springernature.com/gp/open-science/plan-s-compliance) or the [NIH public access policy](https://www.springernature.com/gp/open-science/us-federal-agency-compliance)) then you should select the gold OA route, and we will direct you to the compliant route where possible. Because authors warrant under our subscription licensing terms that they haven't committed to licensing any version of their article under a licence inconsistent with the terms of our agreement – including the applicable embargo period – publication under the subscription model isn't suitable for authors whose funders require no embargo.

If you have not already done so, we strongly recommend that you upload the step-by-step protocols used in this manuscript to protocols.io (<https://protocols.io>), an open online resource that allows researchers to share their detailed experimental know-how. All uploaded protocols are made freely available and are assigned DOIs for ease of citation. Protocols and Nature Portfolio journal papers in which they are used can be linked to one another, and this link is clearly and prominently visible in the online versions of both. Authors who performed the specific experiments can act as primary authors for the Protocol as they will be best placed to share the methodology details, but the Corresponding Author of the present research paper should be included as one of the authors. By uploading your Protocols onto protocols.io, you are enabling researchers to more readily reproduce or adapt the methodology you use, as well as increasing the visibility of your protocols and papers. You can also establish a dedicated workspace to collect your lab Protocols. Further information can be found at <https://www.protocols.io/help/publish-articles>.

Nature Cell Biology encourages authors presenting evidence for cell, biological, molecular, and genetic interactions to consider communicating these findings using Biofactoid (<https://biofactoid.org/>). This tool helps users share a searchable representation of interactions (e.g. binding, gene expression, post-translational modification) between genes, gene products, or chemicals. Information added to Biofactoid, with author attribution, is shared on social media and public databases, such as Pathway Commons, where it can be discovered and analyzed in the context of a large and growing corpus of knowledge.

With kind regards,
Stelios

Dr. Stylianos Lefkopoulos
He/him/his
Senior Editor, Nature Cell Biology
Springer Nature
Heidelberger Platz 3, 14197 Berlin, Germany

E-mail: stylianos.lefkopoulos@springernature.com
Bluesky: [@slefkopoulos.bsky.social](https://bsky.app/profile/@slefkopoulos.bsky.social)
LinkedIn: www.linkedin.com/in/stylianos-lefkopoulos-81b007a0

** Visit the Springer Nature Editorial and Publishing website at http://editorial-jobs.springernature.com?utm_source=ejp_NCB_email&utm_medium=ejp_NCB_email&utm_campaign=ejp_NCB for more information about our career opportunities. If you have any questions please click [here](mailto:editorial.publishing.jobs@springernature.com).**

and the source, provide a link to the Creative Commons license, and indicate if changes were made.

Longitudinal Localization of Leukemia Stem Cells Between Metaphysis and Central Marrow Governs Leukemic Stem Cell Behavior

Response to the Editor and the Reviewers

We thank the Editor and the Reviewers for their thorough and insightful feedback on our manuscript. We have carefully revised the text and performed additional experiments as requested, which have substantially strengthened our study. A summary of our key revisions is provided below, followed by detailed point-by-point responses.

Reviewers' Comments:

Reviewer #1 (Remarks to the Author):

Using two commonly used mouse AML models MLL-AF9 and AML-ETO, the authors showed, DPP4 conditional targeting dramatically changed multiple properties of leukemic stem cells (L-GMP): Quiescence, location, migration from BM to PB, and disease-initiating capacity. Since altered location of AML cells after DPP4 deletion was striking, the authors further examined interaction between L-GMPs and stromal cells resulting in identification of GPC3 as an essential molecule expressed by N-cad⁺ MSCs that interact with DPP4⁺ leukemic cells. Sophisticated experimental approaches including imaging, functional assessment and transplantation were taken throughout the manuscript.

The group previously reported DPP4 as a potential treatment target (Wang et al., Cell Reports, 2023). A different group described a possible role of DPP4 in CAF in breast cancer (PMID 38561380). No significant overlap was found between the submitted work and previously published papers, meaning that novelty nor impact of the submitted paper were negatively affected.

Here are some points that the authors may address to make the quality of the work even higher.

We greatly appreciate Reviewer #1's thorough evaluation and insightful suggestions, which have significantly enhanced our manuscript. Below, we provide detailed responses addressing each point raised:

1. In patient specimens, leukemia-initiating function is not always enriched in phenotypically immature CD34⁺ fractions such as LMPPs, CMPs or GMPs; Even phenotypically mature CD34⁽⁻⁾ cells will initiate leukemia in immune-compromised mice in some cases. Please present changes in frequencies of non-L-GMP cell fractions such as c-Kit⁺Sca1⁺ cells, c-Kit⁺Sca1⁽⁻⁾Il7Ra⁺ cells with or without DPP4 targeting in Fig. 1.

We agree with the reviewer's point about the heterogeneity of leukemia-initiating cells in AML patients, including KMT2A-rearranged leukemia. We (Wang et al., 2023; PMID: 36807138) and others (Herrmann et al., 2020b; PMID: 33085758) have observed that DPP4 is highly expressed in CD34⁺ human AML cells. Notably, we also detected differential DPP4 expression between CD34⁺ AML stem/progenitor cells and CD34⁻ AML blasts. Interestingly, several human AML samples lacking CD34⁺ populations still exhibited moderate levels of DPP4 expression (PMID: 36807138). These observations are further supported by studies using the MLL-AF9 AML model, which phenotypically resembles CD34⁻ AML in patients, in which DPP4 expression is also significantly upregulated in mouse MLL-AF9 AML cells. Collectively, these studies suggest that DPP4 may contribute to leukemogenic activity across both CD34⁺ and CD34⁻ AML subtypes, broadening our understanding beyond prior work.

We focused on L-GMP population for our LSC niche study based on the foundational studies (PMID: 16862118) using MLL-AF9 oncogene (a direct representation of KMT2A-rearranged

leukemia) to develop mouse AML model, which identified L-GMP as the most enriched population for leukemia stem cells (**Response Figure 1a**, PMID: 16862118). To address the reviewer's suggestion regarding whether DPP4 also regulates other AML progenitor populations beyond L-GMPs, we analyzed the frequencies of non-L-GMP cell fractions in the bone marrow (**Response Figures 1b–c**, **new Figure 1g-h**), including c-Kit⁺Sca-1⁺ cells and c-Kit⁺Sca-1⁻IL-7Rα⁺ cells (CLP-like cells), as well as L-CMP (c-Kit⁺Sca-1⁻CD34⁺FcγRIII/III⁻) and L-MEP (c-Kit⁺Sca-1⁻CD34⁻FcγRIII/III⁻) populations (**Response Figures 1d–e**, **new supplementary Figure 1a-b**). In Dpp4^{-/-} mice, the frequency of c-Kit⁺Sca-1⁺ cells was markedly reduced (~0.01%) compared to Dpp4^{+/+} controls (~0.075%, $P < 0.0001$). Similarly, c-Kit⁺Sca-1⁻IL-7Rα⁺ cells decreased from ~3.2% in Dpp4^{+/+} to ~1.3% in Dpp4^{-/-} mice ($P = 0.0002$). As shown in Response Figure 1d–e, the frequencies of both L-CMP and L-MEP cells were significantly decreased in Dpp4^{-/-} mice compared to Dpp4^{+/+} controls. Specifically, the L-CMP population declined from ~1.8% to ~0.7% ($P < 0.0001$), and the L-MEP fraction dropped from ~0.9% to ~0.3% ($P < 0.0001$). These findings indicate that DPP4 deficiency leads to a significant reduction in early hematopoietic and lymphoid-biased progenitors. To assess the functional capacity of non-L-GMP populations, we performed CFU assays in comparison with L-GMP populations (**Response Figure 1f**, **new supplementary Figure 1c**). Taken together with our prior data showing L-GMP expansion in Dpp4^{-/-} AML, these results suggest that loss of DPP4 skews the progenitor composition by promoting L-GMP accumulation at the expense of other non-L-GMP populations.

2. The authors used terms such as “exhaustion” phenotype in L-GMPs in Dpp4KO. Some key survival markers/genes such as Fos, Fosb and Bcl2 family members are shared between memory T lymphocytes, HSCs and tumor cells. I found DEGs and GSEA in Figure 5 to be rather superficial. Which genes or molecules in AML cells are critical in cell survival through interaction with Cxcl12 expressed by Nestin⁺ stromal cells? If the authors address these points, please discuss potential utility of targeting leukemic cell-stromal cell interactions as a new therapeutic option.

We thank the reviewer for this important and thoughtful comment. To evaluate the molecular basis of the exhaustion phenotype observed in L-GMPs, we re-analysis our RNA-seq data from both Dpp4^{-/-} and N-cad-CreEr; Cxcl12^{-/-} AML cells. As shown in **Response Figure 2a–c** (**new supplementary Figures 6a-b**, **new Figure 5h**), both models revealed consistent downregulation of key transcriptional regulators essential for LSC survival, including Bcl6, Fos11, Fos12, Bcl3, Bcl211, and Bcl2I11. These genes have been previously implicated in anti-apoptotic signaling, stemness, and transcriptional maintenance in HSCs, memory T cells, and leukemic cells. Together, these transcriptional changes support the interpretation of LSC exhaustion.

We think there is a typo in the comment “Which genes or molecules in AML cells are critical in cell survival through interaction with Cxcl12 expressed by Nestin+ stromal cells”, which should be N-Cadherin + stromal cells.

To deepen our molecular and signaling mechanism studies, we perform GSEA analysis of AML cells from both models revealed significant downregulation of key pro-survival signaling pathways, including IL6–JAK–STAT3, MAPK, and NFκB (**Response Figure 2d, new Figure 5j**). At the gene level, RNA-seq data showed reduced expression of critical downstream effectors such as *Bcl6* (STAT3 target, PMID: 17951530), *Fos11/Fos12* (MAPK–AP1 axis, PMID:37380804), and *Bcl3*, *Bcl2l1*, *Bcl2l11* (NFκB targets, PMID:10733571) (**Response Figure 2a–c**). These transcriptional changes were corroborated by decreased phosphorylation of STAT3, ERK1/2, p38, and NFκB p65 in AML cells from *Cxcl12*-deficient niches (**Response Figure 2e, new Figure 5k**). Collectively, these data indicate that CXCL12–CXCR4 interactions within the N-Cadherin + stromal niche support AML cell survival via coordinated activation of multiple signaling cascades. Disruption of this axis leads to transcriptional silencing and functional exhaustion of LSCs.

These findings underscore the critical role of stromal-derived CXCL12 in maintaining AML cell viability and suggest that therapeutic targeting of leukemic–stromal interactions may overcome niche-mediated resistance. We are in the procedure to test inhibition of CXCL12–CXCR4 signaling or its downstream effectors—such as with N-cadherin depletion (ADH-1, MCE), CXCR4 antagonists (AMD3100, MCE), JAK/STAT inhibitors (AG490, MCE), or MAPK/NFκB pathway blockers (NF-κB/MAPK-IN-1, MCE)—could potentiate current AML therapies. Although combination strategies were not tested in the current study, our data provide a strong mechanistic rationale for future therapeutic development in this direction.

We have revised the Results and Discussion sections accordingly, and we thank the reviewer for highlighting the need for deeper mechanistic and translational insights.

3. Bone structures are distinct between PM/DM and CM, with PM/DM having increased endosteal regions and possibility for a greater contact with hematopoietic cells. How do those locations contribute to normal HSC function? Effect of *Dpp4* targeting appeared to be larger in PM than in DM. If this is statistically significant, please describe authors’ interpretation. Please also explain what caused differences between PM and DM in multiple figure panels.

We thank the reviewer for this insightful question. The anatomical differences between PM/DM and CM regions are tightly linked to niche specialization and differential support of HSC subsets.

In this study, our primary focus was on how the bone marrow niche supports AML cells and leukemic stem cells (LSCs). In parallel, we collaborated with Dr. Linheng Li's group to investigate the spatial organization of the HSC niche in normal hematopoiesis (Dong et al., bioRxiv 2024; <https://doi.org/10.1101/2024.06.28.601225>, under revision). Growing evidence supports that distinct anatomical regions within the bone marrow provide spatially specialized microenvironments that differentially regulate HSC behavior. The PM/DM region, enriched in trabecular bone and endosteal surfaces, acts as a "reserve" niche that promotes HSC quiescence

and long-term function. Dong et al., using label-retention assays and transplantation analyses demonstrated that deep-quiescent, functionally validated HSPCs preferentially localize to the PM/DM region (**Response Figure 3**), where they are supported by N-cadherin⁺ stromal cells secreting high levels of CXCL12 and other quiescence-enforcing factors. To directly test how niche-derived CXCL12 governs HSC localization and behavior, the study employed bone marrow transplantation into wild-type or N-cadherin-CreEr;Cxcl12^{fl/fl} conditional knockout (cKO) hosts. In wild-type recipients, 52.98% ± 8.26% of homed cells localized to the trabecular bone area (TBA, corresponding to the PM/DM region), while only 13.62% ± 4.10% resided in the central marrow (CM). In contrast, CXCL12-deficient hosts exhibited a marked redistribution, with donor cells in the CM increasing to 35.10%, accompanied by reduced TBA localization. Flow cytometry further

revealed that the LSK (Lin⁻Sca1⁺c-Kit⁺) population expanded significantly that coincided with enriched HSPCs in CM of cKO mice (9.49% vs. 3.25% in WT), and cell cycle analysis showed a shift from quiescence to proliferation: the proportion of LSK cells in G₀ dropped from 59.2% to 38.7%, while those in G₂/S/M increased from 7.7% to 25.4%. These findings demonstrate that CXCL12 derived from N-cadherin⁺ stromal cells are essential for attracting HSCs in the quiescent PM/DM niche. Disruption of this signal leads to spatial redistribution into the CM region that support relatively active HSPCs. Together, these results define the PM/DM region as a specialized niche that maintains long-term, dormant HSCs, while the CM supports HSC expansion—highlighting the importance of spatially compartmentalized microenvironments in regulating normal hematopoiesis.

We appreciate the reviewer's observation. We agree that the effect of DPP4 targeting is more pronounced in the PM than in the DM, an effect we have also observed in both normal hematopoietic and leukemic contexts. Further studies are needed to reveal the underlying mechanism. Our explanation is that PM is uniquely enriched for N-cadherin⁺ stromal cells than DM. GPC3, an endogenous DPP4 inhibitor also highly expressed on N-cadherin⁺ stromal cells. GPC3 blocks local DPP4-mediated cleavage of CXCL12—preserving a steep chemokine gradient. When DPP4 is inhibited, driving the pronounced PM response, whereas the DM niche—with fewer N-cadherin⁺ cells and lower GPC3—maintains a shallower baseline gradient that Dpp4 loss perturbs to a lesser extent. A secondary, non-mutually exclusive factor that may contribute to the observed differences is the technical challenge of sample acquisition from distinct anatomical compartments of the long bone. The complex anatomical structure of the DM makes it more difficult to obtain a complete and intact sample compared to the more accessible PM. Consequently, the incomplete harvesting of the DM niche may lead to the unintended loss of a subpopulation of cells, potentially including the critical N-cadherin⁺ stromal cells that are central to the proposed mechanism. This sampling bias could result in an underrepresentation of the full cellular composition of the DM, thereby influencing the observed response to DPP4 inhibition.

4. Clinically, MLL/KMT2A leukemia exhibit aggressive clinical course with abundant tumor cells both in BM and in circulation. Dpp4KO provides a biologically intriguing phenomenon in two AML mouse models. Discussion of how these models recapitulate patient leukemia and contribute to clinical translation by targeting DPP4-GPC3 interaction would strengthen this point.

We appreciate the reviewer's insightful comments and fully agree that emphasizing the clinical relevance of our models and findings strengthens the translational impact of this study.

The following paragraphs have been added to the Discussion section:

“Our results also have important clinical implications. The MLL-AF9 and AML-ETO9a mouse models recapitulate aggressive human AML subtypes (KMT2A-rearranged and t(8;21)), which are associated with high leukemic burden and poor prognosis. In both models, genetic ablation of Dpp4 confined AML cells within the BM, depleted LSCs from protective metaphyseal niches, and reduced peripheral dissemination—a clinically relevant effect given that leukocytosis and organ infiltration are major drivers of AML morbidity. Thus, targeting the DPP4–CXCL12–GPC3 axis could impair niche-mediated LSC survival while also mitigating systemic complications. This axis offers immediate translational promise. DPP4 inhibitors (e.g., sitagliptin, linagliptin), already FDA-approved for diabetes^{13,54}, could be repurposed to: (1) reverse BM-PB CXCL12 gradients and confine AML cells, reducing complications of hyperleukocytosis^{55,56}; (2) dislodge LSCs from protective niches to sensitize them to chemotherapy; and (3) induce LSC exhaustion. Building on these mechanistic insights, we have initiated preclinical testing of combination strategies—CXCR4 antagonists (e.g., AMD3100)⁵⁷, NF-κB inhibitors (e.g., QNZ)⁵⁸, and N-cadherin antagonist (ADH-1)⁵⁹—to disrupt stromal anchoring, survival pathways, and chemokine gradients in parallel.”

5. scRNAseq demonstrated molecular interaction between the niche and AML. Please provide detailed information regarding cell population annotation in scRNAseq analysis. List of genes used to annotate each cell population and heat map representation of those genes in each defined cell population would be helpful. For instance, how was the MSC cluster defined? Which genes were chosen as the most characteristic genes in MSCs alongside CXCL12?

We thank the reviewer for this important suggestion. To clarify our scRNA-seq analysis and improve transparency in cell type annotation, we have revised the Results and method section to include more detailed information on the marker genes used to define each non-hematopoietic

niche population, including mesenchymal subtypes. In addition, we have incorporated a marker gene heatmap (**Response figure 4; new Supplementary Figure 3**) to illustrate the expression patterns across all annotated clusters.

To identify the specific niche cells secreting CXCL12, we performed single-cell RNA sequencing (scRNA-seq) on non-hematopoietic PM and DM cells from AML mice three weeks post-transplantation. Through the analysis of 11,512 cells (median of 19,367.5 molecules and 4,391 genes per cell), we identified 14 distinct clusters, spanning mesenchymal stem/stromal cells (MSC, marked by *Lepr*, *Cdh2*, and *Cxcl12*^{16,30,31}), osteolineage cells (OLC, marked by *Bglap*), chondrocytes (*Chondro*) and chondrocytes of possible transitional states (*c/r*, cycling/resting; *pro*, progenitor, marked by *Acan* and *Col2a1*), fibroblasts (*Fibro*, marked by *S1004a*), endothelial cells (*EC*, marked by *Cdh5*), pericytes (marked by *Acta2*) (**Fig. 4c; Supplementary Fig. 3**). Consistent with prior reports, *Cxcl12* was predominantly expressed by the MSC cluster (**Fig. 4d**)¹. MSCs are

heterogeneously characterized by several reported markers, such as *LepR*⁺, *N-cad*⁺ (*Cdh2*), *Prx-1*⁺, *Osx*⁺ (*Sp7*), and *Nes*⁺ MSCs^{16,31-36}. To identify the major source of CXCL12 in the PM/DM, we compared the association of *Cxcl12* with MSCs transcriptome and found *Cdh2* and *Lepr* had the highest association with *Cxcl12* expression among the aforementioned MSC sub-populations (**Fig. 4e**). *LepR*⁺ MSCs are known as a heterogenous population of MSCs spanning across the BM, including metaphysis (but mainly perivascular localization in metaphysis³¹) and CM region. The *N-cad* protein is predominantly detected in the endosteum of metaphysis¹⁷. Other MSC subpopulations, such as *Prx-1*⁺, *Osx*⁺, and *Nes*⁺ MSCs, are minimally found within our study population (**Fig. 4f**), which is consistent with prior studies that *Prx-1*⁺ MSCs enriched in periosteal region^{37,38}, *Osx*⁺ enrich in the bone tissue³⁹, and *Nes*⁺ enriched in perivascular region³⁶.

Critically, despite prior reports implicating *Nestin*⁺ MSCs in AML chemoresistance⁴⁰ their *Cxcl12* expression was negligible (**Fig. 4f**). To rule out functional redundancy, we deleted *Cxcl12* in *Nestin*⁺ cells (*Nestin-CreER*; *Cxcl12*^{*fl/fl*}), and observed no impact on AML progression or BM retention (**Supplementary Fig. 4a-b**), confirming *Nestin*⁺ MSCs are dispensable for CXCL12-mediated LSC maintenance in our model.

N-cad⁺ MSCs co-expressed *Cxcl12* with other LSC-supportive factors (e.g., *Gas6*, *Angpt1*, *Kitl*; **Fig. 4g**)^{7,41-43}.

We also updated "Single cell RNA-seq analysis" in Method section.

6. Kaplan-Meier curve presented in Figure 5h strongly supports the authors' claim. Since few AML cells were detected in PB, I presume leukemic infiltration in spleen, liver or other organs were also scant. When leukemic cells disturb normal hematopoiesis in BM, extra-medullary hematopoiesis frequently occurs in spleen. Please show histology of DPP4-wt and DPP4-/- spleen and distribution of normal and malignant blood cells in the organs. CBC (WBC, Hb, platelets) of the two mice would also be very helpful to understand discrepant distribution of AML cells in BM and PB in the model.

We thank the reviewer for this important comment. In response, we have performed histological analyses of spleen tissue from both DPP4-WT and DPP4-KO AML mice (**Response Figure 5, supplementary Figure 2c-d**). In the WT AML spleen (left panel), normal splenic architecture is

almost completely effaced by a diffuse, sheet-like infiltrate of malignant blasts. These blasts are medium-sized, with high nuclear-to-cytoplasmic ratios, finely vesicular chromatin, and conspicuous nucleoli. They permeate both red pulp cords and white pulp follicles, displacing virtually all mature erythrocytes and lymphoid structures. By contrast, the DPP4KO AML spleen (right panel) shows markedly reduced blast infiltration. Residual red pulp sinusoids still contain abundant mature erythrocytes and scattered granulocytes, and periarteriolar lymphoid sheaths and follicles in the white pulp are partially preserved. Malignant blasts are present only in focal clusters rather than confluent sheets, indicating a lower leukemic burden and partial restoration of normal splenic architecture.

Additionally, we have now included complete blood counts (CBC) data on both non-transplanted and AML-transplanted cohorts (n=5 per group). As shown in the **Response Figure 6 (supplementary Figure 2c-d)**, complete blood counts in non-transplanted mice confirmed that *Dpp4^{+/+}* and *Dpp4^{-/-}* animals have virtually identical baseline hematology (Hb \approx 12.4 g/dl, WBC

	Mice without AML transplant		Mice with AML cell transplant	
	DPP4 ^{+/+} (n=5, mean \pm SEM)	DPP4 ^{-/-} (n=5, mean \pm SEM)	DPP4 ^{+/+} (n=5, mean \pm SEM, BM YFP ⁺ 60%)	DPP4 ^{-/-} (n=5, mean \pm SEM, BM YFP ⁺ 60%)
Hemoglobin (g/dl)	12.3 \pm 0.3	12.5 \pm 0.4	8.5 \pm 0.9	12.5 \pm 0.9
Total leukocyte (10 ³ / μ l)	10.1 \pm 0.9	11.5 \pm 1.1	23.1 \pm 3.5	10.6 \pm 1.4
Platelets (10 ³ / μ l)	669 \pm 55	679 \pm 65	40 \pm 21	692 \pm 185

Response Figure 6. Table of complete blood count (CBC) of DPP4^{+/+} and DPP4^{-/-} AML cells transplanted mice (4-week and 13-week post-transplantation, respectively), at time points with similar leukemia cell percentages (\sim 60%) in the BM

\approx 10–11 \times 10³/ μ l, platelets \approx 675 \times 10³/ μ l). Following AML cell engraftment, *Dpp4^{+/+}* recipients developed profound anemia (Hb 8.5 \pm 0.9 g/dl), marked leukocytosis (WBC 23.1 \pm 3.5 \times 10³/ μ l) and severe thrombocytopenia (PLT 40 \pm 21 \times 10³/ μ l), consistent with high leukemic burden in bone marrow and blood. In contrast, *Dpp4^{-/-}* AML-transplanted mice maintained normal Hb (12.5 \pm 0.9 g/dl), WBC (10.6 \pm 1.4 \times 10³/ μ l) and platelet (692 \pm 185 \times 10³/ μ l) counts indistinguishable from non-transplanted controls, demonstrating that DPP4 deficiency limits AML dissemination into the periphery and preserves steady-state hematopoiesis. These data suggest that DPP4 deletion mitigates systemic leukemic burden and helps maintain hematopoietic output. We appreciate the reviewer’s insightful suggestion, which allowed us to further validate the systemic impact of DPP4 targeting in AML.

7. How do normal HSCs and L-GMPs share DPP4-GPC3 axis and how does CXCL12 gradients affect homeostasis of normal hematopoietic stem/progenitor cells?

We thank the reviewer for this important question. We agree that the coordinated regulation of both normal HSCs and LSCs by the DPP4–GPC3–CXCL12 axis is a critical area for investigation. While the primary focus of our study is on LSCs, our flow cytometry analyses of GFP⁻ (normal

Response Figure 7. Distribution of HSPCs and LSPCs in metaphyseal (PM) and central marrow (CM) under different AML burdens.

hematopoietic cells) and GFP⁺ (AML cells) together with lineage phenotypic markers within anatomically separated PM and CM regions in AML-engrafted mice, provides insight into this shared regulatory mechanism (**Response Figure 7**). Our data show a clear spatial segregation between normal and leukemic progenitor cells: Under **low AML burden (typical used in our image studies)**, hematopoietic progenitor cells (HSPCs) are predominantly located in the CM, while leukemic progenitor cells (LSPCs) are enriched in the metaphysis. Under **high AML burden**, LSPCs become preferentially retained in the metaphysis, presumably outcompeting the severely depleted HSPCs for protective stromal signals. This spatial shift supports the idea that the DPP4–GPC3–CXCL12 axis is a shared regulatory mechanism that LSCs exploit to facilitate their retention and survival. These insights provide further rationale for targeting this pathway to restore normal niche function and suppress leukemic stem cell maintenance.

Reviewer #2 (Remarks to the Author):

In this manuscript by Wang et al. the authors show that the distribution of AML cells in the bone marrow influences biological behavior of leukemia cells. Specifically, the authors highlight the influence of location in longitudinal niches of the bone marrow on leukemic stem cell (LSC) survival and proliferation. LSC seem to localize to the metaphysis, showing reduced stemness and aggressiveness when mobilized to the central marrow region. The authors deleted DPP4 in AML cells altered the CXCL12 gradient within the bone marrow niche. DPP4 is thought to deactivate CXCL12, while GPC3, found in N-cad⁺ cells, inhibits DPP4. The manuscript lacks clarity in writing and the data are not convincing. The manuscript also makes several overstatements and is inaccurate in several places. The aspect of niche location correlating with aggressivity is not novel (see Pereira et al., Kumar et al.).

We sincerely thank the reviewer for the critical feedback, which has helped us improve the clarity and rigor of the manuscript.

Regarding novelty, we fully acknowledge prior studies (e.g., Pereira et al.; Kumar et al.) that demonstrated spatial heterogeneity in AML localization and the influence of bone marrow niches on leukemic behavior. Building upon these important foundations, our study introduces a previously unrecognized longitudinal axis of LSC organization within the bone marrow — between the metaphysis and central marrow.

Our key novel contributions are:

1. Longitudinal regulation of LSCs: While prior studies mainly examined *transverse spatial dimensions* (e.g., endosteal vs. sinusoidal/diaphyseal niches), we show that *longitudinal positioning* — metaphysis versus central marrow — fundamentally influences LSC behavior.
2. Distinct functional states: We demonstrate that LSCs in the metaphysis remain quiescent and stem-like, whereas displacement into the central marrow induces cycling, loss of stemness, and increased apoptosis.
3. We identify a new regulatory axis involving DPP4 (AML-derived), CXCL12 (niche chemokine), and GPC3 (from N-cadherin⁺ MSCs) with genetic validation.
4. Therapeutic relevance: By mobilizing LSCs from quiescent metaphyseal regions into more vulnerable states, it could enhance chemosensitivity and prevent relapse. Importantly, this can be achieved using FDA-approved DPP4 inhibitors, providing immediate translational potential.

In summary, while prior work established the importance of BM niches, our study extends this knowledge by uncovering a *longitudinal spatial mechanism* of LSC regulation and identifying a novel DPP4–CXCL12–GPC3 axis as a therapeutic target. We have revised the manuscript to clarify these points, avoid overstatements, and explicitly distinguish our findings from earlier studies.

Major points are:

1. The exact mechanism of how DPP4 deletion alters the CXCL12 gradient remains elusive. Why would deletion of DPP4 in AML cells, which presumably infiltrate the entire BM niche, lead to such regional differences in levels of CXCL12? What happens to the CXCL12 gradient in other organs?

We thank the reviewer for raising this important mechanistic question. We agree that the non-

Response Figure 8. Representative immunofluorescence staining of liver tissue. The image depicts CXCL12 (red) and nuclei (DAPI, blue). Yellow arrowheads indicate regions of high CXCL12⁺ expression within the perivascular niche.

uniform CXCL12 levels were an unexpected finding, as the DPP4 deletion is systemic. Our results indicate that the regional differences are not due to localized deletion, but rather the result of two interacting mechanisms: **niche-specific CXCL12 production** and **spatially localized DPP4 activity**. While the DPP4 deletion is systemic, its effect is most pronounced in the CM, where DPP4 activity is normally highest. Prior studies (PMID: 23434755, 23434756) have shown that Tek⁺ endothelial cells and Prx1⁺ mesenchymal progenitors, predominantly located in the CM, are the primary sources of CXCL12 in steady-state hematopoiesis. Our scRNA-seq data reveal that Cdh2⁺ (N-cadherin⁺) and Lepr⁺ MSCs are the main CXCL12-producing populations in AML BM. These populations are spatially distributed: Cdh2⁺ MSCs more abundant in the metaphysis, and Lepr⁺ MSCs are known as a heterogeneous population of MSCs spanning across the BM, including perivascular localization in metaphysis and CM regions. We hypothesize that in the CM, intrinsic higher CXCL12 production by ECs and Lepr⁺ MSCs is normally balanced by high local DPP4 activity. Upon DPP4 deletion, this degradation mechanism is lost, unmasking the locally produced CXCL12 and leading to its

selective accumulation in the CM. Conversely, in the metaphysis, GPC3 expression in N-cadherin⁺ MSCs normally inhibits local DPP4 activity, thus the effect of DPP4 deletion is less dramatic in this region. Ultimately, the loss of this spatially restricted degradation reverses the pathological gradient, confining LSCs and limiting AML dissemination.

To investigate whether the CXCL12 gradient observed in the bone marrow is preserved in other tissues, we performed immunofluorescence staining for CXCL12 protein in liver sections from AML-bearing mice. As shown in **Response Figure 8**, CXCL12 (red) appeared enriched in the portal vein and along sinusoidal structures. Our collaborator Linheng Li also have similar observations (<https://doi.org/10.1101/2024.06.28.601225>).

2. The statements in the introduction that 'no effective treatments have been developed to restrict leukemia dissemination or eradicate LSC' is very simplified and partially inaccurate. In addition, many publications have studied the bidirectional interactions between leukemia cells and their

niche. Hence reference 3 is only of limited value. A more objective and comprehensive view on the published studies is necessary.

We appreciate the reviewer's important and constructive feedback. To address the first point, we have revised the sentence to state: "Although numerous studies have delineated LSC–niche interactions and inspired translational strategies, clinically validated approaches that reliably restrict leukemia dissemination or durably eradicate leukemic stem cells (LSCs) remain limited." This revised wording acknowledges the significant progress in the field while accurately highlighting the persistent translational challenges.

3. Lines 74-77 and corresponding Figures 1-b. The sentences do not describe what is supposed to be seen in the figures. The abbreviations in the figure are not spelled out.

We appreciate the reviewer's careful attention to detail, which has significantly improved the clarity and accessibility of our manuscript.

We have revised the manuscript text to more clearly describe the findings shown in Figure 1a-b. The new text now emphasizes that while $Dpp4^{+/+}$ and $Dpp4^{-/-}$ AML-transplanted mice have comparable overall AML burden, the $Dpp4^{-/-}$ group shows a significant difference in spatial distribution, with GFP⁺ AML cells being enriched in the proximal metaphysis (PM) region.

We have rewritten the figure legends to explicitly define all abbreviations, including PM (proximal metaphysis), CM (central marrow), DM (distal metaphysis), L-GMP (leukemia granulocyte-

similar timepoints in xenografts.

Nonetheless, to directly address the reviewer's concern, we performed an additional 20 h homing assay and analyzed GFP⁺ leukemic cells in bone marrow (BM), peripheral blood (PB), spleen, and liver. As shown in **Response Figure 9**, no significant differences in GFP⁺ cell distribution were observed between $Dpp4^{+/+}$ and $Dpp4^{-/-}$ mice across all compartments, confirming that our originally used 16 h timepoint reliably captured homing dynamics without missing relevant differences.

monocyte progenitor), and MSC (mesenchymal stromal cell). We have also added details on how the data were quantified to ensure consistency and improve interpretability.

4. A homing experiment of DPP4 KO cells after 18 hours is necessary.

We appreciate the reviewer's suggestion to include a later timepoint for the homing assay. As noted, "homing" traditionally refers to the initial lodging of hematopoietic cells into bone marrow niches, encompassing chemokine-guided migration, vascular adhesion, transmigration, and niche arrest—prior to any significant cell proliferation. To avoid confounding by early proliferation or engraftment signals, most homing assays are conducted within a 4–16-hour window. For example, Christopherson et al. evaluated murine HSPC homing at 16 h, while Lapidot et al. assessed human CD34⁺ cell homing at

Therefore, while our initial 16 h assay was methodologically sound and standard, this additional 20 h dataset further supports our conclusion that DPP4 deletion does not impair the initial homing of AML cells but rather affects their post-homing retention, spatial positioning, and expansion, which are assessed in our long-term localization studies.

5. Please put your manuscript in relation to PMID: 27750279 and PMID: 29276143.

We thank the reviewer for drawing attention to these seminal studies. Hawkins et al. (Nature 2016; PMID: 27750279) used intravital imaging to show that T-ALL blasts infiltrate the bone marrow in a stochastic and highly motile manner, broadly displacing normal stromal niches and impairing hematopoiesis through physical competition and niche exhaustion. Duarte et al. (Cell Stem Cell 2017; PMID: 29276143) further revealed that AML progression remodels the endosteal vascular niche, secondarily impairing HSC maintenance by disrupting vascular integrity. These landmark works collectively established that leukemias can either broadly exploit or destructively remodel the marrow microenvironment.

Building on this foundation, our study provides a distinct and complementary perspective by uncovering a molecularly defined mechanism of AML niche remodeling. Specifically, we show that AML cells enzymatically degrade CXCL12 through DPP4 activity, thereby reshaping chemokine gradients to favor leukemic stem cell (LSC) retention and displacement of normal HSCs. Importantly, we identify GPC3⁺ N-cadherin⁺ stromal cells in the metaphyseal endosteal zone as a protective stromal subset that counteracts this degradation, preserving local CXCL12 gradients and anchoring LSCs. Unlike the physical competition described in T-ALL or the vascular remodeling observed in AML, our findings highlight an orthogonal, chemokine-centric mode of niche regulation that is both mechanistically distinct and readily targetable through DPP4 inhibition.

Thus, while Hawkins and Duarte defined the spatial and vascular dimensions of leukemic niche disruption, our work advances the field by providing a molecular axis (DPP4–GPC3–CXCL12) that explains how AML dynamically sculpts its microenvironment. This not only deepens mechanistic understanding but also introduces a translationally actionable strategy for therapeutically dislodging LSCs from their protective niches.

We have added discussion of both studies (Hawkins et al., 2016; Duarte et al., 2018) in the revised Discussion section to clarify how our mechanistic insights build on and go beyond these earlier works.

“This chemokine-centric mechanism complements structural models of leukemic niche disruption. For example, Hawkins et al. showed that T-ALL cells exhaust niches by physical displacement⁵³, and Duarte et al. reported that AML remodels vascular niches⁶. Our findings add a distinct molecular axis—DPP4-mediated CXCL12 degradation counterbalanced by stromal GPC3—that governs AML niche remodeling.”

6. Figures 2a-b: How can the authors be sure to have isolated the LSC only from the CM, PM and DM? How was this performed?

We appreciate the reviewer’s important question regarding the anatomical fidelity of our niche-specific L-GMP isolation. To ensure spatial accuracy, we followed a rigorously standardized dissection protocol based on consistent anatomical landmarks. Specifically, femurs and tibias were segmented into three reproducible regions: PM, the segment extending approximately ≈3

mm distally from the proximal growth plate; DM, the region extending approximately ≈ 3 mm proximally from the distal growth plate; CM, the mid-diaphyseal shaft between the two metaphyseal regions. Each region was flushed independently with separate syringes to avoid cross-contamination. Subsequently, cells from each compartment were subjected to identical flow cytometric gating to isolate L-GMPs, ensuring technical consistency across samples.

While we acknowledge the possibility of minor anatomical overlap at compartment boundaries, the pronounced differences observed in downstream functional and transcriptional assays (e.g., survival outcomes, stemness signatures) strongly support the biological distinctiveness of each niche. To improve transparency, we have now clarified these technical details in the revised Methods section.

7. A key experiment would be to delete CXCL12 specifically in the CM, PM and DM regions. Have the authors attempted an experiment along these lines? It is appreciated that this is difficult.

We appreciate the reviewer's thoughtful suggestion. We agree that a regional deletion of CXCL12 would be a key experiment to directly test the functional role of the CXCL12 gradient. However, we have not attempted this experiment because it is currently technically difficult. The primary challenge is the lack of specific genetic tools or Cre drivers that can be used to selectively delete CXCL12 in the different anatomical regions of the bone marrow (CM, PM, and DM). Instead, we used a combination of high-resolution spatial imaging, chemokine quantification to demonstrate that the CXCL12 gradient is critical for regulating leukemic cell localization and function. These approaches, while indirect, provide strong evidence for the role of the DPP4–GPC3–CXCL12 axis in controlling LSC behavior.

8. Line 253: Clotting abnormalities and stroke are usually not complications of AML (apart from APML).

We thank the reviewer for this important point. You are correct that frank stroke is rare in non-APL AML patients and is most classically associated with acute promyelocytic leukemia (APL). However, AML more broadly—particularly in the context of hyperleukocytosis—is well documented to induce thrombo-hemorrhagic complications, including leukocytosis, microvascular occlusion, and disseminated intravascular coagulation (DIC). These pathophysiological events can lead to peripheral thromboses, bleeding, and end-organ ischemia, though they may not always manifest as overt cerebrovascular stroke.

To clarify this distinction, we have revised the manuscript text as follows:

“...and reduced peripheral dissemination—a clinically relevant effect given that leukocytosis and organ infiltration are major drivers of AML morbidity. Thus, targeting the DPP4–CXCL12–GPC3 axis may both impair niche-mediated LSC survival and mitigate systemic complications.”

In support of this, our murine AML models (e.g., MLL-AF9 and AML-ETO9a driven) consistently recapitulate pro-thrombotic features in vivo. We frequently observe intestinal bloating, consistent with mesenteric vascular compromise, and unilateral limb paralysis, suggestive of CNS microvascular occlusion. These phenotypes occur in animals with high leukemic burden and often precede signs of respiratory distress or systemic decline. While such presentations are less commonly reported in human AML outside of APL, they are well-aligned with the known vaso-occlusive potential of circulating blasts and highlight the relevance of clotting pathology in high-burden leukemias.

Our revised language aims to balance clinical accuracy with biological plausibility—acknowledging that AML can drive peripheral micro thrombotic and ischemic events, without overstating the frequency of frank stroke in non-APL cases.

9. Leukemia cells should be labelled by GFP and Gr1, e.g. Figure 1a-b.

We are grateful to the reviewer for suggesting the co-labeling of AML blasts using GFP and Gr-1. Following this recommendation, we performed flow cytometry to validate GFP and Gr-1 co-expression in bone marrow samples from five independent *Dpp4*^{+/+} or *Dpp4*^{-/-} AML mice (**Response Figure 10**). After gating on live single cells, GFP⁺ cells accounted for 58.7% of bone marrow events in *Dpp4*^{+/+} mice and 36.6% in *Dpp4*^{-/-} mice (left panels). Further analysis of these GFP⁺ populations for myeloid marker expression revealed that in *Dpp4*^{+/+} AML mice, 94.2% of GFP⁺ cells co-expressed Mac-1 and Gr-1, with an additional 0.5% positive only for Gr-1, resulting in a total of 94.7% Gr-1 positivity (right, top). In *Dpp4*^{-/-} AML mice, 88.5% of GFP⁺ cells were Mac-1⁺ Gr-1⁺, and 0.48% were Mac-1⁻ Gr-1⁺, yielding an overall Gr-1 positivity rate of 88.98% (right, bottom).

These findings demonstrate that in both genotypes, nearly all GFP⁺ AML cells are Gr-1⁺ myeloid blasts, confirming that GFP serves as a faithful marker for leukemia cells in this model.

10. Figure 1: What images represent 2 weeks, which 12 weeks?

We appreciate the reviewer's attention to this detail. In Figure 1, the images for wild-type (WT) AML represent mice at 2 weeks post-transplantation, while the *Dpp4*-KO AML images represent mice at 12 weeks. These time points were chosen to ensure comparable leukemic burden between groups, as *Dpp4* deletion significantly delays disease progression. Thus, both sets of images represent mice with similar disease severity at different time points. We have now clarified this explicitly in the figure legend section to prevent confusion.

11. Figure 1a legend: primary AML mouse models: Do you mean recipients of AML cells? (explain 1st AML model...)

We thank the reviewer for pointing this out. The term "primary AML mouse models" in the figure legend was indeed imprecise. We were referring to recipient mice that received transplantation of primary AML cells, rather than de novo AML models. We have revised the legend of Figure 1a to state "Representative images of BM sections from recipients transplanted with *Dpp4*^{+/+} (left) or *Dpp4*^{-/-} (right) primary AML cells" to avoid confusion and improve accuracy.

12. Figure 1d: Bone sections need to be collected on the same day for recipients of WT and KO cells!

We thank the reviewer for raising this important point. In Figure 1d, bone sections were collected at different time points (2 weeks for WT and 12 weeks for *Dpp4*-KO), reflecting the markedly different disease kinetics between groups. However, these time points were carefully selected to ensure comparable leukemic burden at the time of tissue harvest. This approach allowed us to evaluate the effect of *Dpp4* deletion on bone marrow architecture under equivalent disease conditions.

13. Figure 1g: The authors mean WT versus KO L-GMP cells, not mice, correct? Please be accurate.

We thank the reviewer for this important clarification.

Yes, Figure 1g (now Figure 1i-j) compares recipients transplanted with WT versus Dpp4-KO L-GMP cells, not mice with different germline genotypes. We apologize for the ambiguous wording, and we have now corrected the terminology in the figure legend and main text to clearly state that the comparison refers to the genotype of the transplanted leukemic cells, not the recipient mice. We appreciate the reviewer's careful reading.

14. Fig. 2 e: It is hard to believe that the survival of mice is 50 days after transplantation of 500 L-GMP and that the p value for these survival curves is <0.0001.

We thank the reviewer for raising this important point regarding the survival duration and statistical analysis in Fig. 2e.

1. Clarification of observed survival durations

We acknowledge that the ~40–55 day survival observed in our model may appear longer than expected for AML induced by L-GMP transplantation. However, several biological and experimental factors contribute to this disease latency:

We utilized secondarily transduced L-GMPs, which are known to exhibit reduced stemness and leukemogenic potential relative to freshly transduced or primary blasts.

To minimize disruption of the bone marrow microenvironment, we used 500 cGy of X-ray irradiation, a sublethal dose designed to reduce host niche damage while allowing engraftment.

Our study builds directly upon the established AML transplant framework described in previous study (**Response Figure 1a**, PMID: 16862118), which similarly reported disease progression over multiple weeks under comparable conditions.

These design choices were intentional to enable robust analysis of spatial dynamics and niche influence on leukemic behavior under physiologically relevant conditions.

2. Re-analysis of survival data

We re-ran the analysis in GraphPad Prism using the Mantel–Cox test. For the three-group comparison in Fig. 2e (n = 10 each), the correct statistics are:

- $\chi^2 = 7.835$, $df = 2$, $P = 0.0199$

Our originally reported “P < 0.0001” resulted from a transcription error and has now been corrected. (**Response Figure 11, Figure 2e**)

To further evaluate the influence of bone marrow niche origin on AML aggressiveness, we conducted unadjusted pairwise log-rank (Mantel–Cox) tests between the three groups. These analyses demonstrated that PM-derived AML resulted in significantly worse survival than CM-derived AML (P = 0.0129), while DM-derived AML also conferred a survival disadvantage compared to CM (P = 0.0353). In contrast, no significant difference was observed between PM and DM groups (P = 0.4177). These results suggest a spatial hierarchy of leukemic stem cell aggressiveness, with metaphyseal (PM) and diaphyseal (DM) niches supporting more aggressive disease phenotypes, whereas central marrow (CM) niches are associated with reduced stemness and improved survival outcomes.

15. Fig. 3a-b: Please compare WT versus KO, not different organs within one cohort.

We thank the reviewer for this helpful suggestion. We have revised Figure 3b, which now shows that WT and Dpp4 KO AML cells have similar homing ability to different organs (**Response Figure 12 a-b, new Figure 3b**).

Regarding Figure 3a, our intention is to show the overall AML cell distribution among organs at the peak of disease progression. This figure illustrates a key difference between the two groups. In the WT AML model, AML cells disseminate to and infiltrate various hematopoietic organs. In contrast, in the Dpp4 KO mice, **DPP4 deficiency confines AML cells primarily within the bone marrow**, preventing their trafficking to other organs such as the blood, spleen, and liver. We

have included these comparisons among organs within the same group to highlight the confinement effect of DPP4 deletion.

16. Fig. 3a-b: Please compare Lin⁻ GFP⁺ cells, not just GFP⁺ cells.

We agree that focusing on Lin⁻ GFP⁺ cells provide a more accurate representation of undifferentiated leukemia cells with potential stem-like properties.

In response, we have reanalyzed and re-plotted the data in Figure 3a-b, now specifically comparing Lin⁻ GFP⁺ cells between WT and Dpp4-KO AML groups across bone marrow, spleen, and liver (**Response Figure 13**). We have also updated the figure legend and Methods section to reflect the gating strategy used in the revised analysis.

This change strengthens our conclusions, and we are grateful to the reviewer for the suggestion.

17. Fig. 3g-h: How do the authors explain the discrepant results of CXCL12 levels in BM versus plasma of recipients of WT versus KO AML cells?

We appreciate the reviewer's insightful question regarding the disparity between CXCL12 levels in the bone marrow (BM) and plasma of AML recipients. This observation highlights a fundamental distinction between the localized, niche-dependent regulation of CXCL12 and its systemic, multi-factorial control.

Response Figure 14.
CXCL12 levels in plasma of *Dpp4*^{+/+} and *DPP4*^{-/-} mice.

In the bone marrow microenvironment, Dpp4 acts as a key local regulator of CXCL12 activity. Our data demonstrate that the absence of Dpp4, particularly on leukemic cells, prevents the proteolytic cleavage of CXCL12, thereby preserving a steep chemokine gradient within the tumor niche. This local preservation of CXCL12 in the *Dpp4*^{-/-} AML group directly supports the high BM concentrations observed in our study.

In contrast, systemic CXCL12 levels are regulated by a complex interplay of production and degradation pathways, of which DPP4 is only one of many contributors. The primary factor driving systemic CXCL12 concentration in our model is the overall tumor burden. In recipients of *Dpp4*^{+/+} AML cells, aggressive disease progression results in a high tumor burden, leading to an overwhelming

systemic production of CXCL12 that exceeds the clearance capacity of all systemic proteases. Conversely, the *Dpp4*^{-/-} AML group exhibits significantly suppressed disease and a much lower tumor burden, which in turn leads to a lower total systemic production of CXCL12. Therefore, the lower plasma CXCL12 level in this group is not a direct consequence of systemic Dpp4 deficiency but rather a secondary effect of their reduced tumor burden. We have confirmed this interpretation by demonstrating that in the absence of AML, there is no significant difference in plasma CXCL12 levels between *Dpp4*^{+/+} and *DPP4*^{-/-} mice (**Response Figure 14**), indicating that other systemic proteases are sufficient to maintain homeostatic CXCL12 levels.

18. Please show the levels of SCF in different parts of the bone.

We thank the reviewer for this valuable suggestion. We have now analyzed SCF (Kitl) expression in different anatomical regions of the bone marrow (PM, CM, and DM) using Mouse Mini Samples Stem Cell Factor ELISA Kit (SCF) by Abclonal on regionally dissected marrow samples.

(Response Figure 15; new supplementary Figure 5f)

Response Figure 15. SCF concentrations were measured in the proximal metaphysis (PM), central marrow (CM), and distal metaphysis (DM) regions. (mean ± SEM, n = 3)

Minor points:

1. English syntax and grammar need to be corrected.
2. Figure 1c: Spelling of cells.
3. Please spell out abbreviations, e.g. TIC etc.
4. Fig 3k: What is the label on the x-axis?

We thank the reviewer for these helpful comments. We have made the following corrections throughout the revised manuscript:

1. English syntax and grammar have been carefully reviewed and corrected.
2. In Figure 1c, the spelling of “cells” has been fixed.
3. All abbreviations (e.g., TIC → tumor-initiating cell; LSCs → leukemia stem cells) are now spelled out at first use.
4. In Figure 3k, the x-axis label, indicating the concentration of CXCL12 (ng), has been added.

Longitudinal Localization of Leukemia Stem Cells Between Metaphysis and Central Marrow Governs Leukemic Stem Cell Behavior

Response to the Editor and the Reviewers

We thank the Editor and the Reviewers for their thorough and insightful feedback on our manuscript. We have carefully revised the text and performed additional experiments as requested, which have substantially strengthened our study. All supplementary materials have been reorganized as Extended Data figures and formatted to journal specifications. Figure 8 now moved to Extended Data. We have added a full Data Availability statement, deposited all genome-wide datasets to GEO, and provided unprocessed blots and complete numerical source data as required. A summary of our key revisions is provided below, followed by detailed point-by-point responses.

Reviewers' Comments:

Reviewer #1 (Remarks to the Author):

In the revised manuscript, interaction through DPP4, GPC3 and CXCL12 was more convincingly proven by robust *in vivo* experiments and data analyses with a state-of-art technique. As I made comments in the first round of review, to what degree findings obtained from the two mouse models the authors used (MLL-AF9 and AML-ETO) are helpful in understanding and targeting human leukemia with more heterogeneity and complexity has been a critical key to assess impact and quality of the paper.

I thank the authors for providing me with information on a report showing consistently high expression of DPP4 in CD34⁺ human AML cells. CD34⁽⁻⁾ LICs are present in some patients. Rather than relevance between DPP4 expression and leukemia initiation, I want the authors to touch upon heterogenous leukemia in patients and to what extent the dynamics between LSCs and niche can be applied to human and to therapeutic indication in the future. In human, AML initiating cells and cell cycle quiescent AML cells may not perfectly overlap. This is one minor point that I want the authors to work on.

We sincerely thank the reviewer for this insightful comment highlighting the heterogeneity of leukemia stem cells (LSCs) in human AML and the importance of translating niche–LSC dynamics to clinical contexts. In the revised *Discussion*, we now (i) explicitly acknowledge that human leukemic initiating cells (LICs) can arise from both CD34⁺ and CD34⁻ compartments, reflecting the cellular diversity and lineage plasticity seen across AML subtypes; (ii) discuss that quiescent LSCs and LICs represent overlapping but distinct functional states—wherein dormancy, self-renewal, and therapy resistance may be uncoupled in certain patient contexts; and (iii) elaborate that our proposed DPP4–CXCL12–GPC3 axis represents a niche-governed regulatory mechanism that could be therapeutically exploited to mobilize quiescent or treatment-resistant LSCs from their protective bone marrow niches, thereby enhancing the efficacy of existing AML therapies. We believe this addition strengthens the translational relevance of our findings and acknowledges the complexity of human AML biology. Specifically, we have added:

Discussion: “Human AML exhibits substantial genetic, epigenetic, and functional diversity, with leukemia-initiating cell (LIC) activity distributed across both CD34⁺ and CD34⁻ fractions, depending on subtype.^{1,2} Moreover, quiescence and stemness do not invariably overlap, some LSCs are deeply dormant and therapy-

resistant, whereas others retain self-renewal despite active cycling^{3,4}. This diversity underscores the need to determine whether the DPP4–CXCL12–GPC3 axis governs these distinct LIC states in a similar manner.”

Genomic analyses in Response Figure 2 picked up several survival and metabolic genes differentially expressed. It would be nice to validate in the future which genes or gene sets are mainly involved, but I do not ask the authors to include the validation in the revision.

We sincerely appreciate the reviewer’s constructive and forward-looking suggestion, as well as their understanding that this analysis extends beyond the current scope. In the revised *Discussion*, we now highlight that future work will prioritize functional validation of the identified survival and metabolic pathways to determine which specific gene sets most directly regulate leukemic cell fitness and therapeutic response.

Specifically, we have added:

“In parallel, future work will prioritize functional validation of the survival and metabolic gene modules identified in our genomic analyses to clarify which pathways most directly govern LSC fitness.”

I am satisfied with response and response figures for other comments of mine such as how normal HSCs and L-GMPs share DPP4-GPC3-CXCL12 axis and leukemic infiltration in non-BM organs.

Again, in the revised ms, I feel it convincing that bone structural differences and differential niche availability between PM/DM and CM regions result in quite dynamic distinction of stem cell properties of leukemic cells. Critical molecules between niche and L-GMPs and mechanistic insights into the biology were nicely provided.

We appreciate the reviewer’s positive assessment and are grateful for the recognition of our work.

Regarding the comment by reviewer #2 on the need for transplantation assays, indeed serial transplantation assay is the most powerful method to prove “stemness” and “self-renewal capacity” of stem cells. But asking the authors to provide the in vivo assay would take too long. Instead, serial replating is a reasonable alternative good for the authors to strengthen their claim.

We will incorporate new (serial replating) and old (published in vitro and in vivo) data to prove “stemness” and “self-renewal capacity” of these LSCs.

Reviewer #2 (Remarks to the Author):

In this revised manuscript the authors have addressed the reviewers’ questions mostly in the rebuttal letter, but not all edits were introduced into the manuscript, as far as this reviewer can tell. It also seems as if not all the manuscript’s edits were clearly highlighted in another colour. This makes it difficult to evaluate the changes.

We sincerely apologize for the confusion and inconvenience caused by the previous version. Because the last revision involved extensive edits throughout the manuscript, highlighting every change made the document difficult to read and distracted from the major revisions. In this updated version, we have carefully incorporated all new edits into the main text and clearly highlighted the changes for easier evaluation.

Issues with the manuscript remain:

1. It would be helpful to add to the abstract and the introduction, how DPP4 regulates CXCL12 and not just

cite a publication (in the introduction).

We thank the reviewer for this helpful suggestion. In the revised manuscript, we have now explicitly described the mechanistic relationship between DPP4 and CXCL12 in both the *Abstract* and *Introduction*:

Abstract: “DPP4 proteolytically truncates CXCL12 at its N-terminus, functionally inactivating the chemokine, whereas stromal GPC3 inhibits DPP4 in AML cells at metaphyseal sites, thereby preserving CXCL12 activity.”

Introduction: “CXCL12 is a well-established chemoattractant for leukemic cells, playing a crucial role in cell trafficking and localization. DPP4, a membrane-bound extracellular peptidase, cleaves Xaa-Pro or Xaa-Ala dipeptides from the N terminus of substrates such as CXCL12, leading to its functional inactivation. Conversely, glypican-3 (GPC3) has been identified as an endogenous inhibitor of DPP4. However, how this DPP4–GPC3–CXCL12 regulatory axis operates within the leukemic niche, and how it collectively governs AML stem cell localization and maintenance, remains largely unexplored.

2. The LSC-intrinsic effect in DPP4 KO AML cells could be -at least partly- due to metabolic changes. This is not even discussed. Along the same lines, the conclusions do not consider alternative explanations, e.g. at the end of the text for figure 1. The only way to determine cell exhaustion is to perform serial transplantation assays, or at least serial replating assays. In some cases a conclusion is missing altogether in the results section of some figures.

We thank the reviewer for this insightful comment. In response, we now address the metabolic component

directly. Transcriptomic profiling of *Dpp4*^{-/-} versus control AML cells shows coordinated changes in multiple metabolic pathways, including retinol metabolism, glycerophospholipid remodeling, nitrogen and linoleic-acid metabolism, and oxidative/apoptotic stress responses (**Response Fig. 1a**; **new Extended Data Fig. 1e**). These signatures indicate increased metabolic and biosynthetic activity, membrane remodeling, and heightened stress-response programs. These findings support that metabolic alterations contribute to the intrinsic phenotypes observed in *Dpp4*-deficient AML cells, and this point has been incorporated into the revised manuscript. Our previous work (*Cell Reports*, 2023; PMID: 36807138) addressed DPP4's intrinsic effects in AML cells, whereas the current study focuses primarily on how DPP4 shapes the LSC-supportive microenvironment.

Specifically, "While CXCL12's role in LSC biology has been controversial⁵⁻⁸, our data reconcile these discrepancies by linking LSC exhaustion to altered CXCL12 gradients rather than chemokine abundance per se. In addition, transcriptional profiling indicates that *Dpp4* loss is accompanied by coordinated metabolic changes, which may further contribute to the intrinsic exhaustion phenotype of *Dpp4*-deficient LSCs."

Following the reviewer's guidance, we performed serial replating (serial CFU) assays as a practical surrogate for serial transplantation to functionally assess LSC exhaustion. The results confirmed a marked reduction in colony-forming capacity upon serial replating of *Dpp4*-KO AML cells (**Response Fig. 1b**; **new Fig. 1m**), consistent with our previous data (**Response Fig. 1c**, *Cell Reports*, 2023) and limiting-dilution assays showing nearly a 1000-fold decrease in functional LSC frequency (**Response Fig. 1d-e**, *Cell Reports*, 2023). Together, these findings reinforce that DPP4 is essential for maintaining LSC quiescence and preventing exhaustion.

We have also added concise, one-sentence conclusions at the end of each relevant *Results* subsection to clarify the key take-home message and avoid ambiguity in data interpretation.

For Figure 1: "These findings, combined with increased cell division (**Fig. 1i-j**), elevated apoptosis in *Dpp4* KO L-GMPs (**Fig. 1k-l**), the markedly reduced serial-replating capacity of *Dpp4* KO AML cells (**Fig. 1m**), and evidence of some heightened metabolic activity (**Extended Data Fig. 1e**), together suggest an exhaustion phenotype. Together, these results demonstrate that *Dpp4* loss redistributes AML cells from metaphyseal niches into less supportive central marrow regions, driving functional LSC exhaustion."

For Figure 3: "Overall, our data show that *Dpp4* loss traps AML cells in the BM by generating a reversed CXCL12 gradient, rather than by impairing homing, vascular access, or chemokine responsiveness."

For Figure 4: "Together, these results demonstrate that CXCL12 produced by N-cadherin⁺ MSCs is the key niche signal that anchors LSCs in the metaphysis and dictates their spatial distribution within the BM."

For Figure 7: "Thus, loss of GPC3 in N-cad⁺ MSCs disrupts CXCL12 maintenance, redistributes AML cells away from supportive niches, and improves survival."

3. Why don't the authors try the effect of DPP4 inhibitors in their model system to make a point that their findings are of translational value?

We thank the reviewer for this excellent suggestion, which indeed highlights the translational potential of our

Response Figure 2. (a) Survival curve of MLL-AF9 AML mice treated with V (n = 20 mice), Lgp (n = 12 mice), or Ara-C (n = 12 mice). (b) Survival curve of hAML cell-transplanted NSG mice treated with V or Lgp (n = 9 mice). p values were derived from a log rank test.

findings. In our previous study (*Cell Reports*, 2023; PMID: 36807138), we demonstrated that pharmacologic inhibition of DPP4 using clinically approved agents such as linagliptin significantly delayed AML progression in both the MLL-AF9 mouse model and patient-derived AML xenografts (**Response Fig. 2a–b**), though the effect was less pronounced than with genetic *Dpp4* deletion. We attribute this difference primarily to (i) the incomplete systemic inhibition achieved by small-molecule dosing compared to complete genetic loss, and (ii) the timing and pharmacokinetics of drug delivery relative to disease establishment.

Building on these findings, we are now optimizing a stable relapsed AML mouse model to evaluate the therapeutic potential of DPP4 inhibition in the post-chemotherapy residual disease setting—where we anticipate greater efficacy by targeting quiescent, niche-protected LSCs. As outlined in the *Discussion*, our future translational efforts will also explore combinatorial niche-targeting strategies, including: 1) DPP4 inhibitors (e.g., linagliptin) to modulate CXCL12 bioavailability, 2) N-cadherin⁺ MSC-targeting agents (e.g., ADH-1) to disrupt LSC anchoring, 3) CXCL12–CXCR4 blockade (e.g., AMD3100) to enhance LSC mobilization, and 4) GPC3–DPP4 interaction inhibitors (e.g., peptide 1 TFA) to destabilize the protective LSC niche.

Together, these ongoing and planned studies aim to translate our mechanistic insights into actionable therapeutic strategies capable of eradicating residual LSCs and preventing AML relapse.

4. The end of the manuscript sounds as if niche-targeting therapies in AML and other hematological malignancies has not been attempted, which is not true. The discussion as a whole is not balanced and does not put this manuscript into perspective of published work.

We appreciate the reviewer’s thoughtful comment and have substantially revised the *Discussion* to provide a more balanced and contextualized perspective. We now explicitly acknowledge the breadth of prior efforts targeting the bone marrow niche in AML and related hematologic malignancies, including CXCR4 inhibition, vascular/endosteal remodeling, and adhesion-disrupting therapies (*Nature* 2016; *Nat Cell Biol* 2020; *Nat Commun* 2020; *Front Cell Dev Biol* 2021; *Blood* 2022; *Nat Cancer* 2023).

In this revised context, we position our study as complementary to these advances—revealing a previously unrecognized stromal–protease–chemokine checkpoint (the DPP4–CXCL12 axis modulated by GPC3) that governs leukemia stem cell retention and dormancy within the bone marrow.

Specifically, we have added:

“This axis offers immediate translational promise. DPP4 inhibitors (e.g., sitagliptin, linagliptin), already FDA-

approved for diabetes^{9,10}, could be repurposed to: (1) reverse BM-PB CXCL12 gradients and confine AML

cells, reducing complications of hyperleukocytosis^{11,12}; (2) dislodge LSCs from protective niches to sensitize them to chemotherapy; and (3) induce LSC exhaustion. Building on these mechanistic insights, we have initiated preclinical testing of combination strategies, including CXCR4 antagonists (e.g., AMD3100)¹³, NF- κ B inhibitors (e.g., QNZ)¹⁴, and N-cadherin antagonist (ADH-1)¹⁵, to disrupt stromal anchoring, survival pathways, and chemokine gradients in parallel. In parallel, future work will prioritize functional validation of the survival and metabolic gene modules identified in our genomic analyses to clarify which pathways most directly govern LSC fitness. **Such efforts are aligned with the expanding field of BM niche-targeting therapies in hematopoietic malignancies¹⁶⁻²⁰, exemplified by the E-selectin antagonist uproleselan, which remodels stromal interactions and enhances hematopoietic recovery²¹. These collective advances reinforce that the BM microenvironment is a drug-responsive component of hematologic malignancies.**"

5. Colony assays: How many progenitor cells were plated? A very difficult assay to do in view of the risk of contamination of these cells in such small number during the sorting process.

We thank the reviewer for highlighting this important technical consideration. We have now clearly specified all colony assay inputs in the Methods and Figure Legends. For AML CFU assays, we plated 2,000 GFP⁺/Gr1⁺ AML cells per well in methylcellulose, with triplicate wells for each biological replicate. For serial replating, 2,000 GFP⁺/Gr1⁺ cells were re-plated at each round (every 7–10 days) using a constant input.

To ensure accuracy and minimize contamination risk, GFP⁺/Gr1⁺ AML cells were isolated by two-step high-stringency sorting, consistently achieving >98% purity, rigorous doublet exclusion, and >90% post-sort viability. Sterility controls were included for every experiment, and all assays were performed under mycoplasma-free conditions without antibiotics.

For the progenitor-subset CFU assays (**Extended Data Fig. 1d**), we plated 500 FACS-purified cells per well for each population (L-GMP, c-Kit⁺Sca1⁺, c-Kit⁺Sca1⁻II7R α ⁺, L-CMP, and L-MEP). This input is standard in published murine AML CFU assays and provides reliable colony numbers while avoiding variability associated with ultra-low cell counts.

We fully appreciate the reviewer's concern, and we hope these detailed clarifications demonstrate that the CFU assays were performed under rigorously controlled, contamination-resistant conditions.

6. Figures 1f and 1g: Why are there these differences? Some groups believe the L-GMP are the LSC in this model, while others believe that the c-Kit⁺ Sca-1⁺ fraction harbors the LSC. Therefore, how are these differences possible?

We thank the reviewer for this thoughtful comment. It is well recognized that both L-GMPs and the rarer Lin⁻ c-Kit⁺ Sca-1⁺ (LSK/KSL) fraction can harbor leukemia-initiating activity in the MLL-AF9 model, as demonstrated in the seminal work of Krivtsov et al. (*Nature*, 2006) and Somervaille & Cleary (*Cancer Cell*, 2006). These studies also showed that the two populations are immunophenotypically distinct **and isolated through different gating strategies**.

L-GMPs are derived from the Lin⁻ **Sca-1⁻** c-Kit⁺ compartment, whereas LSK/KSL cells are identified within

the Lin⁻ Sca-1⁺ c-Kit⁺ gate. Because these populations differ in frequency and marker expression, their relative abundance varies correspondingly, which explains the numerical differences shown in Figures 1f and 1g. Our calculations directly reflect the flow cytometry gating presented in Figure 1e.

In this study, we focused on L-GMPs because: 1) Prior functional transplantation studies established L-GMPs as a robust LSC-enriched population in MLL-AF9 leukemia (Nature, 2006, et.al). 2) This population can be reproducibly and unambiguously isolated by flow cytometry. 3) Our own previous work (Wang et al., Cell Reports, 2023) confirmed L-GMPs as the most reliable and experimentally tractable LSC compartment in this model.

7. LSC do not kill the mice in this model. It is the leukemic blasts. Therefore, a conclusion about aggressivity of disease cannot be based on the number of LSC (in Figure 2f).

We appreciate the reviewer's thoughtful clarification. We fully agree that lethality in the MLL-AF9 model is driven by the expansion of leukemic blasts rather than stem cell numbers alone. Accordingly, we have revised the text to avoid implying causality based solely on LSC frequency.

Importantly, our conclusions regarding disease aggressivity are not based on the LDA experiment but instead derive from the in-vivo transplantation in Figure 2e, where equivalent numbers of L-GMPs isolated from PM, CM, or DM regions were transplanted and yielded significantly different survival outcomes. These survival curves directly reflect differential disease kinetics in vivo and therefore provide a more physiologically relevant readout of aggressivity.

Specifically, we revised the result writing:

"To functionally assess niche-specific LSC properties, we transplanted equal numbers of L-GMPs isolated from PM, DM or CM regions into C57/B6 recipients. PM and DM-derived cells produced a more aggressive AML progression (**Fig. 2e**), indicating that these regions harbor LSCs with higher functional leukemogenic potential. Furthermore, limiting dilution assays showed that PM/DM-derived AML cells possessed higher stemness (**Fig. 2f**). Together, these findings indicate that the PM and DM, as opposed to the CM, constitute specialized LSC-supporting niches that promote LSC maintenance and function, as demonstrated by their in vivo leukemogenic capacity."

8. As mentioned, two markers need to be shown for leukemic cells, i.e. GFP and Gr1 or another myeloid marker. It is not sufficient to include the information in the rebuttal. The reader needs to be convinced too.

We have now moved the representative flow cytometry plots from our rebuttal to **Extended Data** Figure 1a. These plots (gating on GFP with Gr-1/Mac-1) demonstrate that nearly all GFP⁺ AML cells are Gr-1⁺ myeloid blasts, confirming GFP as a faithful marker for leukemia in this model.

9. What do the lanes in Supplementary Figure 7 represent?

We have expanded the legend for **Extended Data Figure 7** to specify sample identity, order, loading controls, and replicates for each lane of the Western blots (ERK1/2, NF- κ B p65, STAT3, p38, β -Actin). We also reference the corresponding Methods subsection for sample preparation and lane assignments.

- 1 Taussig, D. C. *et al.* Leukemia-initiating cells from some acute myeloid leukemia patients with mutated nucleophosmin reside in the CD34(-) fraction. *Blood* **115**, 1976-1984 (2010). <https://doi.org/10.1182/blood-2009-02-206565>
- 2 Kikushige, Y. *et al.* Self-renewing hematopoietic stem cell is the primary target in pathogenesis of human chronic lymphocytic leukemia. *Cancer cell* **20**, 246-259 (2011). <https://doi.org/10.1016/j.ccr.2011.06.029>
- 3 van Galen, P. *et al.* Single-Cell RNA-Seq Reveals AML Hierarchies Relevant to Disease Progression and Immunity. *Cell* **176**, 1265-1281.e1224 (2019). <https://doi.org/10.1016/j.cell.2019.01.031>
- 4 Méndez-Ferrer, S. *et al.* Bone marrow niches in haematological malignancies. *Nature reviews. Cancer* **20**, 285-298 (2020). <https://doi.org/10.1038/s41568-020-0245-2>
- 5 Ramakrishnan, R. *et al.* CXCR4 Signaling Has a CXCL12-Independent Essential Role in Murine MLL-AF9-Driven Acute Myeloid Leukemia. *Cell Rep* **31**, 107684 (2020). <https://doi.org/10.1016/j.celrep.2020.107684>
- 6 Anderson, N. R. *et al.* CXCL12 Knock-out Enhances Leukemia Stem Cell Response to Combination Chemotherapy Plus Tyrosine Kinase Inhibition in Flt3-ITD Acute Myeloid Leukemia. *Blood* **136**, 7-8 (2020). <https://doi.org/https://doi.org/10.1182/blood-2020-142579>
- 7 Zeng, Z. *et al.* Targeting the leukemia microenvironment by CXCR4 inhibition overcomes resistance to kinase inhibitors and chemotherapy in AML. *Blood* **113**, 6215-6224 (2009). <https://doi.org/10.1182/blood-2008-05-158311>
- 8 Agarwal, P. *et al.* Mesenchymal Niche-Specific Expression of Cxcl12 Controls Quiescence of Treatment-Resistant Leukemia Stem Cells. *Cell Stem Cell* **24**, 769-784.e766 (2019). <https://doi.org/10.1016/j.stem.2019.02.018>
- 9 Havale, S. H. & Pal, M. Medicinal chemistry approaches to the inhibition of dipeptidyl peptidase-4 for the treatment of type 2 diabetes. *Bioorg Med Chem* **17**, 1783-1802 (2009). <https://doi.org/10.1016/j.bmc.2009.01.061>
- 10 Wang, C. *et al.* Dipeptidylpeptidase 4 promotes survival and stemness of acute myeloid leukemia stem cells. *Cell Rep* **42**, 112105 (2023). <https://doi.org/10.1016/j.celrep.2023.112105>
- 11 Cline, A., Jajosky, R., Shikle, J. & Bollag, R. Comparing leukapheresis protocols for an AML patient with symptomatic leukostasis. *J Clin Apher* **33**, 396-400 (2018). <https://doi.org/10.1002/jca.21588>
- 12 Del Prete, C., Kim, T., Lansigan, F., Shatzel, J. & Friedman, H. The Epidemiology and Clinical Associations of Stroke in Patients With Acute Myeloid Leukemia: A Review of 10,972 Admissions From the 2012 National Inpatient Sample. *Clin Lymphoma Myeloma Leuk* **18**, 74-77 e71 (2018). <https://doi.org/10.1016/j.clml.2017.09.008>
- 13 Nervi, B. *et al.* Chemosensitization of acute myeloid leukemia (AML) following mobilization by the CXCR4 antagonist AMD3100. *Blood* **113**, 6206-6214 (2009). <https://doi.org/10.1182/blood-2008-06->

- 14 Marciano, R. *et al.* High-Throughput Screening Identified Compounds Sensitizing Tumor Cells to Glucose Starvation in Culture and VEGF Inhibitors In Vivo. *Cancers (Basel)* **11** (2019). <https://doi.org/10.3390/cancers11020156>
- 15 Khorsand, M. *et al.* Telmisartan anti-cancer activities mechanism through targeting N-cadherin by mimicking ADH-1 function. *Journal of Cellular and Molecular Medicine* **26**, 2392-2403 (2022). <https://doi.org/https://doi.org/10.1111/jcmm.17259>
- 16 Roversi, F. M., Bueno, M. L. P., Pericole, F. V. & Saad, S. T. O. Hematopoietic Cell Kinase (HCK) Is a Player of the Crosstalk Between Hematopoietic Cells and Bone Marrow Niche Through CXCL12/CXCR4 Axis. *Front Cell Dev Biol* **9**, 634044 (2021). <https://doi.org/10.3389/fcell.2021.634044>
- 17 Grockowiak, E. *et al.* Different niches for stem cells carrying the same oncogenic driver affect pathogenesis and therapy response in myeloproliferative neoplasms. *Nat Cancer* **4**, 1193-1209 (2023). <https://doi.org/10.1038/s43018-023-00607-x>
- 18 Haltalli, M. L. R. & Lo Celso, C. Targeting adhesion to the vascular niche to improve therapy for acute myeloid leukemia. *Nat Commun* **11**, 3691 (2020). <https://doi.org/10.1038/s41467-020-17594-7>
- 19 Haltalli, M. L. R. *et al.* Manipulating niche composition limits damage to haematopoietic stem cells during Plasmodium infection. *Nature cell biology* **22**, 1399-1410 (2020). <https://doi.org/10.1038/s41556-020-00601-w>
- 20 Hawkins, E. D. *et al.* T-cell acute leukaemia exhibits dynamic interactions with bone marrow microenvironments. *Nature* **538**, 518-522 (2016). <https://doi.org/10.1038/nature19801>
- 21 DeAngelo, D. J. *et al.* Phase 1/2 study of uproleselan added to chemotherapy in patients with relapsed or refractory acute myeloid leukemia. *Blood* **139**, 1135-1146 (2022). <https://doi.org/10.1182/blood.2021010721>

Response to the Editor and the Reviewers

We thank the Editor and the Reviewers for their thorough and insightful feedback. In response, we have carefully revised the manuscript and performed additional experiments as requested. Specifically, we have reorganized Fig. 1 and Extended Data Fig. 1, updated the associated figure legends, and refined the text in Results section 1 and the CFU methodology. All changes are highlighted in the revised manuscript.

Reviewers' Comments:

Reviewer #1 (Remarks to the Author):

The authors nicely presented importance of DPP4-GPC3-CXCL12 axis and differential availability of niche between PM/DM and CM regions. Potential translation of the findings has been nicely discussed as well.

I fully appreciate all the effort by the authors to revise the manuscript.

ADDITIONAL COMMENTS ON THE REMAINING ISSUE BY REVIEWER #2 AND THE AUTHORS' RESPONSE TO IT

Expression of Gr1 in mouse AML models is fine. In particular for MLL-AF9 AML model, in addition to references that authors listed, I found another showing Gr1+ MLL-AF9 AML (PMCID PMC3951989). "MN1-CMP leukemias showed a high proportion of myeloid blasts in bone marrow (Figure 1D), infiltrated spleen (Figure 1E and F) and liver (data not shown), and had an immature immunophenotype (high c-kit expression) with some expression of myeloid markers Gr-1 and CD11b, similar to MN1 leukemias generated by transduction of 5-FU treated bulk bone marrow cells (Figure 1G)."

For MLL-AF9, we sometimes find CD34(-) LICs in human as well.

However, it is a little confusing even to me that transplantation of LSK resulted in Gr1+ AML colonies. LSK stands for Lin(-)Sca1+Kit+. If LICs (leukemia-initiating-cells) express Gr1, a critical lineage marker for mouse hematopoiesis, the authors should lose all the leukemic cells.

Reviewer #2 (Remarks to the Author):

In this second revision the authors have addressed the reviewers' concerns. However, the authors claim that the LSC fraction is harbored in the LSK or the L-GMP fraction. Then how is it possible that sorted AML cells positive for Gr1, a marker for mature and immature myeloid cells, give rise to colonies in a colony-forming assay, which assesses the function of leukemic stem and progenitor cells?

ADDITIONAL COMMENTS ON THE AUTHORS' RESPONSE TO MY REMAINING ISSUE ABOVE

In their first reference (Krivtsov et al), which is the manuscript which identified the MLL-AF9 model, I found no reference to Gr1 or Mac1 as a marker for leukaemia stem cells (LSC) (as they identify the L-GMP cells as the LSC). In the second manuscript by Somerville et al. I will admit that they mention LSC activity in the Gr1+ or Mac1+ fraction.

Therefore, Wang et al. may be correct. However, it is VERY unusual to claim LSC activity or perform CFU-assays with Mac1+ or Gr-1+ cells. I do not agree, that Mac1 or Gr1+ cells are widely believed to have LSC activity. Why did they not use L-GMP or LSK cells, as they claim these to be the LSC in their paper? My suggestion would be that the authors perform a CFU assay with L-GMP and LSK cells to prove their point.

We sincerely appreciate Reviewers' further comments on the description and phenotype of LSCs. We recognize that the exchange during the previous rebuttal rounds—specifically regarding the work on

the GFP⁺Kit⁺Sca1⁺ populations—introduced ambiguity regarding our LSC phenotype and description. To clarify this, we have modified the text regarding the description of the populations.

- 1. Clarification regarding the "SK" vs. "LSK" populations and gating strategy.** We wish to clarify that our study primarily relies on the **L-GMP** population as the validated LSC marker. The confusion regarding the "LSK" terminology stems from data generated during the previous revision in response to Reviewer #1's suggestion on the "**c-Kit⁺Sca1⁺ cells, c-Kit⁺Sca1(-) Il7Ra⁺ cells**". The data we provided in response utilized a "SK" gate (**Fig 1g** "GFP⁺Kit⁺Sca1⁺ cells"), which, in this specific model, contains a high proportion of Lin⁺ cells due to aberrant marker expression. To avoid potential confusion, we renamed the y-axis in the original Fig. 1g to "SK" to accurately reflect the gating and moved these data (original Fig. 1g-h) to Extended Data Fig. 1b-c to differentiate them from our central LSC analysis.
- 2. Validation of L-GMP as the primary LSC Population (rationale and new functional data).** We thank the reviewer for the insightful comment regarding LSC markers. It is important to note that the classical 'LSK' population is effectively absent in MLL-AF9 AML, as the immunophenotype shifts dramatically during leukemic transformation. As shown in Somerville et al. (their original Fig. 3C, **Response Figure 1A**) and confirmed by our own analysis (**Response Figure 1B**), the "LSK" population (II) is virtually non-existent. Accordingly, we have ensured the revised manuscript removes references to the LSK fraction as an LSC source, focusing strictly on the L-GMP, which is the biologically relevant and experimentally validated population in our study. To definitively validate this focus—and as requested—we performed serial CFU assays specifically on sorted L-GMP cells. These new data (**Response Figure 1C, New Fig. 1g**) confirm a significant reduction in serial

Response Figure 1. Validation of L-GMP as the primary LSC Population. (A) Assessment of the true LSK compartment in MLL-AF9 AML. Representative FACS profiles showing c-kit⁺/Sca1⁺ gating on Lin⁻ gated splenocytes from a mouse with MLL-AF9 AML. A very small fraction of donor (EGFP⁺) cells in this example is Lin⁻, and these cells essentially lack a defined LSK immunophenotype (Region II). Rare cells are present in the progenitor gate (c-kit⁺ Sca1⁻). (Adapted from Somerville et al., *Cancer Cell* 2006, Figure 3C). (B) Representative FACS profiles of bone marrow cells from a control MLL-AF9 AML mouse from our own analysis, confirming the extreme rarity of the Lin⁻Sca1⁺c-Kit⁺ population. (C) Serial CFU assays performed using 500 sorted L-GMPs per replating round. The results demonstrate a marked reduction in serial colony-forming capacity in Dpp4^{-/-} L-GMPs compared to Dpp4^{+/+} controls. Data represent mean ± SEM (n = 6, p < 0.0001).

colony-forming capacity in Dpp4-KO L-GMPs (confirming the CFU method improvements in the manuscript).

3. **Acknowledgment of LSC phenotypic plasticity and rationale for L-GMP selection.** We appreciate the reviewer highlighting the phenotypic diversity of LSCs in the MLL-AF9 model, particularly the expression of myeloid markers such as Gr1 and CD11b/Mac1 (PMCID PMC3951989). We agree that the precise immunophenotype of these LSCs has been a subject of

Response Figure 2. Distinct functional properties and hierarchy of leukemic populations in MLL-AF9 AML. (A) CFU assay revealed that L-GMP possessed higher CFU activity than other leukemic populations (Adapted from Krivtsov et al., *Nature* 2006, Supplementary Figure 3). (B) Morphological analysis of leukemic cells sorted based on c-Kit expression. The bar graph indicates the proportion of cells exhibiting blast-like versus differentiated features following May-Grünwald-Giemsa staining of cytopsin preparations (Adapted from Somervaille et al., *Cancer Cell* 2006, Figure 4B). (C) Pie chart shows the percentage of clonogenic cells residing within the c-kit⁺ versus c-kit⁻ fractions of EGFP⁺ leukemic cells (n = 6). FACS-sorted leukemic cells were cultured in methylcellulose medium for 6 days to determine the CFU frequencies in each fraction (29.2% ± 3.5% and 5.4% ± 2.0%, respectively; p < 0.001) (Adapted from Somervaille et al., *Cancer Cell* 2006, Figure 4C). (D) Bar graph indicates an average 25% shorter survival across varying cell doses for mice secondarily transplanted with c-kit⁺ AML cells versus recipients of equivalent numbers of c-kit⁻ cells. Disease latencies shown are the means of two separate experiments using sorted cells from separate donor animals with AML, one recipient animal per indicated cell dose. The numbers of transplanted cells are indicated on the left (k = 1000) (Adapted from Somervaille et al., *Cancer Cell* 2006, Figure 4D).

nance in the field, and we interpret this diversity not as a contradiction, but as evidence of the phenotypic plasticity inherent to this leukemia. As the reviewer noted, while Krivtsov et al. (*Nature*, 2006) identified the L-GMP as the highly enriched, self-renewing LSC population, they also observed that colony-forming potential was not strictly limited to this fraction (Krivtsov et al., Supplementary Fig. 3; **Response Fig. 2A**). Similarly, Somervaille et al. (*Cancer Cell*, 2006) demonstrated that functional LSC activity is enriched in Kit⁺GR1⁺Mac1⁺ populations (**Response Fig. 2B-D**), further confirming that LSCs can retain myeloid lineage characteristics. This phenomenon parallels human AML, where LSCs (particularly in MLL-rearranged cases) frequently display aberrant lineage markers and inter-patient variability (PMCID: PMC3007135, PMC4319237). Interestingly, such lineage upregulation (e.g., Mac1) has even been reported in normal HSCs under stress conditions (PMCID: PMC10031298, <https://doi.org/10.1016/j.exphem.2019.06.428>). However, because highly plastic populations can be variable, they present challenges for defining a precise molecular mechanism. Therefore, while we acknowledge the broader immunophenotypic spectrum described by Somervaille and others, we aligned our study with the Krivtsov model, utilizing the **L-GMP** as the most stable, highly enriched, and reproducible definition of the LSC for mechanistic interrogation.